# Major role of ammonia-oxidizing bacteria in $N_2O$ production in the Pearl River Estuary

Li Ma[1,2], Hua Lin[1,2,3], Xiabing Xie[1], Minhan Dai[1,2], Yao Zhang[1,2]

[1]State Key Laboratory of Marine Environmental Science, Xiamen University, Xiamen 361101, China

[2]College of Ocean and Earth Sciences, Xiamen University, Xiamen 361101, China

[3]Key Laboratory of Marine Ecosystem and Biogeochemistry, State Oceanic Administration, Second Institute of Oceanography, Ministry of Natural Resources, Hangzhou 310012, China

*Correspondence to*: Yao Zhang (yaozhang@xmu.edu.cn)

**Abstract.** Nitrous oxide ($N_2O$) has significant global warming potential as a greenhouse gas. Estuarine and coastal regimes are the major zones of $N_2O$ production in the marine system. However, knowledge on biological sources of $N_2O$ in estuarine ecosystems remain controversial, but are of great importance for understanding global $N_2O$ emission patterns. Here, we measured concentrations and isotopic compositions of $N_2O$ as well as distributions of ammonia-oxidizing bacterial and archaeal *amo*A and denitrifier *nir*S genes by quantitative polymerase chain reaction along a salinity gradient in the Pearl River Estuary, and performed in situ incubation experiments to estimate $N_2O$ yields. Our results indicated that nitrification predominantly occurred, with significant $N_2O$ production during ammonia oxidation. In the hypoxic waters of the upper estuary, strong nitrification resulted in the observed maximum $N_2O$ and $\Delta N_2O_{excess}$ concentrations, although minor denitrification might be concurrent at the site with the lowest dissolved oxygen. Ammonia-oxidizing $\beta$-proteobacteria (AOB) were significantly positively correlated with all $N_2O$-related parameters, although their *amo*A gene abundances were distinctly lower than ammonia-oxidizing Archaea (AOA) throughout the estuary. Furthermore, the $N_2O$ production rate and the $N_2O$ yield normalized to *amo*A gene copies or transcripts estimated a higher relative contribution of AOB to the $N_2O$ production in the upper estuary. Taken together, the in situ incubation experiments, $N_2O$ isotopic composition and concentrations, and gene datasets suggested that the high concentration of $N_2O$ (oversaturated) is mainly produced from strong nitrification by the relatively high abundance of AOB in the upper reaches and is the major source of $N_2O$ emitted to the atmosphere in the Pearl River Estuary.

# 1 Introduction

Nitrous oxide ($N_2O$) is a potent greenhouse gas with global warming potential 298 times that of carbon dioxide ($CO_2$) on a 100 yr timescale, and contributes to stratospheric ozone depletion as a major precursor of free radicals (Ravishankara et al., 2009). $N_2O$ emissions from soils and marine systems are estimated to account for 56%–70% (6–7 Tg $N_2O$-N $yr^{-1}$) (Syakila and Kroeze, 2011; Butterbach-Bahl et al., 2013; Hink et al., 2017) and 30% (4 Tg $N_2O$-N $yr^{-1}$) (Nevison et al., 2004; Naqvi et al., 2010; Voss et al., 2013) of the total global $N_2O$ emissions, respectively. The main processes responsible for $N_2O$ emissions are microbial transformation of ammonia, nitrite, and nitrate through nitrification and denitrification (Butterbach-Bahl et al., 2013). It has been estimated that oceanic $N_2O$ production is dominated by nitrification, whereas only 7% is contributed by denitrification (Freing et al., 2012).

$N_2O$ is released as a byproduct during nitrification via incomplete oxidation of hydroxylamine ($NH_2OH$) to nitrite ($NO_2^-$) by ammonia-oxidizing bacteria (AOB) (Stein, 2011). This process may be enhanced under suboxic conditions (Naqvi et al., 2010). While no equivalent of the hydroxylamine-oxidoreductase that catalyzes $N_2O$ formation through $NH_2OH$ oxidation has been found in ammonia-oxidizing archaea (AOA) (Hatzenpichler, 2012), recent studies indicated that AOA possibly produces hybrid $N_2O$ via a combination of an ammonia oxidation intermediate ($NH_2OH$, HNO, or NO) and $NO_2^-$ (Stieglmeier et al., 2014; Frame et al., 2017). In addition, AOB have been shown to produce $N_2O$ from $NO_2^-$ during nitrifier denitrification (Shaw et al., 2006). This process is also promoted under micro-oxic and anoxic conditions (Yu et al., 2010). Denitrification by heterotrophic denitrifiers is another major pathway of $N_2O$ production in marine environments, occurring under anoxic conditions or at the suboxic–anoxic interface (Naqvi et al., 2010; Yamagishi et al., 2007; Ji et al., 2018). $NO_2^-$ is reduced by a copper-containing (NirK) or cytochrome cd1-containing nitrite reductase (NirS) to nitric oxide (NO), and then by a heme-copper NO reductase (NOR) to $N_2O$ (Coyne et al., 1989; Treusch et a1., 2005; Abell et al., 2010; Bartossek et a1., 2010; Lund et a1., 2012; Graf et al., 2014). As an intermediary product during denitrification, production and further reduction of $N_2O$ are sensitive to different $O_2$ conditions (Babbin et al., 2015; Ji et al., 2015).

Biological nitrogen transformations are catalyzed by various microbial enzymes, of which ammonium monooxygenase (AMO) and nitrite reductases (NIRs) are key enzymes responsible for

nitrification and denitrification, respectively (Canfield et al., 2010). The genes encoding for AMO subunit A (*amo*A) and NIRs (*nir*S and *nir*K) have been widely applied as functional marker genes to identify the distribution of ammonia oxidizers and denitrifiers. Previous studies have shown significant correlations of *amo*A with spatial variations of $N_2O$ emissions or $N_2O$ production rates in soils and oceans (Avrahami and Bohanann, 2009; Santoro et al., 2011; Löscher et al., 2012). In addition, significant relationships between *nir*K or *nir*S abundances and $N_2O$ emissions were observed in grasslands (Čuhel et al., 2010), arable soils (Clark et al., 2012; Jones et al., 2014), and the ocean (Arévalo-Martínez et al., 2015).

Estuaries are highly impacted by coastal nutrient pollution and eutrophication because of anthropogenic activity; they play a significant role in nitrogen cycling at the land–sea interface (Bricker et al., 2008; Damashek et al., 2016; Damashek and Francis, 2018). Estuarine and coastal regimes have long been recognized as major zones of $N_2O$ production in the marine system (Seitzinger and Kroeze, 1998; Mortazavi et al., 2000; Usui et al., 2001; Kroeze et al., 2010; Allen et al., 2011). In particular, eutrophic estuaries with extensive oxygen-deficient zones have been considered hotspot regions for $N_2O$ production (Abril et al., 2000; De Wilde and De Bie, 2000; Garnier et al., 2006; Lin et al., 2016), with oversaturated $N_2O$ and high $N_2O$ concentrations and flux (De Wilde and De Bie, 2000; De Bie et al., 2002; Garnier et al., 2006; Rajkumar et al., 2008; Barnes and Upstill-Goddard, 2011; Lin et al., 2016). The dynamics of $N_2O$ emissions in these ecosystems are regulated by complex physical and biogeochemical processes; for example, mixing between freshwater and oceanic waters influences the biogeochemistry of estuarine waters as well as microbial activity (Huertas et al., 2018; Laperriere et al., 2019).

Nitrification is often credited as the dominant $N_2O$ production pathway in estuaries (De Bie et al., 2002; Barnes and Upstill-Goddard, 2011; Kim et al., 2013; Lin et al., 2016; Huertas et al., 2018; Laperriere et al., 2019). Although AOA frequently outnumber AOB and dominate microbial communities, their contribution to nitrification remains controversial in estuarine and coastal waters (Bernhard et al., 2010; Zhang et al., 2014; Hou et al., 2018). Furthermore, the relative contributions of AOB and AOA to $N_2O$ production is inconclusive (Monteiro et al., 2014) and there is a potential niche overlap between nitrifiers and denitrifiers in low oxygen conditions (Frame and Casciotti, 2010; Zhang

et al., 2014; Penn et al., 2016). AOB are reported to thrive in hypoxic environments and denitrification in the oxic ocean is suggested to occur within anoxic particle interiors (Frame and Casciotti, 2010; Ni et al., 2014). It is therefore of great importance to elucidate the biological sources of $N_2O$ production in estuarine ecosystems to better understanding global $N_2O$ emission patterns.

The Pearl River Estuary, surrounded by several big cities, is one of the world's most complex estuarine systems with a total discharge of $285.2 \times 10^9$ $m^3$ $yr^{-1}$ (Dai et al., 2014). A rich nitrogen supply with the river discharge produces eutrophic waters in the estuary (Dai et al., 2008). Moreover, increased oxygen consumption by organic matter degradation leads to the formation of hypoxic zones in the upper reaches of the estuary (Dai et al., 2006; He et al., 2014), which may support strong nitrification, denitrification, and $N_2O$ production (Lin et al., 2016). In this study, $N_2O$-related biogeochemical parameters were measured, and distributions of AOB and AOA *amo*A and denitrifier *nir*S genes were quantified by quantitative polymerase chain reaction (qPCR) to investigate the relationship between $N_2O$ production and spatial distribution of nitrifiers and denitrifiers along a salinity gradient in the Pearl River Estuary (Fig. 1). Moreover, in situ incubation experiments were performed in the hypoxic upper estuary to estimate (1) nitrification and $N_2O$ production rates, (2) whether denitrification occurred during nitrification, and (3) $N_2O$ yield (mol $N_2O$-N produced per mol ammonia oxidized). By combining the genetic datasets and incubation estimates, this study thus identified the relative contributions of AOB and AOA in producing $N_2O$ in the Pearl River Estuary.

## 2 Materials and methods

## 2.1 Study area and sampling

A total of 22 sites along the salinity gradient of the Pearl River Estuary were sampled during a research cruise in July 2015, including 11 sites in the upper reaches (upstream of the Humen outlet) and 11 sites in the lower reaches (Lingdingyang) (Fig. 1). Water samples were taken from the surface (2 m) and bottom (4–15 m) of each site by using a conductivity, temperature, and depth (CTD) rosette sampling system (SBE 25; Sea-Bird Scientific, USA) fitted with 12 L Niskin bottles (General Oceanics). A total of 16 samples (from two depths at eight sites) were subjected to gene analysis (Fig. 1). A total of 1 L of

water for gene analysis was serially filtered through 0.8 μm and then 0.22 μm pore size polycarbonate membrane filters (47 mm diameter, Millipore) within 30 min at a pressure <0.03 MPa to retain the particle-associated communities (>0.8 μm) and free-living communities (0.22–0.8 μm). For the upper estuary samples, more membrane filters were used to avoid the filters clogging. RNAlater solution (Ambion, Austin, Texas, USA) was quickly added to the samples to prevent RNA degradation. All of the filters were immediately flash frozen in liquid nitrogen and then stored at −80 ℃ until further analysis. Water samples for nutrient determination were filtered through 0.45 μm pore size cellulose acetate membranes and then immediately frozen at −20 ℃ until further analysis. Water samples for dissolved $N_2O$ were collected into 125 mL headspace glass bottles to which 100 μL of saturated $HgCl_2$ was added; the bottles were immediately closed with rubber stoppers and aluminum crimp-caps and stored in the dark at 4 ℃ until analysis in the laboratory. All $N_2O$ samples were collected during the July 2015 cruise except for samples from sites P03, P05, A01, A06, and A10 intended for $N_2O$ isotopic composition analyses, which were sampled during a cruise in March 2010. Total suspended material (TSM) was collected by filtering 1–4 L of water onto pre-combusted and pre-weighed glass fiber filters (GF/Fs) (Whatman), and then stored at −20 ℃ until weighing in the laboratory.

## 2.2 Biogeochemical parameters, $N_2O$ emissions and isotopic analysis of environmental samples

Temperature and salinity were measured with the SBE 25 CTD system. Dissolved oxygen (DO) concentrations were measured using the Winkler method (Dai et al., 2006). Ammonia was measured using the indophenol blue spectrophotometric method (Pai et al., 2001) on board; nitrate, nitrite, and silicate were analyzed using routine spectrophotometric methods with a Technicon AA3 Auto-Analyzer (Bran-Lube, GmbH) (Han et al., 2012). $N_2O$ concentrations were analyzed by gas chromatography (GC, Agilent 6890 μECD) coupled with a purge-trap system (Tekmar Velocity XPT) at 25 ℃ (Lin et al., 2016). $N_2O$ standard gases of 1.02 and 2.94 ppmv $N_2O/N_2$ (National Center of Reference Material, China, Beijing) were used. The relative standard deviation of the slope of the standard working curve was 1.77% (n=8). The detection limit was calculated to be ~0.1 nmol $L^{-1}$ and the precision was better than ±5%. When water samples were analyzed, every 5−10 samples were spiked with $N_2O$ standards to calibrate the GC.

The excess N$_2$O ($\Delta$N$_2$O$_{excess}$) and N$_2$O saturation (S%) were calculated with Eq. (1) and (2):
$\Delta$N$_2$O$_{excess}$ = N$_2$O$_{observed}$ − N$_2$O$_{equilibrium}$ (1)
S% = N$_2$O$_{observed}$ / N$_2$O$_{equilibrium}$ ×100% (2)
where N$_2$O$_{observed}$ represents the measured concentrations of N$_2$O in the water, and the equilibrium
values of N$_2$O (N$_2$O$_{equilibrium}$) were calculated by Eq. (3) and (4) (Weiss and Price, 1980):
N$_2$O$_{equilibrium}$ = $xF$ (3)
$\ln F = A_1 + A_2(100/T) + A_3 \ln(T/100) + A_4(T/100)^2 + S[B_1+B_2(T/100) + B_3(T/100)^2]$ (4)
where $x$ is the mole fraction of N$_2$O in the atmosphere and T is the absolute temperature. In this study,
we used the global mean atmospheric N$_2$O (327 ppb) from 2015 (http://www.esrl.noaa.gov/gmd). The
fitted function F and constants A1, A2, A3, A4, B1, B2 and B3 were proposed by Weiss and Price

11   (1980).

12       The N$_2$O flux ($F_{N_2O}$, μmol m$^{-2}$ d$^{-1}$) through the air–sea interface was estimated based on Eq. (5):

$F_{N_2O} = k_{N_2O} \times \rho \times K_H^{N_2O} \times \Delta p N_2O = k_{N_2O} \times 24 \times 10^{-2} \times$ (N$_2$O$_{observed}$ − N$_2$O$_{equilibrium}$) (5)
where $k_{N_2O}$ (cm h$^{-1}$) is the N$_2$O gas transfer velocity depending on wind and water temperature, $K_H^{N_2O}$ is
the solubility of N$_2$O, and $\Delta p$N$_2$O is the average sea–gas N$_2$O partial pressure difference. $k_{N_2O}$ was
estimated using Eq. (6) according to Wanninkhof (1992):
$k_{N_2O} = 0.31 \times u_{av}^2 \times (Sc_{N_2O}/600)^{-0.5}$ (6)
where u$_{av}$ is the average wind speed 10 m above the water surface. In this study, a CO$_2$ Schmidt number
(Sc) of 600 at 20 ℃ in fresh water (Wanninkhof, 1992) was used for estuarine systems (Raymond and
Cole, 2001). The Sc is defined as the kinematic viscosity of water divided by the diffusion coefficient of
the gas and calculated from temperature (Wanninkhof, 1992). For N$_2$O in waters with salinities <35 and
temperatures ranging from 0−30 ℃, Sc$_{N_2O}$ was estimated using Eq. (7) according to Wanninkhof (1992):
$Sc_{N_2O} = 2055.6 − 137.11\, t + 4.3173\, t^2 − 0.05435\, t^3$ (7)
where t is the in situ temperature of the sampling site.

25       To determine the isotopic composition of N$_2$O, the gas samples were introduced into a trace gas

cryogenic pre-concentration device (PreCon, Thermo Finnigan), as described in Cao et al. (2008) and
Zhu et al. (2008), and then $\delta^{15}$N-N$_2$O was analyzed using an isotope ratio mass spectrometer (IRMS,
Thermo Finngan MAT-253, Bremen, Germany). The molecular ions of $N_2O$ ($N_2O^+$, m/z 44, 45 and 46)
were quantified by IRMS to calculate isotope ratios for the entire molecule ($^{15}N/^{14}N$ and $^{18}O/^{16}O$). The
$\delta^{15}N$ values of $N_2O$ in samples were calculated using the $^{15}N/^{14}N$ of the pure $N_2O$ reference gas and
samples (Frame and Casciotti, 2010; Mohn et al., 2014). The reference gas was previously calibrated
against $N_2O$ isotopic standard gas ($\delta^{15}N$ (vs Air-$N_2$) = −0.320‰) produced by Shoko Co. Ltd. (Tokyo,
Japan) and the $\delta^{15}N$ value (vs Air-$N_2$) of the $N_2O$ reference gas is 6.579±0.030‰. The precision of the
method for $\delta^{15}N$-$N_2O$ was estimated as 0.3‰.

## 2.3 Nucleic acid extraction and qPCR

DNA was extracted using the FastDNA$^{TM}$ SPIN Kit for Soil (MP, USA) according to the
manufacturers' protocol with minor modifications. RNA was extracted using TRIzol reagent (Ambion,
Austin, Texas, USA), and then eluted with 50 μL of RNase-free water. The extracted RNA was treated
with DNase I (Invitrogen, Carlsbad, CA) to remove any residual DNA. DNA contamination was
checked by amplifying the bacterial 16S rRNA genes before reverse transcription. Total RNA without
DNA contamination was reverse transcribed to synthesize single-strand complementary DNA (cDNA)
using the First-Strand cDNA Synthesis Kit (Invitrogen, Austin, Texas, USA).
The transcript and copy abundances of bacterial and archaeal *amo*A genes and bacterial *nir*S genes
were examined using qPCR and a CFX96 Real Time PCR system (BIO-RAD, Singapore). The *β*-
proteobacterial and archaeal *amo*A were amplified using primer sets amoA-1F and amoA-2R (Kim et
al., 2008) and Arch-amoAF and Arch-amoAR (Francis et al., 2005), respectively; *nir*S was amplified
using primers nirS-1F and nirS-3R (Braker et al., 1998; Huang et al., 2011). Quantitative PCR
amplification for the *β*-proteobacterial and archaeal *amo*A was carried out as described previously
(Mincer et al., 2007; Hu et al., 2011). For the amplification of *nir*S, the qPCR reaction mixture was
prepared in accordance with Zhang et al. (2014) and thermal cycling conditions were as described in
Huang et al. (2011). Standards for the qPCR reactions consisted of serial 10-fold dilutions ($10^7$ to $10^0$
copies per uL) of plasmid DNA containing amplified fragments of the targeted genes (accession
numbers MH458281 for *β*-proteobacterial *amo*A, KY387998 for archaeal *amo*A and KF363351 for
*nir*S). The amplification efficiencies of qPCR were always between 85%–95% with $R^2$ >0.99. The

specificity of the qPCR reactions was confirmed by melting curve analysis, agarose gel electrophoresis and sequencing analysis. Inhibition tests were performed by 2-fold and 5-fold dilutions of all samples and indicated that our samples were not inhibited.

## 2.4 Incubation experiments

Incubation experiments were performed in the surface and bottom waters at sites P01 (2 and 5 m water depth) and P05 (2 and 12 m) upstream of the Humen outlet (Fig. 1). Water samples were collected from Niskin bottles through a clean polytetrafluoroethylene (Teflon) silicone hose, and carefully filled into 125 mL clean headspace glass bottles without gas bubbles. The bottles were immediately closed with an air-tight butyl rubber stopper and aluminum crimp-cap. A total of 43 bottles were set up for surface and bottom at sites P01 and 34 bottles at P05. Samples from four parallel bottles were taken to determine the initial ($t_0$) dissolved $N_2O$ concentration, and triplicate samples were taken to measure the initial dissolved inorganic nitrogen (DIN) concentration, which included ammonium, nitrite, and nitrate. The remaining 36 (P01) and 27 (P05) bottles were incubated in the dark at in situ temperatures ($\pm 1$ ℃). At site P01, samples from six parallel bottles were taken at 3, 6, 18, and 24 h during the incubation experiment for $N_2O$ determination after injecting saturated mercuric chloride ($HgCl_2$, 1:100 v:v) into the bottles; triplicate samples were also taken at the same time for DIN measurements by filtering through 0.7 μm pore size GF/Fs under pressure <0.03 MPa. Concentrations of $N_2O$, ammonium, nitrite, and nitrate were measured as described in Sect. 2.2. At site P05, samples were taken after 3, 6, and 12 h incubation and the other procedures were the same as described for site P01.

The effect of DIN assimilation is negligible during incubation in the dark (Ward, 2008). Therefore, the potential processes of nitrogen transformation and $N_2O$ production can be determined according to "mass balance" in a closed incubation system. The main processes were analyzed based on the dynamic variations of DIN ($\Delta$DIN), ammonia ($\Delta NH_3+NH_4^+$), nitrite ($\Delta NO_2^-$), nitrate ($\Delta NO_3^-$), and $N_2O$ ($\Delta N_2O$) concentrations during incubation. The average rates of nitrification and $N_2O$ production were estimated using the slopes of the linear regression between concentrations versus incubation time when DIN was in balance (i.e. no denitrification). All of the concentration-based rates described from the incubations represent net rates. The $N_2O$ yield during nitrification was calculated with Eq. (8):

$N_2O_{yield} (\%) = \Delta N_2O\text{-}N / \Delta(NH_3 + NH_4^+)\text{-}N$ (8)

## 2.5 Statistical analyses

Since a normal distribution of the individual data sets was not always met, we used the non-parametric
Wilcoxon rank-sum tests for comparing two variables. The bivariate correlations between
environmental factors and functional genes were described by Spearman correlation coefficients ($\rho$
value). False discovery rate-based multiple comparison procedures were applied to evaluate the
significance of multiple hypotheses and identify truly significant comparisons (False discovery rate-
adjusted $P$ value) (Pike, 2011). The maximum gradient length of detrended correspondence analysis
was shorter than 3.0, thus redundancy analysis based on the qPCR data was used to analyze variations
in the AOA and AOB distributions under environmental constraints in the software R (version 3.4.4)
Vegan 2.5–3 package. The qPCR-based relative abundances and environmental factors were normalized
via Z transformation (Magalhães et al., 2008). The null hypothesis, that the community was independent
of environmental parameters, was tested using constrained ordination with a Monte Carlo permutation
test (999 permutations). Significant environmental parameters ($P < 0.05$) without multicollinearity
(variance inflation factor <20) (Ter Braak, 1986) were obtained. Standard and partial Mantel tests were
run in R (version 3.4.4, Vegan 2.5–3 package) to determine the correlations between environmental
factors and the AOA and AOB distributions. Dissimilarity matrices of communities and environmental
factors were based on Bray-Curtis and Euclidean distances between samples, respectively. Based on
Spearman correlation, the significance of the Mantel statistics was obtained after 999 permutations.
Statistical tests were assumed to be significant at a $P$ value of <0.05.

## 3 Results

### 3.1 Distribution of nutrients, DO, and $N_2O$ along a salinity transect of the Pearl River Estuary

The studied transect was divided into a northern region upstream of the Humen outlet and southern area
(Lingdingyang) (Fig. 1); these regions have distinct biogeochemical characteristics. Salinity exhibited
low values (0.1 to 4.4) upstream of the Humen outlet, and sharply increased from 0.7 to 34.2
downstream in Lingdingyang (Fig. 2a). The ammonium/ammonia concentrations decreased from 167.2

μmol L$^{-1}$ (site P01 surface water) to 20.9 μmol L$^{-1}$ (site P07 bottom water) upstream of the Humen

outlet and consistently decreased downstream in Lingdingyang (5.7 μmol L$^{-1}$ to below detection limit)

(Fig. 2b). Correspondingly, the sum of nitrate and nitrite concentrations increased from 93.6 μmol L$^{-1}$

(site P01 bottom water) to 172.3 μmol L$^{-1}$ (site P03 surface water) upstream, but it sharply decreased

seaward to Lingdingyang (Fig. 2c). The DO concentrations were distinctly lower upstream of the

Humen outlet with nearly one-half of the samples below the hypoxic threshold (63.0 μmol L$^{-1}$; Rabalais

et al., 2010). Generally, the DO concentrations increased seaward from 155.7 to 238.0 μmol L$^{-1}$ in the

surface waters of the Lingdingyang area, whereas they varied from 74.0 to 183.3 μmol L$^{-1}$ in the

bottom waters (Fig. 2d).

In contrast to the DO concentrations, the $N_2O$ concentrations were distinctly higher upstream of the

Humen outlet (48.9–148.2 nmol L$^{-1}$) than in Lingdingyang, where they decreased seaward from 24.6 to

5.4 nmol L$^{-1}$ (Fig. 2e). Similarly, higher $\Delta N_2O_{excess}$ (42.0–141.3 nmol L$^{-1}$) with saturations from

701.1% to 2175.1% was observed upstream; lower $\Delta N_2O_{excess}$ (-1.4–17.8 nmol L$^{-1}$) was present in the

Lingdingyang area with the saturations ranging from 86% to 363% (Fig. 2f). The estimated water–air

$N_2O$ fluxes were 100.4 to 344.0 μmol m$^{-2}$ d$^{-1}$ upstream and decreased in Lingdingyang (42.4 to −2.6

μmol m$^{-2}$ d$^{-1}$) (Fig. 2g). Together, the Pearl River Estuary acts as a $N_2O$ source that releases to the

atmosphere and notably, a significant negative relationship was observed between $\Delta N_2O_{excess}$ or $N_2O$

flux and DO ($P < 0.01$ for each) in the upstream of the Humen outlet (Fig. 2i and j). The isotopic

compositions of $N_2O$ ($\delta^{15}N$-$N_2O$) showed an enrichment of $^{15}N_2O$ seaward, varying from −27.9 to 7.1‰

(Fig. 2h). Overall, upstream of the Humen outlet was characterized by hypoxic waters rich in nitrogen-

based nutrients, where ammonium concentrations decreased and the sum of nitrite and nitrate

concentrations increased seaward, corresponding to distinctly higher $N_2O$ fluxes released to the

atmosphere.

**3.2 Distributions of *amo*A and *nir*S genes along the salinity transect**

The total abundance of AOA *amo*A (sum of free-living and particle-associated communities) varied

from $3.10 \times 10^3$ to $6.87 \times 10^5$ copies L$^{-1}$ in the surface waters (Fig. 3a) and $6.40 \times 10^4$ to $4.21 \times 10^7$ copies

$L^{-1}$ in the bottom waters; an increase along the salinity transect was observed in the bottom (Fig. 3b). In
contrast, the total abundance of AOB *amo*A generally decreased seaward along the salinity transect for
the surface ($4.23 \times 10^2$ to $2.13 \times 10^4$ copies $L^{-1}$) and bottom waters ($4.49 \times 10^3$ to $8.79 \times 10^4$ copies $L^{-1}$) (Fig.
3c and d). Overall, the abundance of AOA *amo*A was significantly higher than AOB ($P < 0.01$). The
total abundance of *nir*S varied from $9.12 \times 10^4$ to $2.00 \times 10^7$ copies $L^{-1}$ and was higher than both AOA ($P$
$< 0.05$) and AOB *amo*A ($P < 0.01$) in the surface waters and AOB *amo*A in the bottom water ($P < 0.01$)
(Fig. 3e and f). Notably, these three genes were predominantly distributed in the particle-associated
communities compared to the free-living communities in the estuary transect (Fig. 3). The transcripts of
the three genes were analyzed in the particle-associated communities of the two incubation sites
upstream of the Humen outlet. The transcript abundances of AOA *amo*A ($7.44 \times 10^3$ to $4.62 \times 10^5$
transcripts $L^{-1}$) were one to three orders of magnitude higher than AOB *amo*A ($3.62 \times 10^2$ to $5.00 \times 10^2$
transcripts $L^{-1}$) at P01 (Fig. 3a–d), whereas the transcript abundances of AOB *amo*A were relatively
higher at P05 (AOB = $8.96 \times 10^4$ to $3.83 \times 10^5$ transcripts $L^{-1}$; AOA = $1.26 \times 10^4$ to $1.39 \times 10^5$ transcripts
$L^{-1}$). The *nir*S gene showed a similar transcript level with AOA *amo*A at P01 ($2.20 \times 10^4$ to $6.69 \times 10^4$
transcripts $L^{-1}$), but one order of magnitude lower transcript level than both AOA and AOB *amo*A at
P05 ($8.59 \times 10^3$ to $1.12 \times 10^4$ transcripts $L^{-1}$) (Fig. 3e and f).

### 3.3 Correlations between genes abundances and biogeochemical parameters

We analyzed the correlations between the genes abundances of AOA, AOB, or denitrifiers and
biogeochemical parameters. The results indicate that AOA *amo*A abundance was significantly
correlated ($P < 0.05{-}0.01$) to the hydrographic parameters temperature (negative) and salinity (positive),
as well as silicate concentration (negative) (Table 1). However, AOB *amo*A abundance was
significantly correlated ($P < 0.05{-}0.01$) to TSM concentration (positive), pH (negative), and DO
(negative). Notably, there were positive correlations between AOB *amo*A abundances and all $N_2O$
parameters as well as ammonia concentrations (Table 1; $P < 0.05{-}0.01$) except for the extremely low-
abundance of free-living AOB. No significant Spearman correlations were found between bacterial
nitrite reductase *nir*S abundance and the measured biogeochemical parameters.
The redundancy analysis was used to further analyze variations in the AOA and AOB distributions
under environmental constraints. The results confirmed that the relatively high AOB abundances in the
upper estuary were constrained by low salinity water, high nitrite and TSM concentrations, low DO
conditions, and high $N_2O$ concentrations whereas high salinity water and opposite environmental
conditions constrained the relatively high AOA abundances in the Lingdingyang area (Fig. 4). These
constraints explained 89.3% of the variation in the ammonia oxidizer distribution along the estuary.
Apparently, the communities with relatively high AOB abundances in the upper estuary positively
influenced the concentration of $N_2O$ in the water.
**3.4 Nitrogen transformation and $N_2O$ production in the incubation experiments**
The in situ biogeochemical conditions of the incubation experiments are shown in Fig. 2 and listed in
Table S1. Site P01 exhibited the lowest in situ DO concentrations (30.0 μmol $L^{-1}$ in the bottom water
and 30.9 μmol $L^{-1}$ in the surface water). The concentration of DIN was generally unchanged in the
early-to-middle (0–18 h) phase for the P01 surface water and early (0–6 h) phase for the P01 bottom
water, but showed a distinct decrease in the end phase (Fig. 5a). The ammonia and nitrite concentrations
consistently decreased and increased, respectively, during the incubation experiments; the nitrate
concentrations decreased in the end phase after a slight increase (Fig. 5b). These results clearly indicate
that nitrification occurred during the entire P01 incubations, and suggest that denitrification may be
present in the end phase (Fig. 5g). The rates of ammonia oxidation during the entire incubations and
nitrite oxidation during the early or early-to-middle phases were estimated by linear regressions of
ammonia and nitrate concentrations, respectively (Fig. 5a and b; Table 2). Correspondingly, the
estimated average $N_2O$ production rate (24 h) was 0.62 nmol $L^{-1}$ $h^{-1}$ in P01 surface water and 0.70 nmol
$L^{-1}$ $h^{-1}$ in P01 bottom water; the estimated $N_2O$ production rates from nitrification were 0.60 nmol $L^{-1}$
$h^{-1}$ in the surface water (18 h) and 1.61 nmol $L^{-1}$ $h^{-1}$ in the bottom water (6 h; Fig. 5c). Thus, the
estimated $N_2O$ yield in the surface and bottom waters based on nitrification was 0.26% and 0.30%
(Table 2).
In the incubation experiments at site P05, the DIN concentrations remained unchanged (Fig. 5d)
and the ammonia concentrations consistently decreased and the nitrite and nitrate concentrations
increased (Fig. 5e). The rates of ammonia and nitrite oxidation were also estimated by linear regressions
of ammonia and nitrate concentrations, respectively (Fig. 5d and e; Table 2). The ammonia oxidation
rates were approximately equal to the sum of the increased nitrite and nitrate concentration rates. Thus,
nitrification occurred during the incubation experiments without denitrification. The estimated $N_2O$
production rates from nitrification were 1.15 nmol $L^{-1}$ $h^{-1}$ in the P05 surface water and 1.41 nmol $L^{-1}$
$h^{-1}$ in the P05 bottom water (Fig. 5f); the estimated $N_2O$ yields based on nitrification were 0.21%
(surface) and 0.32% (bottom) (Table 2).
The $N_2O$ production rates and yields normalized to total AOA and AOB $amo$A gene copies (sum
of particle-associated and free-living fractions or only particle-associated fractions) or transcripts (only
particle-associated fraction) were calculated (Table S3). The highest average $amo$A gene copy-specific
$N_2O$ production rates and yields were in the surface water of site P05, where the highest nitrification
rate was observed (Table 2). The highest average $amo$A gene transcript-specific $N_2O$ production rates
and yields were in the bottom water of site P01, where the highest $N_2O$ production rate was observed
(Table 2).
**4 Discussion**
**4.1 Contribution of nitrification versus denitrification to $N_2O$ production in the hypoxic upper**
**estuary**
The spatial variations of $N_2O$ concentration, its saturation, and water–air $N_2O$ flux along the Pearl River
Estuary are consistent with our previous study (Lin et al., 2016), indicating that higher $N_2O$ in the upper
estuary ensures the Pearl River Estuary acts as a source of atmospheric $N_2O$. The in situ incubation
experiments clearly indicated that nitrification predominantly occurred in the hypoxic waters (e.g. both
the P01 and P05 sites) of the upper estuary along with significant $N_2O$ production, and suggested that
denitrification could be concurrent at the lowest DO site (P01) where the maximum $N_2O$ and $\Delta N_2O_{excess}$
concentrations were observed (Figs. 2 and 5). These results confirm previous speculation that extreme
enrichment of ammonia in the water column due to high loads of anthropogenic-sourced nutrients and
organic matter in an upper estuary (Dai et al., 2008; He et al., 2014) could result in strong nitrification
under low $O_2$ solubility conditions (Dai et al., 2008); thus, $N_2O$ is produced as a byproduct through

nitrification and is oversaturated in the Pearl River Estuary (Lin et al., 2016). The estuary sediments also act as a source of $N_2O$, which is released into the overlying waters through denitrification (Tan et al., 2019); however, in estuarine waters, nitrification apparently is the main source of $N_2O$ production. Previous studies also proposed that nitrification may be the major source of $N_2O$ production in the water column in estuarine systems, such as the Guadalquivir (Huertas et al., 2018), Schelde (De Wilde and De Bie, 2000), and Chesapeake Bay (Laperriere et al., 2019). However, in the estuarine sediments, $N_2O$ production was attributed to both nitrification and denitrification, such as in the Tama (Japan) (Usui et al., 2001) and Yangtze (China) estuaries (Liu et al., 2019; Wang et al., 2019), where denitrification is the major nitrogen removal pathway with $N_2O$ production and consumption.

The isotopic composition of $N_2O$ ($\delta^{15}N$-$N_2O$) was consistent with the above interpretation. According to previous studies (Table S2), the $\delta^{15}N$ of $N_2O$ produced during ammonia oxidation by AOB strains ranged from −68‰ to −6.7‰ (Yoshida, 1988; Sutka et al., 2006; Mandernack et al., 2009; Frame and Casciotti, 2010; Jung et al., 2014; Toyoda et al., 2017) and 6.3−10.2‰ in a marine AOA strain (Santoro et al., 2011). The $\delta^{15}N$ of $N_2O$ produced during denitrification ranged from −37.2‰ to −7.9‰ (Toyoda et al., 2005); during nitrifier-denitrification by AOB strains it ranged from −57.6± 4.1‰ to −21.5‰ (Sutka et al., 2003; Sutka et al., 2006; Frame and Casciotti, 2010). Therefore, the much lower $\delta^{15}N$-$N_2O$ (−27.9‰ to −12.6‰) upstream of the Humen outlet is consistent with AOB nitrification or denitrification processes, whereas enriched $^{15}N$-$N_2O$ (5.2−7.1‰) in the lower reaches approaches AOA nitrification and air $^{15}N$-$N_2O$ (Santoro et al., 2011). Taken together, the isotopic compositions of $N_2O$ (Fig. 2h) and $N_2O$ concentration distribution (Fig. 2e−g) suggest that the high concentrations of $N_2O$ (oversaturation) were produced from strong nitrification by AOB and probably concurrent minor denitrification in the upper estuary, however in the lower reaches, low concentrations of $N_2O$ could be explained by AOA nitrification or water atmospheric exchange of $N_2O$.

**4.2 Correlations of AOB versus AOA with $N_2O$-related biogeochemical parameters along the Pearl River Estuary**

The more abundant AOA *amo*A genes, relative to AOB, and the more abundant genes in the particle-associated communities than free-living communities are consistent with our previous study in the Pearl

River Estuary (Hou et al., 2018), which also reported significant positive correlations between the AOB *amo*A gene abundance and the oxidation rate of ammonia to nitrate. This suggests that AOB might be active in the ammonium and particle-enriched estuary despite their low abundance (Füssel, 2014; Hou et al., 2018). Lower oxygen availability in particle micro-niches has been reported to be favorable for both nitrification and denitrification potential in oxygenated water (Kester et al., 1997). The Spearman correlations and redundancy analysis in this study indicate that high nutrient and TSM concentrations and low DO and pH conditions were favourable for relatively high abundance of AOB in the upper estuary, which is also consistent with our previous Pearl River Estuary study that found high TSM concentrations and low DO and pH influenced substrate availability and thus AOB distribution (Hou et al., 2018). Moreover, AOB *amo*A abundances positively correlated to all $N_2O$-related parameters as revealed by the Spearman correlations and redundancy analysis, suggesting a significant influence of AOB (mainly the particle-associated fraction) on $N_2O$ production/emission in the upper estuary. However, compared to AOB, AOA *amo*A distribution along the estuary transect appears to be regulated more by water mixing since AOA was significantly correlated to the hydrographic parameters and silicate concentration.

To further eliminate the co-varying effects of water mixing, substrate availability, and $N_2O$-related parameters along the salinity transect, and to identify the intrinsic/direct relationship between ammonia oxidizers and $N_2O$ production, we performed standard and partial Mantel tests. We defined four types of environmental constraints: water mixing parameters (temperature, salinity, and silicate), substrate parameters (ammonia/ammonium, nitrite, and nitrate), parameters influencing substrate availability (DO, TSM, and pH), and $N_2O$-related parameters ($N_2O$ and $\Delta N_2O_{excess}$). For the water mixing parameters, we analyzed the relationships between potential temperature ($\theta$), salinity, and silicate concentration with a three-dimensional scatter plot (Fig. S1) that indicates low salinity and high silicate contents were the best indicators for river input in the ocean (Moore, 1986). Thus, we chose temperature, salinity, and silicate as proxies to trace estuarine water masses and mixing. Water mixing parameters (standard and partial Mantel tests, $P < 0.01$) and those influencing substrate availability (standard and partial Mantel tests, $P < 0.05$) significantly controlled variations in the distribution of AOA and AOB along the estuary transect (Fig. 6a and c), supporting the Spearmen and redundancy analyses

conclusions. Notably, variations in the distribution of AOA and AOB were significantly correlated with $N_2O$ production (standard and partial Mantel test, $P < 0.01$) after eliminating the co-varying effects of other parameters (Fig. 6d), demonstrating the significant contribution of ammonia oxidizers to $N_2O$ production.

**4.3 Contribution of AOB versus AOA to $N_2O$ production**

We attempted to accurately assess the relative contributions of AOA and AOB to $N_2O$ production in the Pearl River Estuary by plotting the $N_2O$ production rates (Fig. 7a) and yields (Fig. 7b) normalized to total (sum of AOA and AOB) *amo*A gene copies or transcripts at sites P01 and P05 along the x-y axes that represent the relative contributions of AOA and AOB to the total *amo*A gene or transcript pools. Notably, compared to AOA, higher AOB abundance in the *amo*A gene-based DNA or cDNA pool resulted in distinctly higher (disproportionately higher relative to enhanced abundance) average *amo*A gene copy or transcript-specific $N_2O$ production rates (Fig. 7a) and yields (Fig. 7b), suggesting that AOB may have higher cell-specific activities in the upper estuary and thus be more active in producing $N_2O$ than AOA. Previous studies based on pure cultures of AOB and AOA strains provided evidence that AOB have higher $N_2O$ yields (0.09 to 26%) (Yoshida and Alexander, 1970; Goreau et al., 1980) than AOA (0.002 to 0.09%) during ammonia oxidation (Löscher et al., 2012; Stieglmeier et al., 2014). The higher $N_2O$ yield from AOB has also been observed in soils despite a lower abundance of AOB (Hink et al., 2017; Hink et al., 2018). Based on results indicated by Fig. 7, we conclude that AOB may have higher relative contributions to the high $N_2O$ production in the upper estuary where low DO, high concentrations of $N_2O$ and $\Delta N_2O$, and high $N_2O$ flux were observed.

Ammonia oxidizers are sensitive to oxygen during $N_2O$ production (Santoro et al., 2011; Löscher et al., 2012; Stieglmeier et al., 2014). Studies based on pure cultures of AOB strains *Nitrosomonas marina* NM22 and *Nitrosococcus oceani* NC10, and AOA strain *Nitrosopumilus maritimus* showed higher $N_2O$ yields and production during nitrification by both AOA and AOB when $O_2$ concentrations varied from aerobic to hypoxic conditions (Löscher et al., 2012). However, when $O_2$ concentrations varied from hypoxic to anaerobic conditions (i.e. in a lower $O_2$ concentration range), the AOB strain *Nitrosospira multiformis* and AOA strains *Nitrososphaera viennensis* and *Nitrosopumilus maritimus*

showed that AOB had distinctly higher $N_2O$ yields at lower oxygen conditions and, in contrast, AOA had lower $N_2O$ yields at lower oxygen concentrations (Stieglmeier et al., 2014). In addition, results from the cultured AOB strain *Nitrosomonas marina* C-113a indicated increasing $N_2O$ yields with higher cell concentrations (Frame and Casciotti, 2010). This evidence supports our conclusions that the high concentration of $N_2O$ (oversaturated) may be mainly produced from strong nitrification by the high abundance of AOB in the low DO conditions in the upper estuary.

In addition, it is possible that comammox (COMplete AMMonia OXidiser) species, newly discovered in terrestrial systems (Daims et al., 2015; Santoro, 2016; Kits et al., 2017), are also involved in $N_2O$ production (Hu and He, 2017) given the similar ammonia oxidation pathway to AOB. It has been further reported that the comammox *Nitrospira inopinata* has a lower $N_2O$ yield than AOB due to a lack of NO reductases and the formation of $N_2O$ from the abiotic conversion of hydroxylamine (Kits et al., 2019). However, comammox has not been widely observed in estuarine waters. Also, *nir*K-type denitrifiers may contribute to $N_2O$ production despite being much less abundant than *nir*S-type denitrifiers (Huang et al., 2011; Maeda et al., 2017). Furthermore, *nir*S-type denitrifiers are more likely to be capable of complete denitrification because of a higher co-occurrence of the $N_2O$ reductase gene (*nos*Z) with *nir*S than *nir*K (Graf et al., 2014). However, there is currently no direct evidence that denitrification or nitrifier-denitrification is responsible for $N_2O$ production in the Pearl River Estuary water column. A release of $N_2O$ into the overlying waters through denitrification was reported for the estuary sediments (Tan et al., 2019). Further study is needed to clarify the potential of both *nir*K and *nir*S-type denitrifiers in $N_2O$ production from the interface between sediment and water in the Pearl River Estuary.

## 5 Conclusions

This study explored the relative contributions of AOB and AOA in producing $N_2O$ in the Pearl River Estuary by combining isotopic compositions and concentrations of $N_2O$, distributions and transcript levels of AOB and AOA *amo*A and denitrifier *nir*S genes, and incubation estimates of nitrification and $N_2O$ production rates. Our findings indicate that the high concentrations of $N_2O$ and $\Delta N_2O_{excess}$ and the much lower $\delta^{15}N\text{-}N_2O$ are primarily attributed to strong nitrification by AOB. There is also probably

concurrent minor denitrification in the upper estuary where AOB abundances are higher before decreasing seaward along the salinity transect. Low concentrations of $N_2O$ and $\Delta N_2O_{excess}$ and enriched $^{15}N_2O$ could be explained by AOA nitrification in the lower reaches of the estuary. Collectively, AOB contributed the major part of $N_2O$ production in the upper estuary, which is the major source of $N_2O$ emitted to the atmosphere in the Pearl River Estuary.

## Data availability

All data can be accessed in the form of Excel spreadsheets via the corresponding author.

## The Supplement related to this article is available online.

## Author contribution

M.D. and Y.Z. conceived and designed the experiments. L.M., H.L., and X.X. performed the experiments. L.M., Y.Z., H.L., and X.X. analyzed the data. L.M. and Y.Z. wrote the paper. All authors contributed to the interpretation of results and critical revision.

## Competing interests

The authors declare no conflicts of interest.

## Acknowledgments

We thank Qing Li for measuring ammonia concentrations on board, Jian-Zhong Su and Liguo Guo for measuring dissolved oxygen concentrations on board, and Tao Huang and Lifang Wang for measuring nitrate and nitrite concentrations. We also thank Lei Hou for her assistance in qPCR measurements and data analysis, and Mingming Chen and Huade Zhao for their assistance with the software. This research was supported by the National Key Research and Development Programs (2016YFA0601400) and NSFC projects (41721005, 41676125, and 41706086). We thank Kara Bogus, PhD, from Liwen Bianji, Edanz Editing China (www.liwenbianji.cn/ac), for editing the English text of a draft of this manuscript.

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

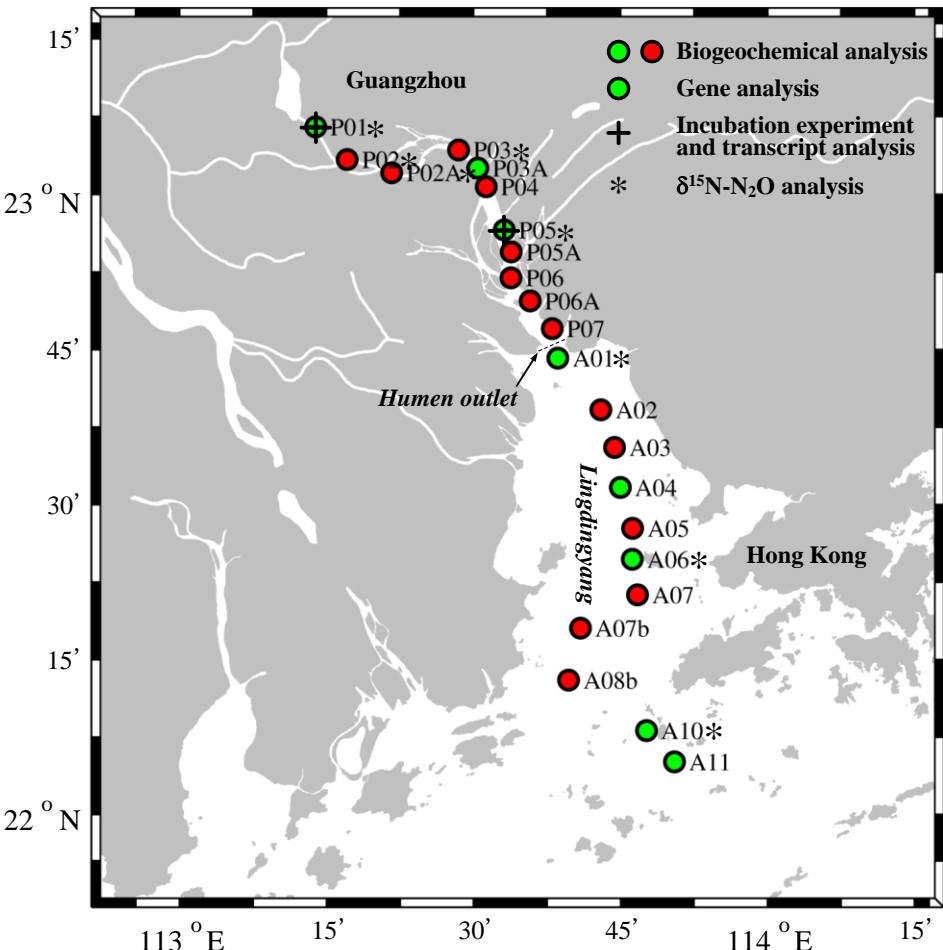

**Figure 1:** Map of the Pearl River Estuary showing the sampling sites. Biogeochemical analyses were
performed on samples from all sites (green and red circles). The green circles indicate sites where genes
were additionally analyzed. The black crosses indicate in situ incubation experiment sites (P01 and
P05). The black asterisks indicate sites where the isotopic composition of $N_2O$ was analyzed.

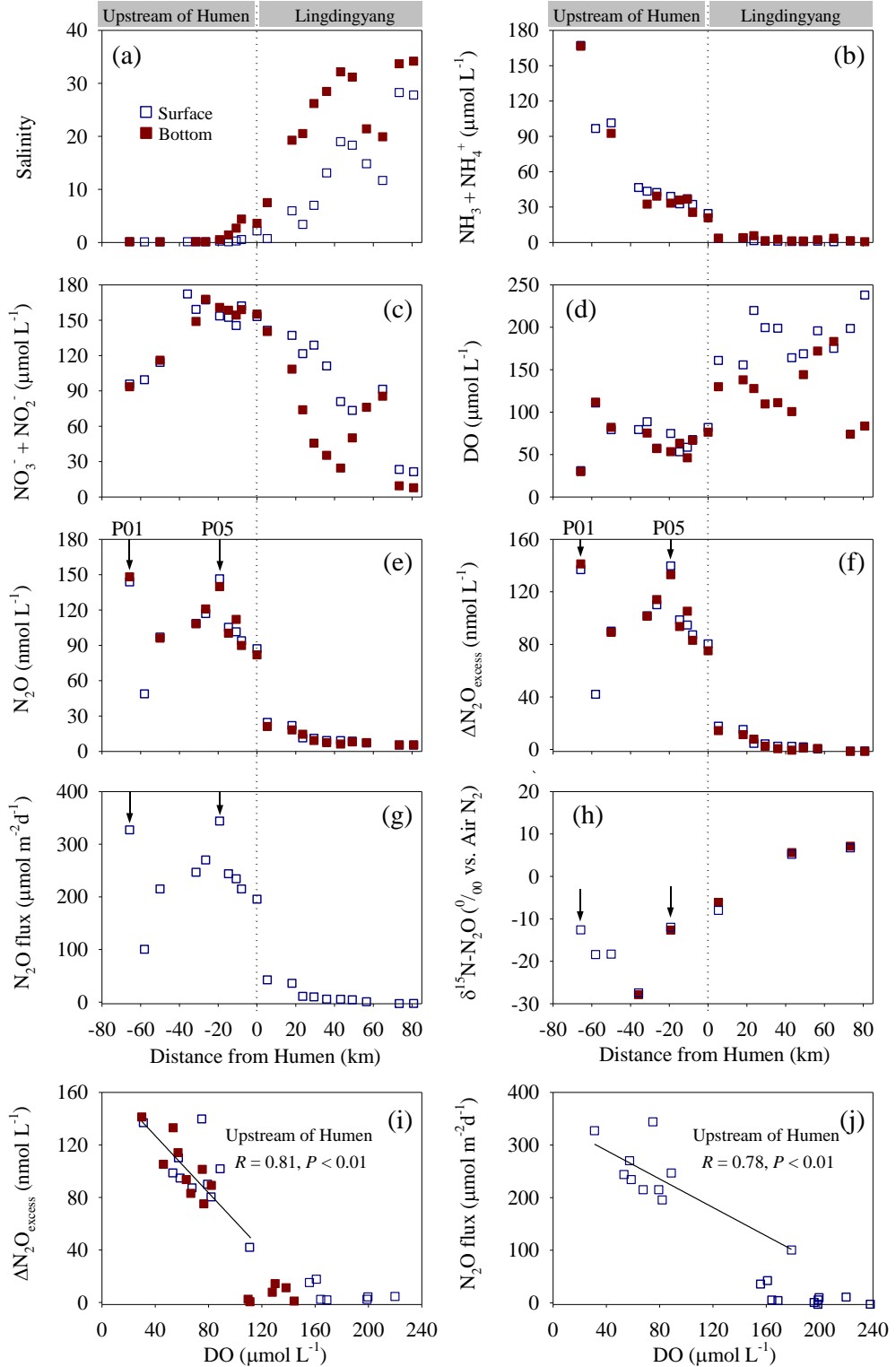

**Figure 2:** Distribution of biogeochemical factors along the Pearl River Estuary transect. (a) Salinity, (b) $NH_3+NH_4^+$, (c) $NO_2^- + NO_3^-$, (d) DO, (e) $N_2O$, and (f) $\Delta N_2O_{excess}$ concentrations, (g) $N_2O$ flux, (h) $\delta^{15}N$-$N_2O$, (i) $\Delta N_2O_{excess}$ vs. DO, and (j) $N_2O$ flux vs. DO. The dashed lines show the division of the transect into the northern (upstream of the Humen outlet) and southern (Lingdingyang) areas. The arrows indicate the sites where in situ incubation experiments were performed.

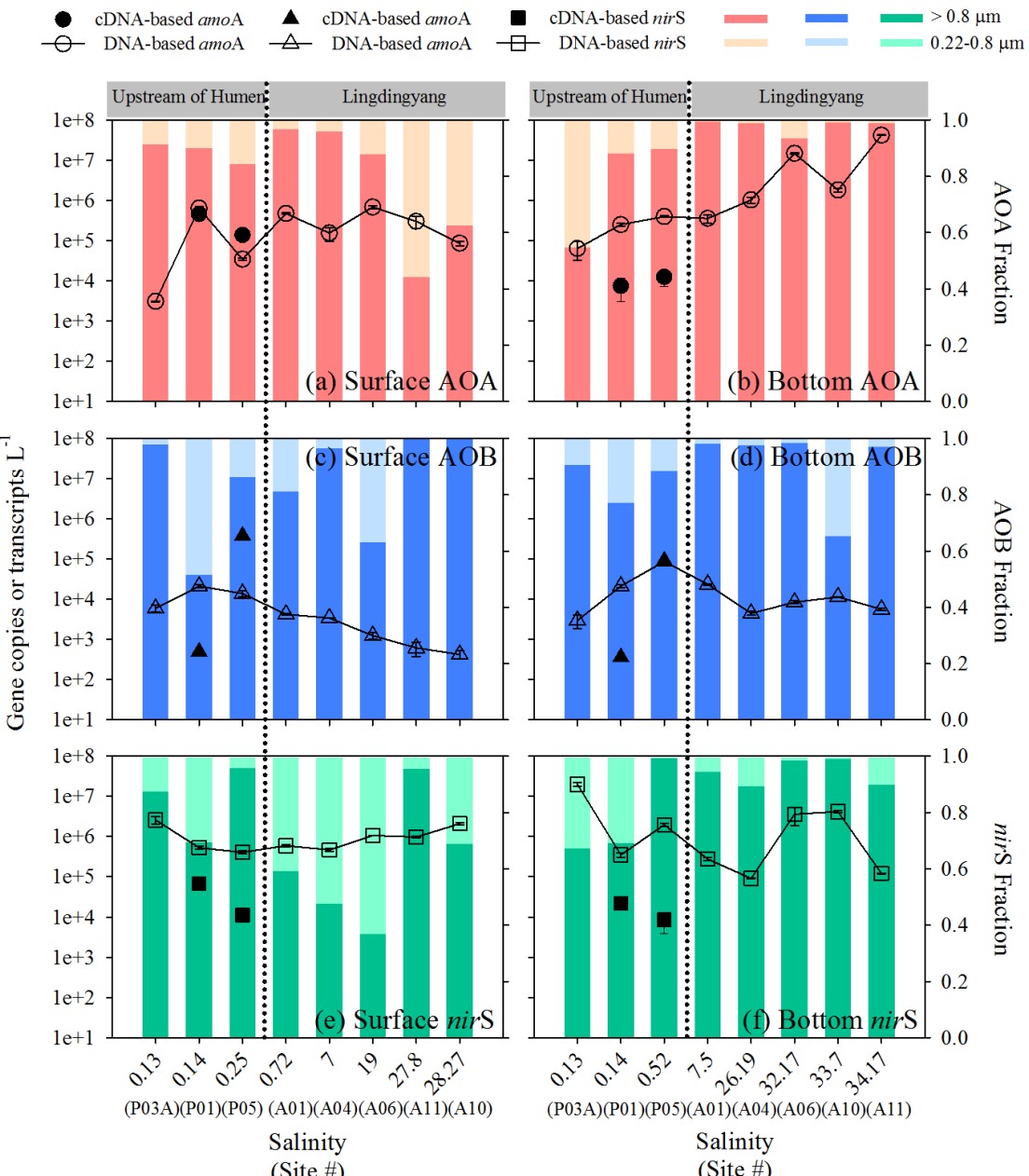

**Figure 3:** Abundance distribution of AOA and AOB *amo*A and bacterial *nir*S along the salinity gradient in the Pearl River Estuary. Abundances of AOA *amo*A genes (open circles) and particle-associated transcripts (closed circles) and the relative abundances of particle-associated and free-living AOA *amo*A genes in (a) surface and (b) bottom waters. Abundances of AOB *amo*A genes (open triangles) and particle-associated transcripts (closed triangles) and the relative abundances of particle-

associated and free-living AOB *amo*A genes in (c) surface and (d) bottom waters. Abundances of bacterial *nir*S genes (open squares) and particle-associated transcripts (closed squares) and the relative abundances of particle-associated and free-living *nir*S genes in (e) surface and (f) bottom waters. The dashed lines indicate the division into the northern (upstream of the Humen outlet) and southern (Lingdingyang) areas.

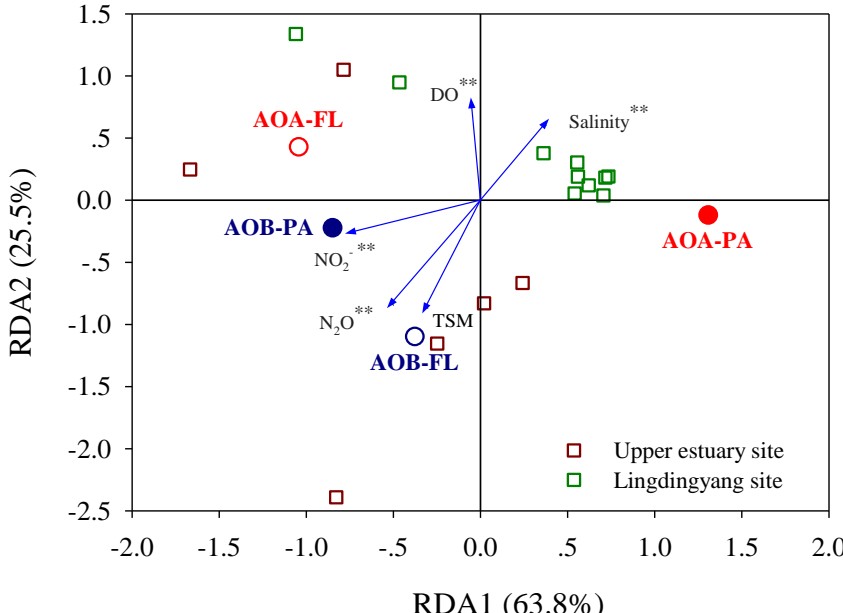

**Figure 4:** The redundancy analysis of the relative abundance of AOA *amo*A and AOB *amo*A under
biogeochemical constraints. PA, particle-associated; FL, free-living. Each square represents an
individual sample. Vectors represent environmental variables. $^*P < 0.05$, $^{**}P < 0.01$ (Monte Carlo
permutation test).

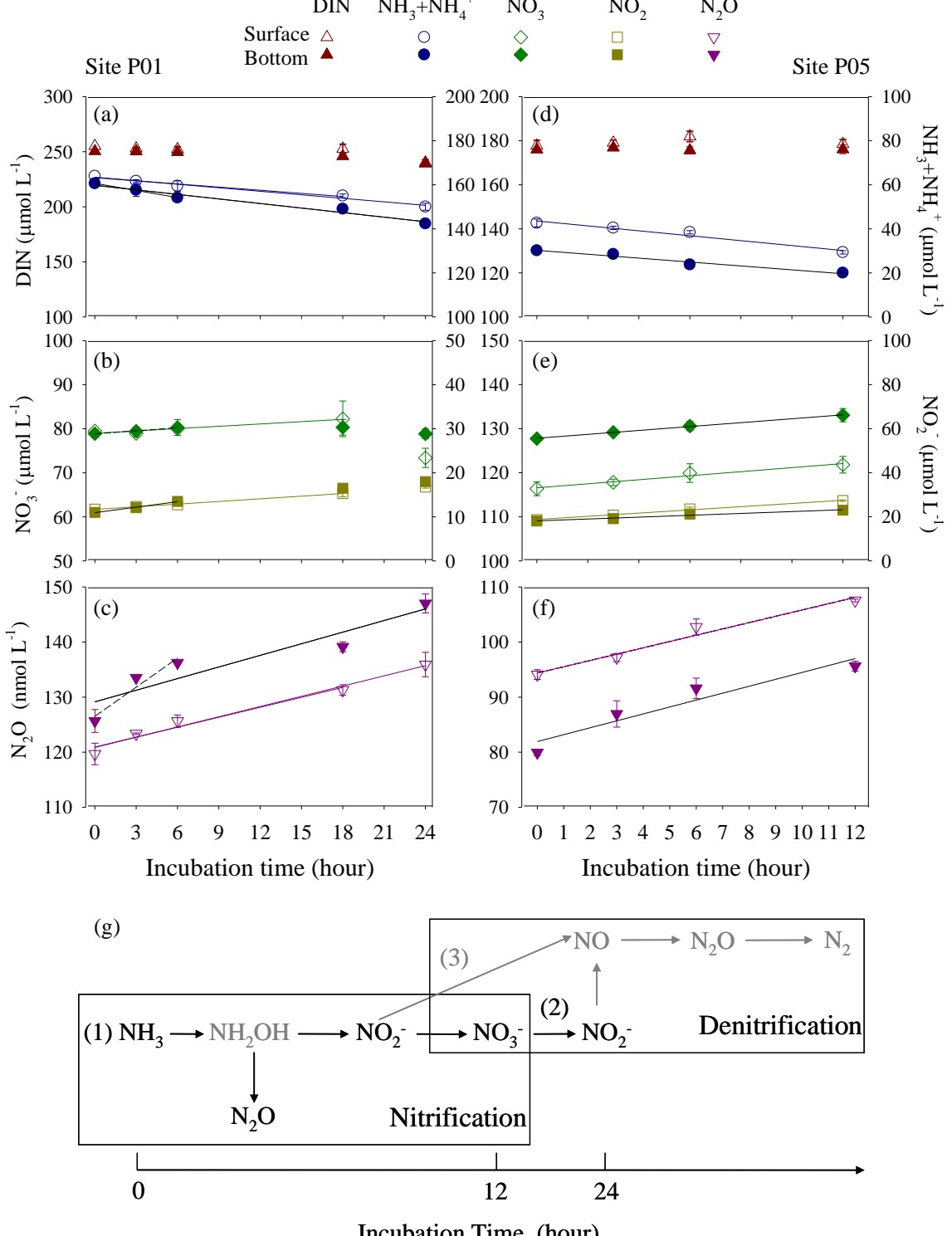

**Figure 5:** Variations in nitrogen compounds and $N_2O$ concentrations at sites P01 and P05 during the incubation experiments in surface (open symbols) and bottom (closed symbols) waters. (a, d) Total DIN (brown triangles) and $NH_3+NH_4^+$ (blue circles), (b, e) $NO_3^-$ (green diamonds) and $NO_2^-$ (dark yellow squares), (c, f) $N_2O$ (purple inverted triangles). Linear regressions depend on whether variations in DIN concentration against time retain "mass balance" in a closed incubation system. The linear regressions of ammonia were used to estimate ammonia oxidation rates in (a) P01 over 18 and 24 h (surface water, blue lines) and 6 and 24 h (bottom water, black lines), and (d) P05 over 12 h (surface, blue line; bottom, black line). The linear regressions of nitrate estimated nitrite oxidation rates in (b) P01 over 18 h (surface water, green line) and 6 h (bottom water, black line), and (e) P05 after 12 h (surface, green line; bottom, black line). The nitrite linear regressions after 18 h (surface water, dark yellow line) and 6 h (bottom water, black line) in P01 and 12 h (surface, dark yellow line; bottom, black line) in P05 are also shown, but do not indicate oxidation rates. The $N_2O$ linear regressions were used to estimate $N_2O$ production rates in (c) P01 after 18 and 24 h (surface water, purple lines) and 6 and 24 h (bottom water, black lines; dashed line, no statistical significance test), and (f) P05 after 12 h (surface, purple line; bottom, black line). All regression equations, $R^2$, and $P$ values are shown in Table 2. (g) A diagram showing transformations of nitrogen compounds and $N_2O$ production during incubation experiments. Nitrification (1) occurred during the entire P01 and P05 incubations and denitrification (2 and/or 3) may be present in the end phase of the P01 incubation. The gray arrows indicate the pathways of nitrogen loss unanalyzed here, and the gray compounds indicate the unmeasured nitrogen compound.

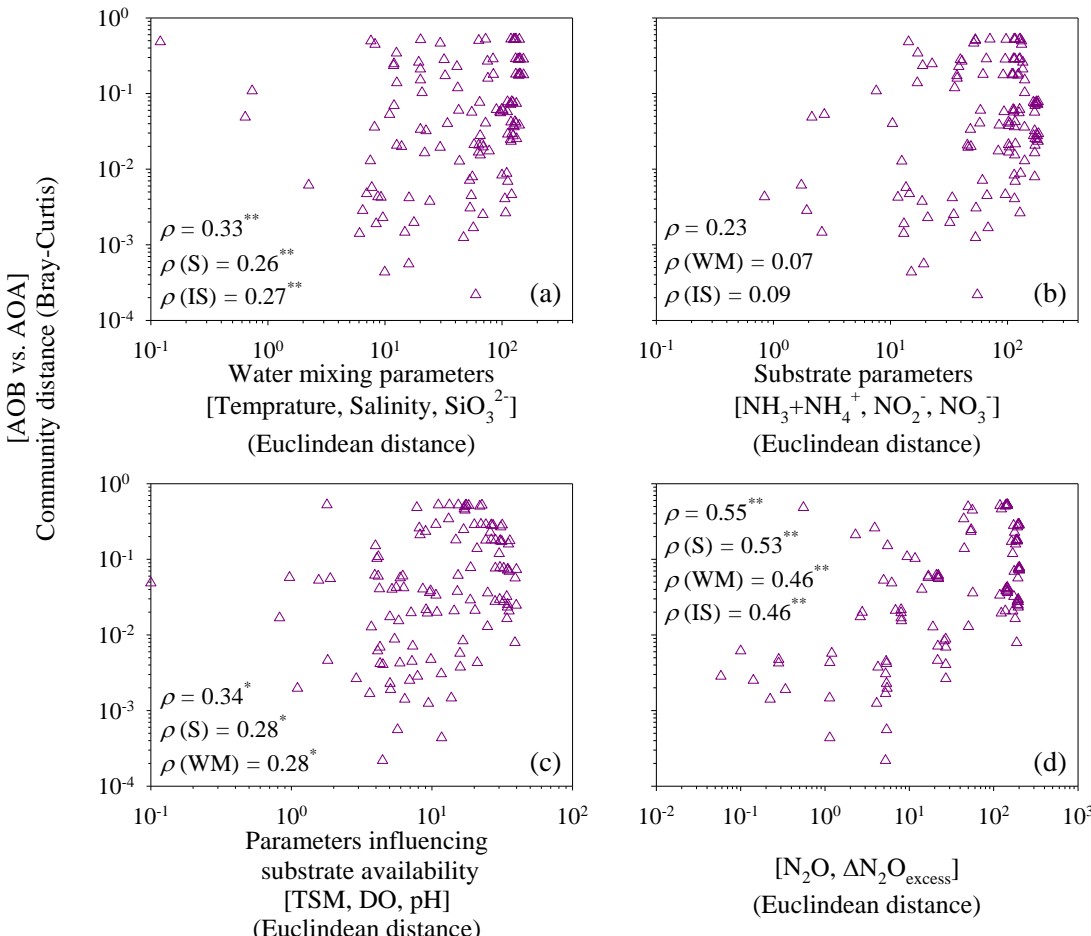

**Figure 6:** Correlations between the relative abundance of AOB versus AOA and (a) water mixing
parameters (temperature, salinity, and silicate), (b) substrate parameters (ammonia/ammonium, nitrite,
and nitrate), (c) parameters influencing substrate availability (TSM, DO, and pH), or (d) $N_2O$
parameters ($N_2O$ and $\Delta N_2O$). The ammonia oxidizers matrix was calculated according to the relative
AOA and AOB abundances. Dissimilarity matrices of the relative abundance of AOB *amo*A and AOA
*amo*A were based on Bray-Curtis distances and environmental factors were based on Euclidean
distances between samples. Standard and partial Mantel tests were run to measure the correlation
between two matrices. Spearman correlation coefficient ($\rho$) values are shown for standard (first value)

1 and partial Mantel (second, third, and fourth) tests. The *P* values were calculated using the distribution

2 of the Mantel test statistics estimated from 999 permutations. $^*P < 0.05$; $^{**}P < 0.01$.

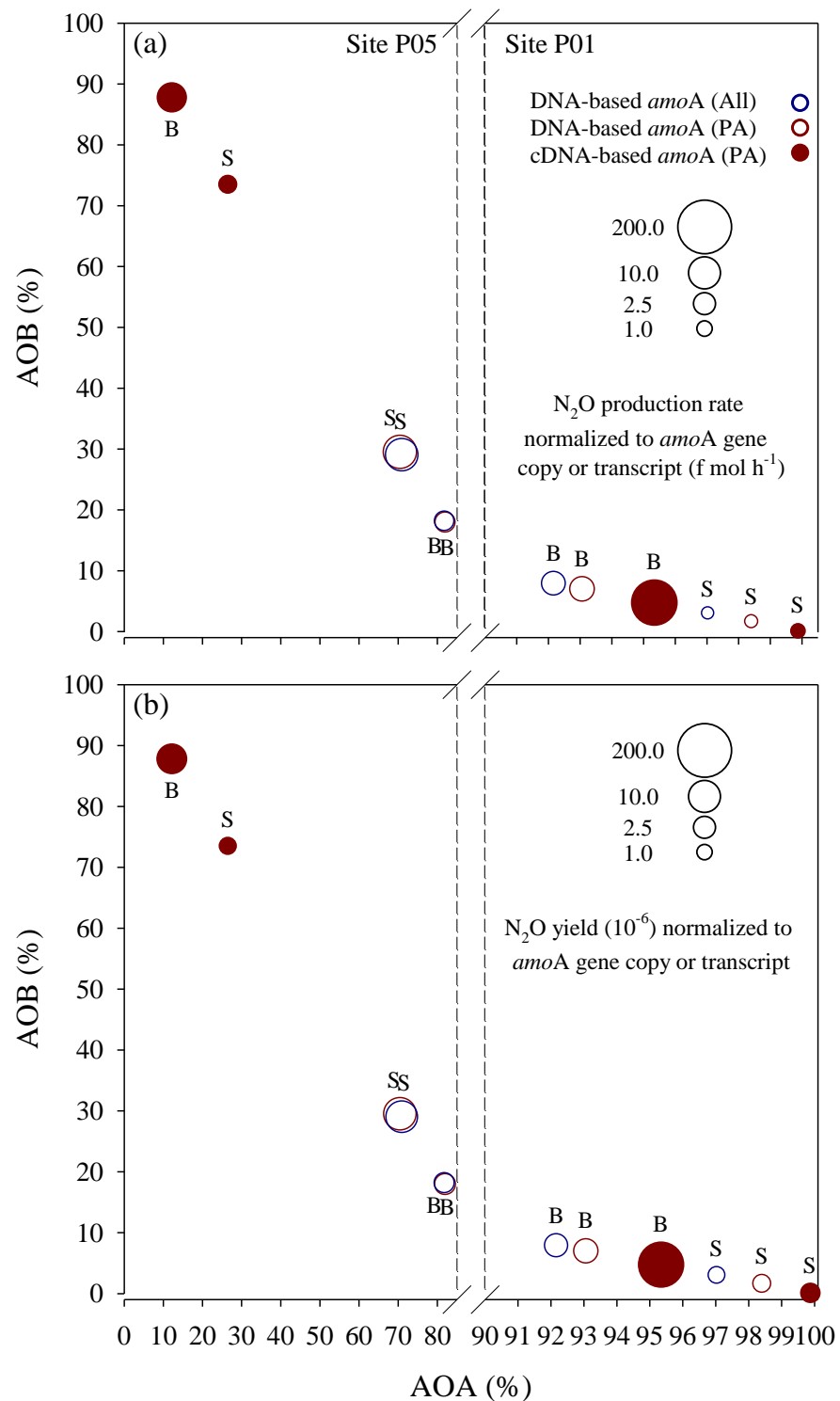

**Figure 7:** N$_2$O (a) production rates and (b) yields normalized to total *amo*A gene copy or transcript
numbers of AOA and AOB in a given sample. They are presented along the x-y axes that represent the
relative contributions of AOA and AOB to the total *amo*A gene or transcript pools. S, surface; B,
bottom. All, sum of free-living and particle-associated communities; PA, particle-associated
communities.

1 **Table 1** Rho (ρ) values for the relationships between nitrifier and denitrifier gene abundances and biogeochemical parameters in the Pearl River

2 Estuary.

| Biogeochemical parameters | PA + FL | | | PA (> 0.8 μm) | | | FL (0.22–0.8 μm) | | |
|---|---|---|---|---|---|---|---|---|---|
| | AOA-*amo*A (n =16) | AOB-*amo*A (n =16) | *nir*S (n =16) | AOA-*amo*A (n =16) | AOB-*amo*A (n =14) | *nir*S (n =16) | AOA-*amo*A (n =16) | AOB-*amo*A (n =16) | *nir*S (n =16) |
| Temperature | −0.694[*] | 0.359 | 0.085 | −0.676[*] | 0.303 | 0.165 | −0.438 | 0.358 | 0.229 |
| Salinity | 0.644[*] | −0.339 | −0.018 | 0.604[*] | −0.270 | −0.047 | 0.403 | −0.351 | −0.356 |
| $SiO_3^-$ | −0.541[*] | 0.559[*] | 0.206 | −0.497 | 0.503[*] | 0.282 | −0.350 | 0.481 | 0.238 |
| TSM | −0.109 | 0.668[*] | 0.047 | −0.097 | 0.612[*] | 0.194 | 0.191 | 0.565[*] | −0.071 |
| pH | 0.381 | −0.656[*] | 0.157 | 0.316 | −0.615[*] | 0.088 | 0.377 | −0.605[*] | −0.059 |
| DO | −0.074 | −0.771[**] | −0.026 | −0.121 | −0.729[**] | −0.144 | 0.009 | −0.697[*] | 0.218 |
| $NH_3/NH_4^+$ | −0.482 | 0.646[*] | 0.068 | −0.482 | 0.571[*] | 0.196 | −0.325 | 0.587[*] | 0.000 |
| $NO_3^-$ | −0.485 | 0.359 | −0.138 | −0.444 | 0.353 | −0.112 | −0.588[*] | 0.213 | 0.115 |
| $NO_2^-$ | −0.588[*] | 0.447 | 0.126 | −0.556[*] | 0.356 | 0.212 | −0.421 | 0.288 | 0.265 |
| $N_2O$ | −0.421 | 0.641[*] | −0.194 | −0.356 | 0.606[*] | −0.121 | −0.385 | 0.490 | 0.047 |
| $\Delta N_2O_{excess}$ | −0.527[*] | 0.559[*] | −0.160 | −0.480 | 0.517[*] | −0.081 | −0.369 | 0.504 | 0.096 |
| $N_2O$ flux[a] | −0.190 (n = 8) | 1.000[**] (n = 8) | −0.524 (n = 8) | −0.143 (n = 8) | 1.000[**] (n = 8) | −0.310 (n = 8) | −0.571 (n = 8) | 0.657 (n = 6) | −0.524 (n = 8) |

1 [a]Surface data; [*]False discovery rate-adjusted $P < 0.05$; [**] False discovery rate-

2 adjusted $P < 0.01$.

3 PA, particle-associated communities; FL, free-living communities.

**Table 2** Linear regressions of ammonia, nitrite, nitrate, and $N_2O$ concentrations against time and $N_2O$ yields during incubation experiments.

| Site_Layer | Time (hour) | $\Delta(NH_3+NH_4^+)$ ($\mu$mol $L^{-1}$ $h^{-1}$) | | | $\Delta NO_2^-$ ($\mu$mol $L^{-1}$ $h^{-1}$) | | | $\Delta NO_3^-$ ($\mu$mol $L^{-1}$ $h^{-1}$) | | | $\Delta N_2O$ (nmol $L^{-1}$ $h^{-1}$) | | | $N_2O$ yield (%) |
|---|---|---|---|---|---|---|---|---|---|---|---|---|---|---|
| | | Equation | $R^2$ | Rate[a] | Equation | $R^2$ | Rate[a] | Equation | $R^2$ | Rate[a] | Equation | $R^2$ | Rate[a] | |
| P01_S | 18 | $y=-0.47x+163.20$ | 0.96[*] | 0.47 | $y=0.20x+11.69$ | 1.00[**] | 0.20 | $y=0.18x+78.98$ | 0.90[*] | 0.18 | $y=0.60x+120.93$ | 0.96[*] | 0.60[b] | 0.26[b] |
| | 24 | $y=-0.53x+163.44$ | 0.98[**] | 0.53 | – | – | – | – | – | – | $y=0.62x+120.85$ | 0.98[**] | 0.62 | –[c] |
| P01_B | 6 | $y=-1.08x+160.65$ | 1.00[*] | 1.08 | $y=0.42x+10.95$ | 1.00[*] | 0.42 | $y=0.23x+78.84$ | 0.98 | 0.23 | $y=1.61x+127.04$ | 0.98 | 1.61[b] | 0.30[b] |
| | 24 | $y=-0.69x+159.76$ | 0.96[**] | 0.69 | – | – | – | – | – | – | $y=0.70x+129.14$ | 0.86[*] | 0.70 | –[c] |
| P05_S | 12 | $y=-1.12x+43.58$ | 0.96[*] | 1.12 | $y=0.73x+18.78$ | 1.00[**] | 0.73 | $y=0.46x+116.58$ | 0.98[**] | 0.46 | $y=1.15x+79.79$ | 0.98[**] | 1.15[b] | 0.21[b] |
| P05_B | 12 | $y=-0.89x+30.25$ | 0.96[*] | 0.89 | $y=0.42x+18.17$ | 0.96[*] | 0.42 | $y=0.44x+127.83$ | 1.00[**] | 0.44 | $y=1.41x+81.57$ | 0.96[*] | 1.41[b] | 0.32[b] |

[a]These rates are net rates since $\Delta(NH_3+NH_4^+)$ is the net consumption and $\Delta NO_2^-$, $\Delta NO_3^-$, and $\Delta N_2O$ is the net production during incubation.
[b]These rates and yields (when only nitrification occurred) were used to calculate the average *amo*A gene copy-specific $N_2O$ production rates and
$N_2O$ yields in Figure 7.
[c]No estimation of $N_2O$ yield was made due to nitrification and denitrification may occur concurrently and DIN was not in balance.
[*]$P < 0.05$; [**]$P < 0.01$.
–No regression analysis or no estimation made due to DIN was not in balance.