# Peer review of "Major role of ammonia-oxidizing bacteria in N2O production in the Pearl River Estuary"

_Biogeosciences, 2019_

## Referee Comment (RC1) · Anonymous Referee #1 · 5 Jun 2019

Ma et al. investigated the relationship between N2O production and spatial distribution of AOA and AOB along a salinity gradient in the Pearl River Estuary, China by using qPCR, chemical analysis and in situ incubation experiment. Data are well analyzed and presented. However, the manuscript's structure should be modified because the some results were presented in the discussion section, and some conclusions needs to be rephrased because the main findings in this study were mainly based on the correlation analysis OR statistical analysis (e.g., between N2O production and the abundance of functional genes), which can't provide a solid support for a causal relationship between microbial contributors and N2O production. More specific comments and suggestions are given below:

1. As mentioned by authors, both nirK and nirS genes are the key functional genes in

the denitrification pathway, so why did not determine the abundance of nirK gene here?

2. Page 7, line 18-19, make subscript for some chemistry formulas (N2O, NH3 etc.);

3. Page 7, line 24, please correct the P value using the Bonferroni correction or other multiple-comparison methods;

4. Page 7, line 25, and Fig. 5. Please check the multicollinearity problems before perform the RDA analysis. Some environmental parameters are highly correlated with each other, some of them should be removed from the RDA analysis;

5. Page 8, line 5-8 and Fig. 6. I am not convinced with the usage of Mantel and partial Mantel tests here due to two following reasons: 1) for ammonia oxidizer community, actually there were only four variables based on qPCR analysis (PA AOA, FL AOA, PA AOB and FL AOB) but not community data based on sequencing, so I don't think the results of qPCR reflected the truly community composition of ammonia oxidizers; and 2) the authors divided the environmental into four groups, but the classification seems a bit confusing. For example, why classify silicate into water mass but not substrate parameters? And TSM, DO and pH were classify as water mass parameters by numerous previous studies;

6. Page 8, line 20, is the 63.0  $\mu \text{mol/L}$  the hypoxic threshold?

7. Page 9, line 11, please re-phrase this subtitle because only the transcripts of amoA and nirS genes from two freshwater stations were quantified here;

8. Page 12, line 12-13, too much speculation;

9. Page 12, line 15-26, please move this part into Results section, and again, I don't think the classification for environmental parameters is on the right way;

10. Page 12, line 23, "positive correlations between AOB amoA abundances and all N2O parameters", should be except for FL AOB;

11. Page 12, line 27, the results of RDA analysis also should be presented in Results
section;

12. The most part of first paragraph of 4.3 subsection should be moved into Results section;

13. How about the potential role of comammox and nirK-type denitrifier for N2O production in PRE, please discuss it in the 4.3 subsection.

14. Fig. 7. It is a little difficult to understand this figure. It seems like the AOA contributed more for N2O production and yield in site P01, right?

15. Table 2, Spearman rank correlation analysis generate a rho () value rather than a R value.

---

## Referee Comment (RC2) · Anonymous Referee #2 · 12 Jun 2019

Major comments:

More caution is needed on these concentration-based "rate" measurements. Without isotope tracers, very little can be said about actual rates. Evidence for this is in the N2O yields. The yields reported here are about 100X lower than ever reported from cultures or the field (see Ji et al. 2018 GBC).

The strength of the correlation between genes and rates absolutely cannot be used to apportion a relative importance of one group of ammonia oxidizers or the other to the total rates. Nothing can be concluded from the data presented about who the important nitrifiers are. One possibility would be to obtain to a range of cell-specific ammonia oxidation rates from the literature and then use those in combination with the qPCR data to calculation the relative contribution of each group to the observed

"rates."

The literature review, both in the Introduction and Discussion, is severely lacking. There is a substantial literature about nitrification and N2O production in estuaries, almost none of which are referenced here. Normally I would provide some specific suggestions, but the omissions are too vast to list. One place to start would be a review by Damashek and Francis 2018 Estuaries and Coasts, or a nice earlier paper with a summary of nitrification rates in estuaries, Damashek et al. 2016 Estuaries and Coasts.

The nirS data are not very useful to this manuscript in that there is essentially no relationship between nirS abundance and N2O production from denitrification. nirS presence could just as easily be a marker for N2O consumption.

All the physical dynamics in the system have been reduced to a very naive "water mass" identification. Basic concepts in estuarine biogeochemistry are absent–for example, using salinity as a conservative tracer in a two-end member mixing model to determine production and loss of the various biogeochemical parameters.

Specific comments:

p. 3 lines 15-16 Unclear to me what is meant by "runoff ranked 17th".

p. 5 lines 2-4 What N2O standards were used? How was the GC calibrated?

p. 5 line 6 How was N2Oaquatic calculated?

p.7 lines 3 How much did DO concentration change over the course of the 24 h incubations? What effect would this have on the measured N2O production?

p. 7 lines 18-19 Were both N2O yield equations used? Compared? Were they equal?

More details are needed about how you arrived at the Schmidt number for N2O. Is this the Raymond and Cole reference?

Need additional details of the calibration of the isotopic values.

p.7 Why is N2O yield in units of permil? (line 18-19, and also in the Discussion). Also would be more conventional to list this as N2O-N not N-N2O

No discussion of particle attached versus free living amoA copies. Data is presented in multiple figures. Previous literature show no association. Did the filters clog?

P. 9 lines 4-6 "the entire PRE acts as a N2O source" but negative air-sea fluxes are reported in the previous sentence?

p.11 lines 19-26: This paragraph confuses some important concepts. Some of these numbers are the isotopic composition of N2O produced by ammonia oxidizers, but some of these numbers are the isotope effect (epsilon). Also, the isotopic composition of the N2O being produced by nitrification is dependent on the isotopic composition of the NH3 being oxidized, for which no measurements or even estimates are provided.

p.12 lines 15-17: Doesn't make sense to refer to 'water masses' in estuaries. There is a tremendous amount of mixing that leads to variation in these parameters. Just because something is a different salinity doesn't mean it's a different 'water mass.' These parameters are just 'hydrography.'

p. 12 lines 15-28 and p. 13 lines 1 – 18 A lot of results presented that should be moved to the results section.

p. 12 line 27 "ammonia oxidizer community" The use of the word "community" throughout the paper is confusing. More accurate to state the abundances of AOA and AOB?

p. 24 Fig 1 i,j It looks like two different slopes in the data upstream and Lingdingyang. This could be quantified using a break point analysis.

p. 32 I found this figure confusing. Perhaps it would be useful to have a table with the data presented in the figure? It is unclear using AOB and AOA% if the normalized N2O production values are a result of the N2O yield or low/high amoA abundance.

Technical corrections p. 2 lines 18-22 Needs citation

"Denitrification by heterotrophic denitrifiers is another major pathway of N2O production in marine environments. NO2- is reduced by a copper-containing (NirK) or cytochrome cd1-containing nitrite reductase (NirS) to nitric oxide (NO), and then by a heme-copper NO reductase (NOR) to N2O."

p.3 lines 3 citation should be after "soil"

"and arable (Clark et al., 2012; Jones et al., 2014) soils"

p. 3 lines 10-11 Needs citation

"Moreover, there is a potential niche overlap between nitrifiers and denitrifiers in low oxygen conditions."

p. 4 lines 15-16 Should be moved to results section 2.2 discussing ammonia analysis

"Ammonia/ammonium concentrations were analyzed onboard."

p. 4 line 25-26 What salinity, temperature and DO probes were used?

p. 5 lines 5-23 Not all variables in the equations are defined.

p. 31 Fig 6 Should axes be swapped?

---

## Author Comment (AC1) · 22 Jul 2019

**Manuscript Number: bg-2019-132**

**Manuscript title: Major role of ammonia-oxidizing bacteria in N$_2$O production in the Pearl River Estuary**

**Response to Reviewer #1**

We greatly thank the reviewer for the valuable comments, useful suggestions and careful revisions, based on which we have revised the manuscript. And the point-by-point responses to the comments are shown below.

**Anonymous Referee #1**

Ma et al. investigated the relationship between N$_2$O production and spatial distribution of AOA and AOB along a salinity gradient in the Pearl River Estuary, China by using qPCR, chemical analysis and in situ incubation experiment. Data are well analyzed and presented. However, the manuscript's structure should be modified because the some results were presented in the discussion section, and some conclusions needs to be rephrased because the main findings in this study were mainly based on the correlation analysis OR statistical analysis (e.g., between N$_2$O production and the abundance of functional genes), which can't provide a solid support for a causal relationship between microbial contributors and N$_2$O production.
**Response**:

Many thanks for the reviewer's comments. We moved the results pointed out by the reviewer into the Discussion section. We also revised some conclusion sentences with the appropriate tone according to the reviewer's suggestions.

More specific comments and suggestions are given below:

1. As mentioned by authors, both nirK and nirS genes are the key functional genes in the denitrification pathway, so why did not determine the abundance of nirK gene here?
**Response**:

The *nir*S and *nir*K genes encode cytochrome cd1 and copper-containing nitrite

reductase, respectively. They were functionally and physiologically equivalent, but structurally different and could not be detected in the same strains in the previous research (Coyne et al., 1989), while the recent genomic analyses found a few bacteria contain both *nir*S and *nir*K (Graf et al., 2014). A recent genomic analysis revealed that a great many *nir*K-encoding bacteria have both denitrification and DNRA (Dissimilatory Nitrate Reduction to Ammonium) pathways (Helen et al., 2016). Furthermore, it was reported that *nir*S genes were more widely distributed than the *nir*K genes (Zumft, 1997; Bothe et al., 2000), and *nir*S genes were both more abundant and more diverse than *nir*K in the estuarine water columns (Zhu et al., 2018; Wang et al., 2019) and various estuarine sediments (Nogales et al, 2002; Santoro et al, 2006; Abell et al., 2010; Mosier and Francis, 2010; Beman, 2014; Smith et al., 2015; Lee and Francis, 2017). The previous study on the Pearl River sediment also showed that *nir*K abundance was much lower than *nir*S abundance (Huang et al., 2011). Therefore, *nir*S can be used to identify the distribution of denitrifiers in the PRE, suggesting the denitrification potential and indirectly indicating $N_2O$ production.

**References**

Abell, G. C. J., Revill, A. T., Smith, C., Bissett, A. P., Volkman, J. K., and Robert, S. S.: Archaeal ammonia oxidizers and *nirS*-type denitrifiers dominate sediment nitrifying and denitrifying populations in a subtropical macrotidal estuary, ISME J., 4, 286–300, 2010.

Beman, J. M.: Activity, abundance, and diversity of nitrifying archaea and denitrifying bacteria in sediments of a subtropical estuary: Bahía del Tóbari, Mexico. Estuar. Coast, 37, 1343–1352, 2014.

Bothe, H., Jost, G., Schloter, M., Ward, B. B., and Witzel, K. P.: Molecular analysis of ammonia oxidation and denitrification in natural environments, FEMS Microbiol. Rev., 24, 673–690, 2000.

Coyne, M. S., Arunakumari, A., Averill, B. A., and Tiedje, J. M.: Immunological identification and distribution of dissimilatory heme cd1 and non-heme copper nitrite reductases in denitrifying bacteria, Appl. Environ. Microbiol., 55, 2924–2931, 1989.

Graf, D. R. H., Jones, C. M., and Hallin,. S.: Intergenomic comparisons highlight modularity of the denitrification pathway and underpin the importance of

community structure for N$_2$O emissions. PloS One 9: e114118. https://doi.org/10.1371/journal.pone.0114118. s008. 2014.

Helen, D., Kim, H., Tytgat, B., and Anne, W.: Highly diverse nirK genes comprise two major clades that harbour ammoniumproducing denitrifiers, BMC Genomics 17, 155, 2016. https://doi.org/10.1186/s12864-016-2465-0

Huang, S., Chen, C., Yang, X., Wu, Q., and Zhang, R.: Distribution of typical denitrifying functional genes and diversity of the nirS-encoding bacterial community related to environmental characteristics of river sediments, Biogeosciences, 8, 3041–3051, 2011.

Lee, J. A., and Francis, C.A.: Spatiotemporal characterization of San Francisco Bay denitrifying communities: A comparison of *nir*K and *nir*S diversity and abundance. Microbial Ecology 73, 271–284, 2017.

Mosier, A. C., and Francis, C. A.: Denitrifier abundance and activity across the San Francisco Bay Estuary, Env. Microbiol. Rep., 2, 667–676, 2010.

Nogales, B., Timmis, K. N., Nedwell, D. B., and Osborn, A. M.: Detection and diversity of expressed denitrification genes in estuarine sediments after reverse transcription-PCR amplification from mRNA, Appl. Environ. Microbiol., 68, 5017–5025, 2002.

Santoro, A. E., Boehm, A. B., and Francis, C.A.: Denitrifier community composition along a nitrate and salinity gradient in a coastal aquifer. Appl. Environ. Microbiol., 72, 2102–2109, 2006.

Smith, J. M., Mosier, A. C., and Francis, C. A.: Spatiotemporal relationships between the abundance, distribution, and potential activities of ammonia-oxidizing and denitrifyingmicroorganisms in intertidal sediments. Microb. Ecol., 69, 13–24, 2015.

Wang, J., Kan, J., Qian, G., Chen, J., Xia, Z., Zhang, X., Liu, H., and Sun, J.: Denitrification and anammox: Understanding nitrogen loss from Yangtze Estuary to the east China sea (ECS), Environ. Pollut., 2019. https://doi.org/10.1016/j.envpol.2019.06.025.

Zhu, W., Wang, C., Hill, J., He, Y., Tao, B., Mao, Z., and Wu, W.: A missing link in the estuarine nitrogen cycle?: Coupled nitrification-denitrification mediated by suspended particulate matter, Sci. Rep., 8, 2282, 2018. DOI:10.1038/s41598-018-20688-4

Zumft, W. G.: Cell biology and molecular basis of denitrification, Microbiol. Mol. Biol. R., 61, 533–616, 1997.

2. Page 7, line 18-19, make subscript for some chemistry formulas ($N_2O$, $NH_3$ etc.);
**Response**:
Revised as suggested.

3. Page 7, line 24, please correct the P value using the Bonferroni correction or other multiple-comparison methods;
**Response**:
Thanks for the reviewer's suggestion. As normal distribution of the individual data sets was not always met, we could not use Bonferroni correction to correct $P$ value. So, we used the False Discovery Rate (FDR)-based procedures to identify truly significant comparisons, which has been considered as the best choice available in many studies of ecology and evolution (Pike, 2011). In the revised manuscript, we corrected the $P$ value using the False Discovery Rate (FDR)-based multiple comparison procedures, and added the statement in the "2.5 Statistical analyses": "*False Discovery Rate (FDR)-based multiple comparison procedures were adopted to evaluate the significance of multiple hypotheses and identify truly significant results (FDR-adjusted P value) (Pike, 2011).*" The FDR-adjusted $P$ values have been added in the Table 2.

**Reference**
Pike, N.: Using false discovery rates for multiple comparisons in ecology and evolution, Methods in Ecology and Evolution, 2, 278–282, 2011.

4. Page 7, line 25, and Fig. 5. Please check the multicollinearity problems before perform the RDA analysis. Some environmental parameters are highly correlated with each other, some of them should be removed from the RDA analysis;
**Response**:
Thanks for the reviewer's suggestion. We used the value of Variance Inflation Factor (VIF) to check the multicollinearity. The variables Temperature, Silicate, $NH_3$, $NO_3^-$, and pH had a high VIF (> 20) (Ter Braak et al., 1998; Ricart et al., 2010), so we removed these factors from the RDA analysis. We modified this part in the 3.3

subsection of the revised manuscript. —"*The RDA was used to further analyze variations in the relative abundances of ammonia oxidizers under the environmental constraints. The results confirmed that the relatively high AOB abundances in the upper estuary were constrained by low salinity water, high nitrite and TSM concentrations and low DO conditions, as well as high N$_2$O concentration; whereas the water with high salinity and opposite environmental conditions constrained the high AOA abundances in the Lingdingyang area (Fig. 5). These constraints explained 89.3% of the variation in the ammonia oxidizers distribution along the PRE.*" Please see the revised Figure 5 (below).

**References**

Ricart, M., Guasch, H., Barceló, D., Brix, R., Conceicao, M. H., Geiszinger, A., de Alda, M. J. L., López-Doval, J. C., Munoz, I., Postigo, C., Romaní, A. M., Villagrasa, M., Sabater, S.: Primary and complex stressors in polluted mediterranean rivers: Pesticide effects on biological communities, J. Hydrol., 383, 52–61, 2010.

Ter Braak, C .J. F., Smilauer, P.: CANOCO Reference Manual and User's Guide to Canoco for Windows: Software for Canonical Community Ordination (Version 4). Microcomputer Power, Ithaca, New York, 352, 1998.

[Figure]

Figure R1 (Figure 5 in the MS): Redundancy analysis of the relative abundance of AOB-*amo*A and AOA-*amo*A under biogeochemical constraints. Each square represents an individual sample. Vectors represent environmental variables. $^{*}P <0.05$, $^{**}P <0.01$ (Monte Carlo permutation test).

5. Page 8, line 5-8 and Fig. 6. I am not convinced with the usage of Mantel and partial Mantel tests here due to two following reasons: 1) for ammonia oxidizer community, actually there were only four variables based on qPCR analysis (PA AOA, FL AOA, PA AOB and FL AOB) but not community data based on sequencing, so I don't think the results of qPCR reflected the truly community composition of ammonia oxidizers; and 2) the authors divided the environmental into four groups, but the classification seems a bit confusing. For example, why classify silicate into water mass but not substrate parameters? And TSM, DO and pH were classify as water mass parameters by numerous previous studies;

**Response:**
1) Sequencing-based community structure has higher resolutions than qPCR-based community structure. For community composition based on sequencing, the dissimilarity matrices were calculated with the relative abundance of OTUs (Operational Taxonomic Units). Similarly, for community composition based on

qPCR, the relative abundance of PA AOA, FL AOA, PA AOB, and FL AOB were used to calculate the dissimilarity matrices, just like merging some OTUs into one OTU. Despite lower resolutions of community composition, the dissimilarity matrices can be calculated and the Mantel and partial Mantel tests can be performed. Similarly, Castellano-Hinojosa et al. (2018) and Huang et al. (2011) also used qPCR data in NDMS analysis and CCA analysis of community structure.

2) Silicate has long been recognized as one of the most common indicators to trace river water in the ocean, and the low salinity and high silicate contents were the best indicators for river source (Moore, 1986). We added a three-dimensional scatter plot in the revised MS (Figure S1; see below) to show the relationships between Potential temperature ($\theta$) ($^{o}$C), salinity, and silicate ($SiO_3^{2-}$) concentration. The waters from the upper estuary where the salinity of most sites was close to zero, had high potential temperature and silicate concentrations. The mixing behaviors of waters occurred at the Humen outlet (sites P07 and A01), and the waters from the off-shore sites (A10 and A11) had high salinity and low potential temperature and silicate concentrations. Therefore, we chose silicate, temperature, and salinity as the indicators to trace estuarine water masses and mixing.

We defined the substrate parameters as nitrogen substrates (ammonium, nitrite, and nitrate), which are related to the $N_2O$ producing processes of nitrification and denitrification. TSM, DO, and pH are not conservative parameters and thus cannot trace water masses. These factors represent the biogeochemical characteristics of waters and could influence the availability of electron donors (or substrates) during nitrification and denitrification. For example, the suspended particles could be beneficial to microbial activity because of nutrients or substrates supply (Belser, 1979; Crump et al., 1998; Ouverney and Fuhrman, 2000; Teira et al., 2006; Zhang et al., 2014); DO concentration and pH also could influence the availability of ammonia, etc. (Geets et al., 2006; Ward, 2008; Martens-Habbena et al., 2009; Zhu et al., 2013; Huesemann et al., 2002; Hutchins et al., 2009; Fulweiler et al., 2010; Beman et al., 2011).

[Figure]

**Figure R2 (Figure S1 in the MS)**: Three-dimensional scatter plot of Potential temperature ($\theta$) ($^{o}$C), salinity, and silicate ($SiO_3^{2-}$).

**References**

Castellano-Hinojosa, A., González-López, J., and Bedmar, E. J.: Distinct effect of nitrogen fertilisation and soil depth on nitrous oxide emissions and nitrifiers and denitrifiers abundance, Biol. Fert. Soils., 54, 829–840, 2018.

Huang, S., Chen, C., Yang, X., Wu, Q., and Zhang, R.: Distribution of typical denitrifying functional genes and diversity of the nirS-encoding bacterial community related to environmental characteristics of river sediments, Biogeosciences, 8, 3041–3051, 2011.

Belser, L. W.: Population ecology of nitrifying bacteria, Annu. Rev. Microbiol., 33, 309–333, 1979.

Beman, J. M., Chow, C. E., King, A. L., Feng, Y., Fuhrman, J. A., Andersson, A., Bates, N. R., Poppa, B. N., and Hutchins, D. A.: Global declines in oceanic nitrification rates as a consequence of ocean acidification, PNAS, 108, 208–213, 2011.

Crump, B. C., Baross, J. A., and Simenstad, C. A.: Dominance of particle-attached bacteria in the Columbia River estuary, USA, Aquat. Microb. Ecol., 14, 7–18, 1998.

Fulweiler, R. W., Emery, H. E., Heiss, E. M., and Berounsky, V. M.: Assessing the Role of pH in Determining Water Column Nitrification Rates in a Coastal System, Estuaries and Coasts, 34, 1095–1102, 2010.

Geets, J., Boon, N., and Verstraete, W.: Strategies of aerobic ammonia-oxidizing bacteria for coping with nutrient and oxygen fuctuations, FEMS Microbiol Ecol., 58, 1–13, 2006.

Huesemann, M. H., Skillman, A. D., and Crecelius, E. A.: The inhibition of marine nitrification by ocean disposal of carbon dioxide, Mar. Pollut. Bull., 44, 142–148, 2002.

Hutchins, D. A., Mulholland, M. R., and Fu, F.: Nutrient cycles and marine microbes in a $CO_2$-enriched ocean, Oceanogr., 22, 128–145, 2009.

Martens-Habbena, W., Berube, P. M., Urakawa, H., de la Torre, J. R., and Stahl, D. A.: Ammonia oxidation kinetics determine niche separation of nitrifying Archaea and Bacteria, Nature, 461, 976–979, 2009.

Moore, W. S., Sarmiento, J. L., and Key, R. M.: Tracing the Amazon component of surface Atlantic water using 228Ra, salinity and silica, J. Geophys. Res., 91, 2574–2580, 1986.

Ouverney, C. C., and Fuhrman, J. A.: Marine planktonic archaea take up amino acids, App. Environ. Microbiol., 66, 4829–4833, 2000.

Ward, B. B. Nitrification in marine systems, In. Capone, D. G., Bronk, D. A., Mulholl, M. R., and Carpenter, E. J. (ed), Nitrogen in the Marine Environment (2nd Edition) [M]. Burlington: Academic Press, 199–261, 2008.

Teira, E., van Aken, H., Veth, C., and Herndl, G. J.: Archaeal uptake of enantiomeric amino acids in the meso-and bathypelagic waters of the North Atlantic, Limnol. Oceanogr., 51, 60–69, 2006.

Zhang, Y., Xie, X., Jiao, N., Hsiao, S. S. Y., and Kao, S. J.: Diversity and distribution of amoA-type nitrifying and nirS-type denitrifying microbial communities in the Yangtze River estuary, Biogeosciences, 11, 2131–2145, 2014.

Zhu, X., Burger, M., Doane, T. A., and Horwath, W. R.: Ammonia oxidation pathways and nitrifier denitrification are significant sources of $N_2O$ and NO under low oxygen availability, PNAS, 110, 6328–6333, 2013.

6. Page 8, line 20, is the 63.0 $\mu$mol/L the hypoxic threshold?

**Response**:

Yes, the DO concentration of 63.0 μmol $L^{-1}$ (equaling to 2 mg $L^{-1}$) is the hypoxic threshold, which was cited from Rabalais et al. (2010).

**Reference**

Rabalais, N. N., D´ıaz, R. J., Levin, L. A., Turner, R. E., Gilbert, D., and Zhang, J.: Dynamics and distribution of natural and human-caused hypoxia, Biogeosciences, 7, 585–619, 2010.

7. Page 9, line 11, please re-phrase this subtitle because only the transcripts of *amo*A and *nir*S genes from two freshwater stations were quantified here;

**Response**:

We re-phrased this subtitle as "*Distributions of amoA and nirS genes along the salinity transect*" in the revised manuscript.

8. Page 12, line 12-13, too much speculation;

**Response**:

We deleted this sentence.

9. Page 12, line 15-26, please move this part into Results section, and again, I don't think the classification for environmental parameters is on the right way;

**Response**:

This part was moved into the Results section as suggested by the reviewer. As for the water mass parameters and the parameters influencing substrate availability, please refer to our response above.

10. Page 12, line 23, "positive correlations between AOB *amo*A abundances and all $N_2O$ parameters", should be except for FL AOB;

**Response**:

This sentence was revised as "*Notably, there were positive correlations between AOB amoA abundances and all N₂O parameters as well as ammonia concentration (Table 2; P <0.05−0.01), except for the FL AOB communities, suggesting a significant influence of AOB on N₂O production*".

11. Page 12, line 27, the results of RDA analysis also should be presented in Results section;

**Response**:

We moved this part into the Results section according to the reviewer's suggestion (3.3 subsection).

12. The most part of first paragraph of 4.3 subsection should be moved into Results section;

**Response**:

The descriptions on Figure 7 in the original 4.3 subsection were moved into the Results section (3.5 subsection).

13. How about the potential role of comammox and *nir*K-type denitrifier for $N_2O$ production in PRE, please discuss it in the 4.3 subsection.

**Response**:

Thanks for the reviewer's suggestion. We added this discussion in the 4.3 subsection. —"*The complete nitrification activity in a single organism (comammox) was newly discovered in terrestrial systems (Daims et al., 2015; Santoro, 2016; Kits et al., 2017). Given the similarity of ammonia oxidation pathway to that of classic AOB, it is possible that comammox may be also involved in $N_2O$ production (Hu and He, 2017). It has been further reported that the comammox Nitrospira inopinata has a low $N_2O$ yield due to the lack of NO reductases and $N_2O$ formed by N. inopinata originates from abiotic conversion of hydroxylamine, indicating that comammox microbes may produce less $N_2O$ during nitrification than AOB (Kits et al., 2019). However, comammox has not been observed to be widespread in the estuary waters.*"

"The *previous study in the Pearl River sediment detected both nirK and nirS, however, the nirK gene abundance was much lower than nirS abundance as nirK only prevails in conditionally oxygen-exposed environment (Huang et al., 2011). Recent studies proposed that nirK-type denitrifiers were responsible for $N_2O$ production despite being less abundant than nirS denitrifiers (Maeda et al., 2017). NirS-type denitrifiers are more likely to be capable of complete denitrification, because the nosZ gene had a*

*significantly higher frequency of co-occurrence with nirS than with nirK, and thus contribute less to N₂O emissions than nirK-type denitrifiers under favorable environmental conditions (Graf et al., 2014). So far, there is no direct evidence that denitrification or nitrifier-denitrification pathways contribute much to N₂O production in the PRE water column, but a release of N₂O into the overlying waters through denitrification was reported in the PRE sediments (Tan et al., 2019). It was possible that the nirK-type denitrifier contributed to N₂O production from the interface between sediment and water. Further study is needed to clarify the potential of nirK-type denitrifier in N₂O production in PRE.*"

**References**

Daims, H., Lebedeva, E. V., Pjevac, P., Han, P., Herbold, C., Albertsen, M., Jehmlich, N., Palatinszky, M., Vierheilig, J., and Bulaev, A.: Complete nitrification by Nitrospira bacteria, Nature, 528, 504–509, 2015.

Graf, D. R. H., Jones, C. M., and Hallin,. S.: Intergenomic comparisons highlight modularity of the denitrification pathway and underpin the importance of community structure for N₂O emissions. PloS One 9: e114118. https://doi.org/10.1371/journal.pone.0114118. s008. 2014

Huang, S., Chen, C., Yang, X., Wu, Q., and Zhang, R.: Distribution of typical denitrifying functional genes and diversity of the nirS-encoding bacterial community related to environmental characteristics of river sediments, Biogeosciences, 8, 3041–3051, 2011.

Hu, H. W., and He, J. Z.: Comammox–a newly discovered nitrification process in the terrestrial nitrogen cycle, J. Soils Sediments, 17, 2709–2717, 2017.

Kits, K. D., Sedlacek, C. J., Lebedeva, E. V., Han, P., Bulaev, A., Pjevac, P., Daebeler, A., Romano, S., Albertsen, M., Stein, L. Y., Daims, H., and Wagner, M.: Kinetic analysis of a complete nitrifier reveals an oligotrophic lifestyle, Nature, 549, 269–272, 2017.

Kits, K. D., Jung, M.Y., Vierheilig, J., Pjevac, P., Sedlacek, C. J., Liu, S., Herbold, C., Stein, L. Y., Richter, A., Wisse, H., Brüggemann, N., Wagner, M., and Daims, H.: Low yield and abiotic origin of N₂O formed by the complete nitrifier Nitrospira inopinata, Nat. Commun., 2019. https://doi.org/10.1038/s41467-019-09790-x

Maeda, K., Toyoda, S., Philippot, L., Hattori, S., Nakajima, K., Ito, Y., and Yoshida, N.: Relative Contribution of nirK- and nirS- Bacterial Denitrifiers as Well as Fungal Denitrifiers to Nitrous Oxide Production from Dairy Manure Compost, Environ. Sci. Technol., 51, 14083–14091, 2017.

Santoro, A. E: The do-it-all nitrifier, Science, 351, 342–343, 2016.

Tan, E., Zou, W., Jiang, X., Wan, X., Hsu, T. C., Zheng, Z., Chen, L., Xu, M., Dai, M., Kao, S.: Organic matter decomposition sustains sedimentary nitrogen loss in the Pearl River Estuary, China, Sci. Total. Environ., 648, 508–517, 2019.

14. Fig. 7. It is a little difficult to understand this figure. It seems like the AOA contributed more for $N_2O$ production and yield in site P01, right?

**Response**:

We attempted to accurately assess the relative contributions of AOA and AOB to $N_2O$ production in the PRE by plotting the $N_2O$ production rates (Fig. 7a) and yields (Fig. 7b) normalized to total AOA and AOB *amo*A gene copies (sum of PA and FL fractions or only PA fractions) or transcripts (only PA fraction) along X-Y axes that represent the relative contributions of AOA and AOB to the total *amo*A gene or transcript pools. For both incubation sites, the more abundant AOB were in the *amo*A gene-based DNA or cDNA pool, the distinctly higher (disproportionately higher relative to enhanced abundance) the average *amo*A gene copy or transcript-specific $N_2O$ production rates (Fig. 7a) and yields (Fig. 7b), suggesting that AOB may have higher cell-specific activity in the upper estuary and thus be more active in producing $N_2O$ than AOA.

15. Table 2, Spearman rank correlation analysis generate a rho () value rather than a R value.

**Response**:

Sorry for this mistake. We revised "R" as "rho ($\rho$) ".

---

## Author Comment (AC2) · 22 Jul 2019

**Manuscript Number: bg-2019-132**
**Manuscript title: Major role of ammonia-oxidizing bacteria in N$_2$O production in the Pearl River Estuary**

**Response to Reviewer #2**

We greatly thank the reviewer for the valuable comments, useful suggestions and careful revisions, based on which we have revised the manuscript. And the point-by-point responses to the comments are shown below.

**Anonymous Referee #2**

Major comments:

More caution is needed on these concentration-based "rate" measurements. Without isotope tracers, very little can be said about actual rates. Evidence for this is in the N$_2$O yields. The yields reported here are about 100X lower than ever reported from cultures or the field (see Ji et al. 2018 GBC).

**Response**:

(1) We agree that the [15]N-labeled methods are of high sensitivity, which is more reliable for low nitrification activity in natural environments (Damashek and Francis 2018). However, in the nutrient-rich estuary waters, changes in nutrient concentrations (ammonium, nitrite, and nitrate) during incubations can be used to calculate nitrification rates. Previous studies reported concentration-based nitrification rates ranging from 0−153.6 μM d$^{-1}$ (Bianchi et al., 1994; Pakulski et al., 1995; Pakulski et al., 2000; De Wilde and De Bie, 2000; Dai et al., 2008; Grundle and Juniper, 2011). In the upper-PRE, where high nitrification activity has been reported in the hypoxic zone (Dai et al., 2008; Hou et al., 2018), the in-situ concentrations of ammonium (33.3-167.2 μM), nitrite (11.6-24.5 μM), and nitrate (82.0-126.1 μM) at the incubation sites were high, so the changing of the nutrients can be sensitively detected during incubations.

(2) We compared our ammonium oxidation rates with the [15]N-labeled-based rates from Hou et al. (2018) (see Table R1 below). Hou et al. reported that during PRE

cruises in July to August 2012 and September 2014, the nitrification rates in the bottom waters of the PRE reached to 40.25 to 40.70 $\mu$mol $L^{-1} d^{-1}$ in the hypoxic sites. Actually, during our cruise, the nitrification rates in the upstream of Humen were also measured using the $^{15}$N-labeled method by simulating in-situ condition incubations, which ranged from 51.05$-$1182.81 nmol $L^{-1} h^{-1}$ (1.23-28.32 $\mu$mol $L^{-1} d^{-1}$) (Zhang, 2016, Thesis). Thus, the nitrification rates estimated in our study are comparable to other studies in the upper reach of PRE.

(3) We compared our $N_2O$ yields with reported from cultures (see Table R1 below). The $N_2O$ yield in the estuarine waters in this study is lower than those from the cultured AOB strains (*Nitrosomonas europaea*, $N_2O$ yield of 2.6$-$26% relative to $NO_2^-$ production), however, the cultures were of high cell densities ($10^9$ cells $mL^{-1}$) (Yoshida and Alexander, 1970) and were incubated with high concentration of ammonium (mM) (~10$-$100X higher than the natural ammonium concentration in the estuary). Obviously, the $N_2O$ yield is the result of the physiological response of ammonia-oxidizing microorganisms to the environment (Mendum et al., 1999), as shown by the previous study on the AOB strain (*Nitrosomonas marina* C-113a) that $N_2O$ yield increased in higher cell concentration cultures and higher ammonium concentration conditions (Frame and Casciotti, 2010).

(4) We also compared our $N_2O$ yields with reported by Ji et al. (2018, GBC) (see Table R1 below). The $N_2O$ yields were 0.003$-$0.06% at >50 $\mu$M $O_2$ and >2% at <0.5 $\mu$M $O_2$ in the Eastern Tropical Pacific (Ji et al., 2018), which are 2$-$10-fold lower than those from the AOB strain cultures under the 10$-$100 $\mu$M $O_2$ concentration (Goreau et al., 1980). Our $N_2O$ yield ranged from 0.21 to 0.32% during nitrification (the initial in-situ $O_2$ concentration: 30$-$61.3 $\mu$M; the terminal $O_2$ concentration: 0.7$-$2.5 $\mu$M). The estimated range of $N_2O$ yield is 0.16$\pm$0.09 to 0.37$\pm$0.23% when fitting our measured $O_2$ concentrations into the empirical equation of the relationship between $N_2O$ yield (%) from nitrification and $O_2$ concentration ($\mu$M) given by Ji et al. (2018), which was comparable with our measured $N_2O$ yield.

**Table R1** Nitrification rates/ammonia oxidation rates and $N_2O$ yield from literatures and our study.

| Study area/Microorganisms | Method | Nitrification rates ($\mu M$ day$^{-1}$) | NH$_3$ concentrations ($\mu M$) | $N_2O$ yield | Reference |
|---|---|---|---|---|---|
| Rhône River plume | Nutrients + N-serve | 0.23 – 2.20 | 0 – 10 | – | Bianchi et al., 1994 |
| Mississippi River | Nutrients | 0 – 13.44 | 0.3 – 2.4 | – | Pakulski et al., 1995 |
| Mississippi & Atchafalaya River plume | Nutrients | 0 – 14.16 | 0.5 – 2.5 | – | Pakulski et al., 2000 |
| Scheldt | $^{14}C$+ methylfluoride / Nutrients | Up to 19.2 / Up to 153.6 | 0 – 400 | 0.10–0.40% | De Wilde & De Bie, 2000 |
| Saanich Inlet | Nutrients + allyithiourea | 0 – 7.66 | 0 – 4.9 | – | Grundle & Juniper, 2011 |
| Pearl River | Nutrients + allyithiourea | 12.47 – 33.10[a] | 1.2 – 341.9 | – | Dai et al., 2008 |
| Pearl River | $^{15}N$, denitrifier method | 40.25 – 40.70[b] | – | – | Hou et al., 2018 |
| Pearl River | $^{15}N$, denitrifier method | 1.23–28.32 | – | – | Zhang, 2016 |
| Eastern Tropical Pacific | $^{15}N$ tracer | – | 0 – 0.5 | 0.003−0.06%[c] >2%[d] | Ji et al., 2018 |
| *Nitrosomonas europaea* | Nutrients, $N_2O$ | – | – | 2.6–26%[e] | Yoshida & Alexander, 1970 |

| | | | | | |
|---|---|---|---|---|---|
| | concentrations | $2.38 – 23.8$[f] | $7.14–714.3$[g] | $2.6–18\%$[h] | |
| *Nitrosomonas sp.* | Nutrients, $N_2O$ | | | $0.26–0.99\ \%$[i] | |
| (Marine) | concentrations | – | – | $2.5–9.9\ \%$[j] | Goreau et al., 1980 |
| *Nitrosomonas marina* | Nutrients, $N_2O$ | | | | |
| *C-113a* | isotopic analyses | – | $50$[g] | $0.04–2.2\%$ | Frame & Casciotti, 2010 |
| Pearl River | Nutrients | $11.28–26.88$[a] | $33.3 – 167.2$ | $0.21–0.32\%$ | This study |

[a] The ammonia oxidation rates observed at the upper reach of PRE in summer.

[b] The nitrification rates observed at the upper estuary where the $O_2$ concentration from 0.67 to 1.41 mg $L^{-1}$, which were little lower than that ranging from 0.9 to 2.0 mg $L^{-1}$ in our study.

[c] $N_2O$ yield from ammonia oxidation under the $O_2 > 50$ μM.

[d] $N_2O$ yield from ammonia oxidation under the $O_2 < 0.5$ μM.

[e] This experiment was designed to study the influence of different levels of ammonium concentration on $N_2O$ formation by *Nitrosomonas europaea*.

[f] The ammonia oxidation rates were estimated based on the difference of ammonium concentrations between initial- and terminal- incubation time using the data from Yoshida and Alexander, 1970.

[g] Ammonium concentrations in the medium.

[h] This experiment was designed to study the influence of cells in different growth stages on $N_2O$ formation by *Nitrosomonas europaea*.

[i] $N_2O$ yield from ammonia oxidation under the $O_2$ ranging from 5–20% (56.3 –218.8 μM ).

[j] $N_2O$ yield from ammonia oxidation under the $O_2$ ranging from 0.5–1% (5.6 –10.9 μM).

**References**

Bianchi, M., Bonin, P., and Feliatra.: Bacterial nitrification and denitrification rates in the Rhône River plume (northwestern Mediterranean Sea), Mar. Ecol-Prog. Ser., 103, 197–202, 1994.

Damashek, J., and Francis, C. A.: Microbial Nitrogen Cycling in Estuaries: From Genes to Ecosystem Processes, Estuaries and Coasts, 41, 626–660, 2018.

Dai, M., Wang, L., Guo, X., Zhai, W., Li, Q., He, B., and Kao, S. J.: Nitrification and inorganic nitrogen distribution in a large perturbed river/estuarine system: the Pearl River Estuary, China, Biogeosciences, 5, 1545–1585, 2008.

DeWilde, H. P. J., and De Bie, M. J. M.: Nitrous oxide in the Schelde Estuary: production by nitrification and emission to the atmosphere, Mar. Chem., 69, 203–216, 2000.

Frame, C. H., and Casciotti, K. L.: Biogeochemical controls and isotopic signatures of nitrous oxide production by a marine ammonia-oxidizing bacterium, Biogeosciences, 7, 2695–2709, 2010.

Goreau, T. J., Kaplan, W. A., Wofsy, S. C., McElroy, M. B., Valois, F. W., and Watson, S. W.: Production of $NO_2^-$ and $N_2O$ by Nitrifying Bacteria at Reduced Concentrations of Oxygen, Appl. Environ. Microbiol., 40, 526–532, 1980.

Grundle, D. S., and Juniper, S. K.: Nitrification from the lower euphotic zone to the sub-oxic waters of a highly productive British Columbia fjord, Mar. Chem., 126, 173–181, 2011.

Hou, L., Xie, X., Wan, X., Kao, S. J., Jiao, N., and Zhang, Y.: Niche differentiation of ammonia and nitrite oxidizers along a salinity gradientnfrom the Pearl River estuary to the South China Sea, Biogeosciences, 15, 5169–5187, 2018.

Ji, Q., Buitenhuis, E., Suntharalingam, P., Sarmiento, J. L., and Ward, B. B.: Global Nitrous Oxide Production Determined by Oxygen Sensitivity of Nitrification and Denitrification, Global Biogeochem. Cy., 32, 1790–1802, 2018.

Mendum, T. A., Sockett, R. E., and Hirsch, P. R.: Use of molecular and isotopic techniques to monitor the response of autotrophic ammonia-oxidizing populations of the beta subdivision of the class Proteobacteria in arable soils to nitrogen fertilizer, Appl. Environ. Microbiol., 65, 4155–4162, 1999.

Pakulski, J. D., Amon, R., Eadie, B., and Whitledge, T.: Community metabolism and

nutrient cycling in the Mississippi River: evidence for intense nitrification at intermediate salinities. Mar. Ecol-Prog. Ser., 117, 207–218, 1995.

Pakulski, J. D., Benner, R., Whitledge, T., Amon, R., Eadie, B., Cifuentes, L., Ammerman, J., and Stockwell, D.: Microbial metabolism and nutrient cycling in the Mississippi and Atchafalaya River plumes. Estuar. Coast. Shelf S., 50, 173–184, 2000.

Yoshida, T., and Alexander, M.: Nitrous Oxide Formation by Nitrosomonas europaea and Heterotrophic Microorganisms, Soil Sci. Soc. Amer. Proc., 34, 880–882, 1970

Zhang, X: Rates and Influence Factor of Water Nitrification and Inorganic Nitrogen Uptake in Pearl River Estuary, MA.Sc thesis, Xiamen University, Xiamen, China, 22 pp., 2016.

The strength of the correlation between genes and rates absolutely cannot be used to apportion a relative importance of one group of ammonia oxidizers or the other to the total rates. Nothing can be concluded from the data presented about who the important nitrifiers are. One possibility would be to obtain to a range of cell-specific ammonia oxidation rates from the literature and then use those in combination with the qPCR data to calculation the relative contribution of each group to the observed "rates."

**Response**:

The cell-specific ammonia oxidation rates, nitrite production rates, and $N_2O$ production rates from the literature on AOA and AOB strains varied in a very large range, due to the different species cultures, cell densities, cell stages, and incubation conditions such as $O_2$ or substrates concentrations (see Table R2 below). It is fairly uncertain to use these greatly varying cell-specific rates from cultures to estimate the contribution of AOA and AOB to the $N_2O$ production in natural environments. Notably, although the cell-specific $N_2O$ production rates from AOB and AOA strains varied greatly, the $N_2O$ yields from the AOB strains, ranging from 0.09 to 26 % (Table R2), were generally higher than the $N_2O$ yield from the AOA strains (0.002−0.09%; Table R2). In addition, the higher $N_2O$ yield from AOB has been observed in soils although the abundance of AOB was lower than AOA (Hink et al., 2017, 2018).

We admit that the conclusions of this study mainly based on the correlation analysis and statistical analysis between multi-parameters. But there are two analyses providing more strong evidence supporting these statistical analyses:

(1) We attempted to accurately assess the relative contributions of AOA and AOB to $N_2O$ production in the PRE by plotting the $N_2O$ production rates (Fig. 7a in the MS) and yields (Fig. 7b in the MS) normalized to total AOA and AOB *amo*A gene copies (sum of PA and FL fractions or only PA fractions) or transcripts (only PA fractions) along X-Y axes that represent the relative contributions of AOA and AOB to the total *amo*A gene or transcript pools. For both incubation sites, the more abundant AOB were in the *amo*A gene-based DNA or cDNA pool, the distinctly higher (disproportionately higher relative to enhanced abundance) the average *amo*A gene copy or transcript-specific $N_2O$ production rates (Fig. 7a) and yields (Fig. 7b), suggesting that AOB may have higher cell-specific activity in the upper estuary and thus be more active in producing $N_2O$ than AOA.

(2) The values of N stable isotopes in $N_2O$ ($\delta^{15}N$) were analyzed. The much lower $^{15}N$-$N_2O$ (−27.9 to −12.6‰) upstream of the Humen outlet is consistent with AOB nitrification or denitrification processes, whereas enriched $^{15}N_2O$ (5.2−7.1‰) in the lower reaches approaches AOA nitrification and air $^{15}N$-$N_2O$ (Santoro et al., 2011). Taken together, the isotopic compositions of $N_2O$ (Fig. 2h in the MS) and $N_2O$ concentration distribution (Fig. 2e−g) suggest that the high concentrations of $N_2O$ (oversaturation) were produced from strong nitrification by AOB and probably concurrent minor denitrification in the upper estuary, however in the lower reaches, low concentrations of $N_2O$ could be explained by AOA nitrification or water atmospheric exchange of $N_2O$.

**Table R2** Cell-specific ammonia oxidation rates, cell-specific N$_2$O production rates, and N$_2$O yield from archaeal and bacterial strains.

| Microorganisms | Species (source of isolate) | Ammonia oxidation rates (fmol cell$^{-1}$ h$^{-1}$)[a] | N$_2$O production rates (fmol cell$^{-1}$ h$^{-1}$)[b] | N$_2$O yield[c] | Reference |
|---|---|---|---|---|---|
| AOA | | 19.0 | – | – | Martens-Habbena et al., 2009 |
| | *Nitrosopumilus maritimus* (Marine) | – | 0.02–1.01 | 0.002–0.026% | Löscher et al., 2012 |
| | | – | – | 0.03–0.05% | Stieglmeier et al., 2014 |
| | *Nitrososphaera viennensis* (Soil) | 2.6–2.8 | 0.004–0.005 | 0.03–0.09% | Stieglmeier et al., 2014 |
| AOB | *Nitrosomonas sp.* (Marine) | 2.0–15.4 | 0.04–0.21 | 0.26–9.9% | Goreau et al., 1980 |
| | *Nitrosomonas marina* (Marine) | 0.9–4.9 | – | – | Glover, 1985 |
| | | 13.7–31.3 | – | – | Glover, 1985 |
| | *Nitrosococcus oceanus* (Ocean) | 83.3 | – | – | Waston, 1965 |
| | | – | – | 0.26±0.1% | Goreau et al., 1980 |
| | *Nitrosomonas europaea* (Soil) | 12.4–18.3 | – | 2.6–26% | Yoshida & Alexander, 1970 |
| | | | | 0.47±0.1% | Goreau et al., 1980 |
| | *Nitrosospira tenuis* NV12 (Soil) | – | 0.002 | – | Shaw et al., 2006 |
| | *Nitrosomonas europaea* ATCC 19718 | – | 0.06 | – | Shaw et al., 2006 |
| | *Nitrosospira multiformis* (Soil) | – | – | 0.09–0.27% | Stieglmeier et al., 2014 |

| | | | | |
|---|---|---|---|---|
| *Nitrosolobus multiformis* (Soil) | – | – | 0.09 ±0.02% | Goreau et al., 1980 |
| *Nitrosospira briensis* (Soil) | – | – | 0.11 ±0.04% | Goreau et al., 1980 |

[a] The units for cell-specific ammonia oxidation rates in the citied references were unified as fmol cell$^{-1}$ h$^{-1}$.

[b] The units for cell-specific $N_2O$ production rates in the citied references were unified as fmol cell$^{-1}$ h$^{-1}$.

[c] The range of $N_2O$ yield of different cell densities under different $O_2$ conditions.

**Response**:

Very sorry for this problem. We added the estuarine studies literature review on nitrification and $N_2O$ production in the Introduction and Discussion of the revised version.

"*Estuaries, being highly impacted by coastal nutrient pollution and eutrophication due to anthropogenic activity, play a significant role in nitrogen cycling at the land-sea interface (Bricker et al., 2008; Damashek et al., 2016). Estuarine and coastal regimes have long been recognized major zones of $N_2O$ production in the marine system (Seitzinger and Kroeze, 1998; Mortazavi et al., 2000; Usui et al., 2001; Kroeze et al., 2010; Allen et al., 2011). In particular, the eutrophic estuaries with significant nitrification and extensive oxygen-deficient zones (ODZs) has been considered as the hot spot regions for $N_2O$ production (Abril et al., 2000; DeWilde and De Bie, 2000; Garnier et al., 2006; Lin et al., 2016), and nitrification is often credited as the dominant $N_2O$ production pathway in estuaries (deBie et al. 2002; Barnes and Upstill-Goddard 2011; Kim et al. 2013; Lin et al. 2016; Huertas et al., 2018; Laperriere et al., 2019). The estuaries have been reported with high of $N_2O$ saturation and large $N_2O$ flux range, and $N_2O$ concentrations are highly variable (Hashimoto et al., 1999; deWilde and de Bie 2000; deBie et al. 2002; Xu et al., 2005; Chen et al., 2008; Rajkumar et al., 2008; Zhang et al., 2010; Barnes and Upstill-Goddard 2011; Stocker et al. 2013; Wu et al., 2013; Murray et al., 2015; Lin et al., 2016; Wells et al., 2018). The dynamics of $N_2O$ emissions in these ecosystems are regulated by complex physical and biological processes, e.g. mixing between freshwater and oceanic waters influenced biogeochemistry of estuarine waters and microbial activity in the water column (Huertas et al., 2018; Laperriere et al., 2019), yet studies on estuarine $N_2O$ production and emission in the water column based on integrated biogeochemical parameters, function genes, and in-situ incubations remain sparse.*"

*"Previous studies also proposed that nitrification may be the major source of N₂O production in the water column in estuarine systems, such as the Guadalquivir estuary (Huertas et al., 2018), the Schelde estuary (De Wilde and De Bie, 2000), and the Chesapeake Bay estuary (Laperriere et al., 2019). However, in the estuarine sediments, N₂O production was attributed to both nitrification and denitrification, such as the Tama estuary of Japan (Usui et al., 2001) and the Yangtze Estuary of China (Liu et al., 2019; Wang et al., 2019), where denitrification is the major nitrogen removal pathway with the N₂O production and consumption"*

All the physical dynamics in the system have been reduced to a very naive "water mass" identification. Basic concepts in estuarine biogeochemistry are absent–for example, using salinity as a conservative tracer in a two-end member mixing model to determine production and loss of the various biogeochemical parameters.

**Response**:

Silicate has long been recognized as one of the most common indicators to trace river water in the ocean, and the low salinity and high silicate contents were the best indicators for river source (Moore, 1986). Therefore, we used temperature, salinity, and silicate to trace water masses and mixing in the estuary transect. We believe that fresh and saline water masses mixing might directly mix nitrifiers and denitrifiers as well as $N_2O$ from fresh and saline waters. Thus, in order to peel off the directly mixing effects, we used Partial Mantel tests to eliminate the co-varying of water mixing, substrate concentrations, and $N_2O$ production along the transect and to identify the intrinsic/direct relationship between ammonia oxidizers and $N_2O$

production. We revised the relevant statements on "water mass" and emphasized "water mixing" throughout the MS for a clearer expression.

According to the reviewer's suggestion, we also performed the end-member mixing analysis in the supplementary materials of the revised MS.

(1) Figure R1 (see below) is a three-dimensional scatter plot showing the relationships between Potential temperature ($\theta$) ($^{o}C$), salinity, and silicate ($SiO_3^{2-}$) concentration. The waters from the upper estuary where the salinity of most sites was close to zero, had high potential temperature and silicate concentrations. The mixing behaviors of waters occurred at the Humen outlet (sites P07 and A01), and the waters from the off-shore sites (A10 and A11) had high salinity and low potential temperature and silicate concentrations. Figure R2 (below) shows the linear relationships between Potential temperature ($\theta$) or silicate and salinity as well as between observed and conservative silicate. These analyses indicate a two end-member mixing in this estuary and silicate, temperature, and salinity can be used as the indicators to trace estuarine water masses and mixing.

(2) Figure R3 (below) shows the scatter plot of $R$N$_2$O (the two end-member mixing model prediction minus field observation) versus salinity as well as the relationship between $R$N$_2$O and $\Delta$NH$_3$/NH$_4^+$ in the Lingdingyang. $R$N$_2$O indicates biogeochemical produced and then outgassing N$_2$O through the water-air exchange (see Lin et al., 2016). $R$N$_2$O decreased with salinity indicating N$_2$O removal through the estuarine mixing behavior and/or water-air exchange. Meanwhile, the positive correlation between $R$N$_2$O and $\Delta$NH$_3$/NH$_4^+$ (ammonium consumption) suggested that N$_2$O may be mostly related to ammonia oxidation in Lingdingyang.

[Figure]

**Figure R1**: Three-dimensional scatter plot of Potential temperature (θ) ($^{o}$C), salinity, and silicate ($SiO_3^{2-}$).

[Figure]

**Figure R2:** Relationships between (a) potential temperature (θ) (°C) or (b) silicate and salinity in the PRE estuary. The fitted curves represent the conservative distribution controlled by physical mixing processes. (c) Relationship between observed and conservative silicate concentrations. The straight line represents a 1:1 reference line.

[Figure]

**Figure R3:** (a) $R$N$_2$O versus salinity in Lingdingyang; (b) the relationship between $R$N$_2$O and $\Delta$NH$_3$/NH$_4^+$.

**Response**:

Sorry for the confusion. We only used Eq. (8) to estimate $N_2O$ yield (the ratio of $N_2O$ production rate to ammonia oxidation rate). Eq. (9) was deleted in the revised MS.

In addition, we compared the $N_2O$ yield estimated by Eq (8) and Eq (9) for site P05, where the only nitrification occurred during 12 hour-incubation. The $N_2O$ yield estimated by Eq (8) and Eq (9) was 0.21% and 0.19%, respectively in the surface water and 0.32% and 0.33%, in the bottom water.

More details are needed about how you arrived at the Schmidt number for $N_2O$. Is this the Raymond and Cole reference?

**Response**:

In the Eq. (5) for $N_2O$ flux estimation, $k$ (cm h$^{-1}$) is the gas transfer velocity depending on wind and water temperatures. In this study, $k_{600}$, the gas transfer velocity at a Schmidt number of 600, was used for the estuarine system (Raymond and Cole, 2001). The Schmidt number (Sc) is defined as the kinematic viscosity of water divided by the diffusion coefficient of the gas, and is usually expressed as a function of temperature and salinity (Wanninkhof, 1992). For steady winds with the average climatological wind speed at 10 m above the water surface, the relationship between gas transfer and wind speed is estimated using Eq. (6) according to Wanninkhof (1992):

$$k_{600} = 0.31 \times u^2_{10} \times (\mathrm{Sc}/600)^{-0.5} \tag{6}$$

For $N_2O$ in waters of salinity <35 and temperature ranging from 0−30 ℃, $\mathrm{Sc_{N2O}}$ is estimated using the following Eq. (7) according to Wanninkhof (1992):

$$\mathrm{Sc_{N2O}} = 2055.6 - 137.11\ t + 4.3173\ t^2 - 0.05435\ t^3 \tag{7}$$

We added more details in the revised 2.2 section as suggested by the reviewer.

Need additional details of the calibration of the isotopic values.

**Response**:

We added more details of the calibration of the isotopic values in the revised version (2.2 subsection).

"*The $\delta^{15}N$ values in $N_2O$ were analyzed by quantifying the molecular ions ($N_2O^+$, m/z 44, 45 and 46) of $N_2O$ by isotope ratio mass spectrometry (IRMS) at the State Key Laboratory of Soil and Sustainable Agriculture, Institute of Soil Science, Chinese Academy of Sciences, Nanjing. The values for $\delta^{15}N$-$N_2O$ in the sample were calculated using the raw peak area ratios of 45/44 for a reference gas, which was previously calibrated using stable isotope $N_2O$ standard gas produced by SHOKO, Co., Ltd., Japan ($\delta^{15}N_{Air}$ = −0.320‰), and the sample peak (Frame and Casciotti, 2010; Mohn et al., 2014). In this study, the precision of the isotope method for $\delta^{15}N$-$N_2O$ was estimated to be 0.3‰.*"

**Response**:
The estimated water–air $N_2O$ fluxes were 100.4 to 344.0 μmol m$^{-2}$ d$^{-1}$ upstream and decreased in Lingdingyang (42.4 to -2.6 μmol m$^{-2}$ d$^{-1}$). Taken together, the PRE was a strong source. We revised this sentence as "*Together, the PRE acts as a $N_2O$ source*".

p.11 lines 19-26: This paragraph confuses some important concepts. Some of these numbers are the isotopic composition of $N_2O$ produced by ammonia oxidizers, but

some of these numbers are the isotope effect (epsilon). Also, the isotopic composition of the $N_2O$ being produced by nitrification is dependent on the isotopic composition of the $NH_3$ being oxidized, for which no measurements or even estimates are provided.

**Response**:

We only used the isotopic composition of $N_2O$ in this paragraph and supplementary Table S2. But sorry for the wrong supplementary Table S2 title. We revised it as "Isotopic composition of $^{15}N$-$N_2O$ during bacterial and archaeal ammonia oxidation, bacterial nitrifier-denitrification, and bacterial denitrification."  These data all are from literature.

p.12 lines 15-17: Doesn't make sense to refer to 'water masses' in estuaries. There is a tremendous amount of mixing that leads to variation in these parameters. Just because something is a different salinity doesn't mean it's a different 'water mass.' These parameters are just 'hydrography.'

**Response**:

We revised "water masses parameters" as "hydrographic parameters".

p. 12 lines 15-28 and p. 13 lines 1-18 A lot of results presented that should be moved to the results section.

**Response**:

Thanks for the reviewer's suggestion. We moved this part into the Results section (3.3 subsection).

p. 12 line 27 "ammonia oxidizer community" The use of the word "community" throughout the paper is confusing. More accurate to state the abundances of AOA and AOB?

**Response**:

We revised "ammonia oxidizer community" as "AOA and AOB distribution", and moved this part into the Results section (3.3 subsection) according to the reviewer's suggestion.

p. 24 Fig 1 i,j It looks like two different slopes in the data upstream and Lingdingyang. This could be quantified using a break point analysis.

**Response**:

Thanks for the suggestion. We re-quantified using a break point analysis. See below.

[Figure]

**Figure R4:** (i) $\Delta N_2O$ vs. DO and (j) $N_2O$ flux vs. DO.

p. 32 I found this figure confusing. Perhaps it would be useful to have a table with the data presented in the figure? It is unclear using AOB and AOA% if the normalized $N_2O$ production values are a result of the $N_2O$ yield or low/high amoA abundance.

**Response**:

We added Table S3 (below) with the data that presented in the figure. We attempted to accurately assess the relative contributions of AOA and AOB to $N_2O$ production in the PRE by plotting the $N_2O$ production rates (Fig. 7a) and yields (Fig. 7b) normalized to total AOA and AOB *amo*A gene copies (sum of PA and FL fractions or only PA fraction) or transcripts (only PA fraction) along X-Y axes that represent the relative contributions of AOA and AOB to the total *amo*A gene or transcript pools. For both incubation sites, the more abundant AOB were in the *amo*A gene-based DNA or cDNA pool, the distinctly higher (disproportionately higher relative to enhanced abundance) the average *amo*A gene copy or transcript-specific $N_2O$ production rates (Fig. 7a) and yields (Fig. 7b), suggesting that AOB may be more active in producing $N_2O$ than AOA. AOB may contribute the major part in $N_2O$ production with their high cell-specific activity in the upper estuary.

**Table R3 (Table S3 in the MS)** The abundances of DNA/cDNA-based *amo*A gene and the $N_2O$ production rates and yields normalized to total *amo*A gene copy or transcript numbers of AOA and AOB in a given sample at the incubation experiment sites.

| Site_Layer | DNA-based AOB (All) (copies $L^{-1}$) | DNA-based AOA (All) (copies $L^{-1}$) | $N_2O$ production rates (All) (f mol cell$^{-1}$ h$^{-1}$) | $N_2O$ yields (All) ($10^{-6}$) | DNA-based AOB (PA) (copies $L^{-1}$) | DNA-based AOA (PA) (copies $L^{-1}$) | $N_2O$ production rates (PA) (f mol cell$^{-1}$ h$^{-1}$) | $N_2O$ yields (PA) ($10^{-6}$) | cDNA-based AOB (PA) (copies $L^{-1}$) | cDNA-based AOA (PA) (copies $L^{-1}$) | $N_2O$ production rates (PA) (f mol cell$^{-1}$ h$^{-1}$) | $N_2O$ yields (PA) ($10^{-6}$) |
|---|---|---|---|---|---|---|---|---|---|---|---|---|
| P05_S | 14030 | 34427 | 23.70 | 21.30 | 12125 | 29082 | 27.90 | 25.00 | 382928 | 138646 | 2.20 | 1.97 |
| P05_B | 87915 | 397740 | 2.90 | 3.25 | 77820 | 357308 | 3.24 | 3.63 | 89559 | 12559 | 13.80 | 15.50 |
| P01_S | 19623 | 642905 | 0.91 | 1.93 | 9343 | 578974 | 1.02 | 2.18 | 500 | 461578 | 1.30 | 2.77 |
| P01_B | 21334 | 251163 | 5.91 | 5.47 | 16458 | 221184 | 6.77 | 6.27 | 362 | 7436 | 206.00 | 191.00 |

S, surface; B, bottom; All, sum of particle-attached and free-living fractions; PA, particle-attached fraction.

Technical corrections

[revised manuscript text omitted]

p. 4 lines 15-16 Should be moved to results section 2.2 discussing ammonia analysis "Ammonia/ammonium concentrations were analyzed onboard."

**Response**:

We moved this sentence to section 2.2.

"*Ammonia was measured using the indophenol blue spectrophotometric method (Pai et al., 2001) on board*"

p. 4 line 25-26 What salinity, temperature and DO probes were used?

**Response**:

We revised this sentence.

"*Temperature and salinity were determined with a SBE 25 conductivity–temperature–depth/pressure unit (Sea-Bird Co.). DO were determined with a SBE 43 Dissolved Oxygen Sensor (Sea-Bird Co.). All DO concentrations used in this study was measured using the Winkler method.*"

p. 5 lines 5-23 Not all variables in the equations are defined.

**Response**:

We added more details for the equations and defined the variables in the revised section 2.2 (see below highlighted).

"*The excess N$_2$O (ΔN$_2$O) and N$_2$O saturation were calculated with Eq. (1) and (2):*

$$\Delta N_2O = N_2O_{observed} - N_2O_{equilibrium} \tag{1}$$

$$S(\%) = N_2O_{observed} / N_2O_{equilibrium} \times 100\% \tag{2}$$

*where $N_2O_{observed}$ represents the measured concentrations of $N_2O$ in the water, and the equilibrium values of $N_2O$ ($N_2O_{equilibrium}$) are calculated by Eq. (3) and (4) (Weiss and Price, 1980):*

*$N_2O_{equilibrium} = xF$* (3)

*$lnF = A_1 + A_2(100/T) + A_3 ln(T/100) + A_4(T/100)^2 + S[B_1+B_2(T/100) + B_3(T/100)^2]$* (4)

*where $x$ is the mole fraction of $N_2O$ in the atmosphere and $T$ is the absolute temperature. In this study, we used the global mean atmospheric $N_2O$ (327 ppb) from 2015 (http://www.esrl.noaa.gov/gmd). The fitted function F with constants A1, A2, A3, A4, B1, B2, and B3 was proposed by Weiss and Price (1980).*

*The $N_2O$ flux ($F_{N2O}$, $\mu mol\ m^{-2}\ d^{-1}$) through the air–sea interface was estimated based on Eq. (5):*

*$F_{N2O} = k_{N2O} \times \rho \times K_H^{N2O} \times \Delta pN_2O = k_{N2O} \times 24 \times 10^{-2} \times (N_2O_{observed} - N_2O_{equilibrium})$* (5)

*where $k_{N2O}$ ($cm\ h^{-1}$) is the $N_2O$ gas transfer velocity depending on wind and water temperatures, $K_H^{N2O}$ is the solubility of $N_2O$, and $\Delta pN_2O$ is the average sea-gas $N_2O$ partial pressure difference. In this study, $k_{600}$, the gas transfer velocity at a Schmidt number of 600, is used for the estuarine system (Raymond and Cole, 2001). The Schmidt number (Sc) is defined as the kinematic viscosity of water divided by the diffusion coefficient of the gas, and is usually expressed as a function of temperature and salinity (Wanninkhof, 1992). For steady winds with the average climatological wind speed at 10 m above the water surface, the relationship between gas transfer and wind speed is estimated using Eq. (6) according to Wanninkhof (1992):*

*$k_{600} = 0.31 \times u^2_{10} \times (S_c/600)^{-0.5}$* (6)

*For $N_2O$ in waters of salinity <35 and temperature ranging from 0−30 ℃, $Sc_{N2O}$ is estimated by the following Eq. (7) according to Wanninkhof (1992):*

*$Sc_{N2O} = 2055.6 − 137.11\ t + 4.3173\ t^2 − 0.05435\ t^3$* (7)

*where t is in situ temperature of the sampling site.*

p. 31 Fig 6 Should axes be swapped?

**Response**:

We swapped X and Y axes in Fig 6 of the revised version according to the reviewer's suggestion.

---

## Author Response (AR1)

Dear Dr. Pantoja,

Thank you for taking the time to handle our manuscript and your assessment. We have carefully addressed each comment from you and two referees and tried our best to improve the manuscript according to the suggestions. Our responses to all comments are listed below. We welcome any further comments. Thank you again for your time and kind efforts.

Best wishes,

Yao Zhang

**Manuscript Number: bg-2019-132**
**Manuscript title: Major role of ammonia-oxidizing bacteria in N$_2$O production in the Pearl River Estuary**

**Response to Editor**

Comments to the Author:
Review of bg-2019-132 Major role of ammonia-oxidizing bacteria in N$_2$O production in the Pearl River Estuary

August 7, 2019

Dear Dr. Zhang

Thanks for submitting responses to reviewers of bg-2019-132 "Major role of ammonia oxidizing bacteria in N$_2$O production in the Pearl River Estuary". Based on those and my own reading I think the article is promising due to sound data and interpretation but it is not publishable in the present form in Biogeosciences since it lacks some structure as pointed out by Reviewer 1. This aspect precludes full understanding of your work in mainly two aspects:

1) Mixing of results and discussion such as in Figure 2 that shows field data combined with panels i and j that, according to text, ΔN$_2$O is calculated for incubations so, they are probably results from incubations. Since there is no explanation of experimental treatments with variable O$_2$ concentration, I would assume that those are derived from field sampling, right? If that were the case, results are mixed with discussion in this figure and text in page 8, paragraph starting in L24. Alternatively, oxygenation conditions of incubations are missing in the method section.

**Response**:

$\Delta N_2O$ in Figure 2 is not calculated for incubations; it is calculated from the filed data as the difference between the measured concentrations of $N_2O$ in the waters and the estimated equilibrium values of $N_2O$ (based on the global mean atmospheric $N_2O$ from http://www.esrl.noaa.gov/gmd). We described the calculation in the Methods section 2.2 (Page 6, Lines 1−11). $O_2$ concentration in Figure 2 is also from filed sampling. So, Figure 2 and text in the original page 8, L24 (Page 10, Lines 10−19 in the revised version) all are the results from field, not mixed with discussion.

$\Delta N_2O$ in Table 2 (original Table 1) is calculated for incubations as the variation of $N_2O$ concentration along incubation time. This was described in the Methods section 2.4 (Page 8, Lines 22−24). To distinguish $\Delta N_2O$ from field data (the excess $N_2O$) and incubations, we revised $\Delta N_2O$ in field as "$\Delta N_2O_{excess}$" throughout the manuscript (Page 1, Line 16; Page 6, Lines 1, 2; Page 10, Lines 12, 13, 17; Page 13, Line 19; Page 15, Line 15; Page 17, Lines 21, 24; Page 31, Lines 2, 3; Page 41, Table 1). $O_2$ concentration was not measured during incubations; we listed the in situ $O_2$ concentration of incubation sites in Table S1.

2) Better explanation is needed regarding "concentration-based "rate" measurements …" (Reviewer 2) that not only refers to "… changing of the nutrients can be sensitively detected during incubations" as you pointed out, but also to multiple and simultaneous sources and sinks, therefore at the most you obtain a net rate since it is likely that these nutrients are simultaneously removed.
In any case, this aspect is not clear in the text. Please clearly show how you interpret each of rates (net production or decay/ incubation time) of Table 1:
$\Delta N_2O$ (nmol $L^{-1}h^{-1}$)
$\Delta NO_3^-$ (μmol $L^{-1}h^{-1}$)
$\Delta NO_2^-$ (μmol $L^{-1}h^{-1}$)
$\Delta(NH_3 + NH_4^+)$ (μmol $L^{-1}h^{-1}$). You mean $NH_3 + NH_4^+$ instead of $NH_3/NH_4^+$ (a ratio), don't you?

**Response**:

Many thanks for your comment. We agree that concentration-based "rate" measurements essentially give a net rate. We clarified these rates as net rates in Table

2 (original Table 1) and the text. Please see below.

Table 2: "*"These rates are net rates since Δ(NH₃+NH₄⁺) is the net consumption and ΔNO₂⁻, ΔNO₃⁻, and ΔN₂O is the net production during incubation.*" (Page 42)

Methods 2.4 subsection: "*All of the concentration-based rates described from the incubations represent net rates.*" (Page 8, Lines 26−27).

We also revised $\Delta NH_3/NH_4^+$ as $\Delta(NH_3 + NH_4^+)$ in Table 2 as well as Figures 2, 5 (original Figure 4), and 6.

A cartoon similar to the one below may help to explain your rationale for interpreting these rates and better sustain conclusions such as "… results clearly indicate that nitrification occurred during the entire P01 incubations, and suggest that denitrification may be present in the ending phase… "(Page 10, L11-12).

[Figure]

**Response**:

Thanks for your suggestion. We added a diagram in Figure 5 (original Figure 4) for a better understanding of our results. Please see below:

[Figure]

**Figure R1 (Figure 5 in the revised MS):** A diagram showing the transformations of nitrogen compounds and $N_2O$ productions during incubation experiments. Nitrification (1) occurred during the entire P01 and P05 incubations and denitrification (2 and/or 3) may be present in the end phase of the P01 incubation. The gray arrows indicate the pathways of nitrogen loss unanalyzed here, and the gray compounds indicate the unmeasured nitrogen compound.

In addition, please consider the following:

1. Pag. 1. Line 19. All "$N_2O$ parameters". Do you mean $N_2O$-related parameters?

**Response**:

Yes, we mean $N_2O$-related parameters. Revised as suggested (Page 1, Line 19; Page 15, Lines 5, 10 and 15).

2. P. 1, Line 22-25 "Taken together, the in situ incubation experiments, $N_2O$ isotopic composition and concentrations, and gene datasets suggested that the high concentration of $N_2O$ (oversaturated) is mainly produced from strong nitrification by the relatively high abundance of AOB in the upper reaches as the major source of $N_2O$ emitted to the atmosphere in the whole estuary. "

What is the evidence for the whole estuary? What about seasonal variability?

**Response**:

Sorry, "the whole estuary" could be confusing. We mean that the upper reaches acts as the major source of $N_2O$ emitted to the atmosphere in the Pearl River Estuary. We revised this sentence as "*Taken together, the in situ incubation experiments, $N_2O$ isotopic composition and concentrations, and gene datasets suggested that the high concentration of $N_2O$ (oversaturated) is mainly produced from strong nitrification by the relatively high abundance of AOB in the upper reaches and is ==the major source of $N_2O$ emitted to the atmosphere in the Pearl River Estuary==.*" (Page 1, Lines 22−26)

While this study was performed in summer, the Pearl River Estuary acts as a net source of $N_2O$ all year according to Lin et al. (2016). Lin et al. reported the seasonal variability of $N_2O$ distributions in the Pearl River Estuary based on six cruises covering Spring, Summer, Autumn, and Winter. The results indicate that the

saturations of $N_2O$ in the water column varied from 101–3800% along the Pearl River Estuary, acting as a net source of atmospheric $N_2O$, and $N_2O$ production was predominantly modulated by nitrification in the upper estuary. There are significantly higher $N_2O$ concentrations and elevated $N_2O$ fluxes during winter and spring compared with summer and autumn.

**Reference:**

Lin, H., Dai, M., Kao, S. J., Wang, L., Roberts, E., Yang, J., Huang, T., and He, B.: Spatiotemporal variability of nitrous oxide in a large eutrophic estuarine system: The Pearl River Estuary, China, Mar. Chem., 182, 14–24, 2016.

3. Pag. 3, L.12. "anaerobic particle interiors " . Do you mean anoxic particle interiors?

**Response**:

Sorry for this mistake. Revised (Page 4, Line 2).

4. Page 3, L 20 (de)nitrification. Why using ()?

**Response**:

We revised "which support (de)nitrification and $N_2O$ production" as "*which may support strong nitrification, denitrification, and $N_2O$ production*" (Page 4, Lines 9−10).

5. Page 4, L25. "Temperature and salinity were continuously measured with the CTD system. " Define continuously and in detail depths

**Response**:

Sorry for this misleading. We deleted "continuously". The sampling depths were described in Methods section 2.1 (Page 4, Lines 23−25) — "*water samples were taken from the surface (2 m) and bottom (4–15 m) of each site by using a conductivity, temperature, and depth (CTD) rosette sampling system (SBE 25; Sea-Bird Scientific, USA) fitted with 12 L Niskin bottles (General Oceanics).*"

6. Page 4. L24. "2.2 Biogeochemical parameters, $N_2O$ emissions, and isotopic analysis" Detail whether this is in the water column in your 11 sites or in experiments,

or both

**Response**:

Section 2.2 is for the water column in 22 sites in the PRE (11 sites in the upper reaches and 11 sites in the lower reaches). We revised the 2.2 title as "*Biogeochemical parameters, $N_2O$ emissions and isotopic analysis of environmental samples*" for clarification. All measurements for incubation experiments were described in Methods section 2.4 Incubation experiments.

Figure 1. Enlarge symbols + and * (or change colors). There is an extra red dot, isn't?
**Response**:

We enlarged symbols + and * in Figure 1 according to the editor's suggestion and revised the legend as "*Map of the PRE showing the sampling sites. Biogeochemical analyses were performed on samples from all sites (green and red circles). The green circles indicate sites where genes were analyzed. The black crosses indicate in situ incubation experiment sites (P01 and P05). The black asterisks indicate sites where the isotopic composition of $N_2O$ was analyzed.*" (Page 29, Lines 2−5).

Table 1. a) Fix typos such as "Liner Equation". Use either regression or equation b) Since this is regression, R2 is the coefficient of determination!
**Response**:

We revised "Liner Equation/Regression" as "Equation" and "$R$" as "$R^2$" in Table 2 (original Table 2) as suggested.

Figure 6. Since there is an equation (y = f(x)) with a line, it is a regression analysis (one independent and one dependent variable) whereas in correlation there are not dependent or independent variable. Coefficient must be R2 for regression.

**Response**:

Thanks for your comment. We deleted the regression lines and revised "R" as "$\rho$" in Figure 6 since the Mantel statistic was calculated as the Spearman rank correlation coefficient.

Page 7, L 19. Explain this please

**Response**:

Sorry for the confusion. We deleted Eq. (9) ($N_2O_{yield}$ (%) = $\Delta N_2O$-N / $\Delta(NO_2^- + NO_3^-)$-N) in the revised MS (Page 9, Line 1), which is not suitable to estimate $N_2O$ yield from nitrification in this study since denitrification could occur and nitrate and nitrite concentrations decreased in the ending phase of the incubation at site P01.

When the only nitrification occurs during incubation, the decrease of ammonia-N ($\Delta(NH_3 + NH_4^+)$-N is theoretically equal to the increase of nitrite/nitrate-N ($\Delta(NO_2^- + NO_3^-)$-N (part of nitrite may have been oxidized to nitrate). In this case, we can estimate the $N_2O$ yield based on either Eq (8) ($N_2O_{yield}$ (%) = $\Delta N_2O$-N / $\Delta(NH_3 + NH_4^+)$-N) or Eq (9).

Page 10, L10. Why is there a "… but… " here? This sentence is not clear.

**Response**:

We deleted "but". This sentence was revised as "*The ammonia and nitrite concentrations consistently decreased and increased, respectively, during the incubation experiments; the nitrate concentrations decreased in the end phase after a slight increase (Fig. 5b).*" (Page 12, Lines 11−13)

**Response to Reviewer #1**

**Anonymous Referee #1**

Ma et al. investigated the relationship between $N_2O$ production and spatial distribution of AOA and AOB along a salinity gradient in the Pearl River Estuary, China by using qPCR, chemical analysis and in situ incubation experiment. Data are well analyzed and presented. However, the manuscript's structure should be modified because the some results were presented in the discussion section, and some conclusions needs to be rephrased because the main findings in this study were mainly based on the correlation analysis OR statistical analysis (e.g., between $N_2O$ production and the abundance of functional genes), which can't provide a solid

support for a causal relationship between microbial contributors and $N_2O$ production.

**Response**:

Many thanks for the reviewer's comments. We moved the results pointed out by the reviewer into the Results section. We also revised some conclusion sentences with the appropriate tone according to the reviewer's suggestions. Please see below for detail.

More specific comments and suggestions are given below:

1. As mentioned by authors, both nirK and nirS genes are the key functional genes in the denitrification pathway, so why did not determine the abundance of nirK gene here?

**Response**:

The *nir*S and *nir*K genes encode cytochrome cd1 and copper-containing nitrite reductase, respectively. They were functionally and physiologically equivalent, but structurally different and could not be detected in the same strains in the previous research (Coyne et al., 1989), while the recent genomic analyses found a few bacteria contain both *nir*S and *nir*K (Graf et al., 2014). A recent genomic analysis revealed that a great many *nir*K-encoding bacteria have both denitrification and DNRA (Dissimilatory Nitrate Reduction to Ammonium) pathways (Helen et al., 2016). Furthermore, it was reported that *nir*S genes were more widely distributed than the *nir*K genes (Zumft, 1997; Bothe et al., 2000), and *nir*S genes were both more abundant and more diverse than *nir*K in the estuarine water columns (Zhu et al., 2018; Wang et al., 2019) and various estuarine sediments (Nogales et al, 2002; Santoro et al, 2006; Abell et al., 2010; Mosier and Francis, 2010; Beman, 2014; Smith et al., 2015; Lee and Francis, 2017). The previous study on the Pearl River sediment also showed that *nir*K abundance was much lower than *nir*S abundance (Huang et al., 2011). Therefore, we used the *nir*S gene to identify the distribution of denitrifiers in the PRE and reflect the denitrification potential.

[Figure]

**Figure R2 (Figure 4 in the revised MS):** RDA of the relative abundance of AOA *amo*A and AOB *amo*A under biogeochemical constraints. Each square represents an individual sample. Vectors represent environmental variables. $^*P < 0.05$, $^{**}P < 0.01$ (Monte Carlo permutation test).

5. Page 8, line 5-8 and Fig. 6. I am not convinced with the usage of Mantel and partial Mantel tests here due to two following reasons: 1) for ammonia oxidizer community, actually there were only four variables based on qPCR analysis (PA AOA, FL AOA, PA AOB and FL AOB) but not community data based on sequencing, so I don't think the results of qPCR reflected the truly community composition of ammonia oxidizers; and 2) the authors divided the environmental into four groups, but the classification seems a bit confusing. For example, why classify silicate into water mass but not substrate parameters? And TSM, DO and pH were classify as water mass parameters by numerous previous studies;

**Response:**

1) Sequencing-based community structure has higher resolutions than qPCR-based community structure. For community composition based on sequencing, the

dissimilarity matrices were calculated with the relative abundance of OTUs (Operational Taxonomic Units). Similarly, for community composition based on qPCR, the relative abundance of PA AOA, FL AOA, PA AOB, and FL AOB were used to calculate the dissimilarity matrices, just like merging some OTUs into one OTU. Despite lower resolutions of community composition, the dissimilarity matrices can be calculated and the Mantel and partial Mantel tests can be performed. Similarly, Castellano-Hinojosa et al. (2018) and Huang et al. (2011) also used qPCR data in NDMS analysis and CCA analysis of community structure.

2) Silicate has long been recognized as one of the most common indicators to trace river water in the ocean, and the low salinity and high silicate contents were the best indicators for river source (Moore, 1986). We added a three-dimensional scatter plot in the revised MS (Figure S1; see below) to show the relationships between potential temperature ($\theta$) (°C), salinity, and silicate ($SiO_3^{2-}$) concentration. The waters from the upper estuary, where the salinity of most sites was close to zero, had high potential temperature and silicate concentration. The mixing behaviors of waters occurred at the Humen outlet (sites P07 and A01), and the waters from the off-shore sites (A10 and A11) had high salinity and low potential temperature and silicate concentration. Therefore, we chose temperature, salinity, and silicate as the indicators to trace estuarine water masses and mixing. The related statements and explanations were added in the Discussion 4.2 subsection (Page 15, Lines 15−19).

We defined the substrate parameters as nitrogen substrates (ammonium, nitrite, and nitrate), which are related to the $N_2O$ producing processes nitrification and denitrification. TSM, DO, and pH are not conservative parameters and thus cannot trace water masses. These factors represent the biogeochemical characteristics of waters and could influence the availability of electron donors (or substrates) during nitrification and denitrification. For example, the suspended particles could be beneficial to microbial activity because of nutrients or substrates supply (Belser, 1979; Crump et al., 1998; Ouverney and Fuhrman, 2000; Teira et al., 2006; Zhang et al., 2014); DO concentration and pH also could influence the availability of ammonia, etc. (Geets et al., 2006; Ward, 2008; Martens-Habbena et al., 2009; Zhu et al., 2013; Huesemann et al., 2002; Hutchins et al., 2009; Fulweiler et al., 2010; Beman et al., 2011).

[Figure]

**Figure R3 (Figure S1 in the revised MS)**: Three-dimensional scatter plot of potential temperature ($\theta$) ($^{o}$C), salinity, and silicate ($SiO_3^{2-}$) concentration.

Table 2: "*ᵃThese rates are net rates since Δ(NH$_3$+NH$_4^+$) is the net consumption and ΔNO$_2^-$, ΔNO$_3^-$, and ΔN$_2$O is the net production during incubation.*" (Page 42)

Methods 2.4 subsection: "*All of the concentration-based rates described from the incubations represent net rates.*" (Page 8, Lines 26−27).

(2) We also agree that the [15]N-labeled methods are of high sensitivity, which is more reliable for low nitrification activity in natural environments (Damashek et al., 2016; Damashek and Francis 2018). However, in the nutrient-rich estuary waters, changes in nutrient concentrations (ammonium, nitrite, and nitrate) during incubations can be used to calculate nitrification rates when dissolved inorganic nitrogen is in balance (i.e. no nitrogen loss). In the upper-PRE, where high nitrification activity has been reported in the hypoxic zone (Dai et al., 2008; Hou et al., 2018), the in-situ concentrations of ammonium (33.3−167.2 μM), nitrite (11.6−24.5 μM), and nitrate (82.0−126.1 μM) at the incubation sites were high, so the changing of the nutrients can be sensitively detected during incubations.

(3) We compared our ammonium oxidation rates with the [15]N-labeled-based rates in the PRE from Hou et al. (2018) (see Table R1 below). Hou et al. reported that during the PRE cruises in July to August 2012 and September 2014, the nitrification rates in the bottom waters of the PRE were 40.25 to 40.70 μmol L$^{-1}$ d$^{-1}$ in the hypoxic sites. Actually, during our cruise, the nitrification rates in the upstream of Humen were also

measured using the [15]N-labeled method by simulating in-situ condition incubations, which ranged from 1.23-28.32 $\mu$mol $L^{-1}$ $d^{-1}$ (Zhang, 2016, Thesis). These [15]N-labeled-based nitrification rates are comparable to our estimated rates (11.28–26.88 $\mu$mol $L^{-1}$ $d^{-1}$) in the upper reach of PRE.

(4) We also compared our $N_2O$ yields with reported by Ji et al. (2018, GBC) (see Table R1 below). The $N_2O$ yields were 0.003−0.06% at >50 $\mu$M $O_2$ and >2% at <0.5 $\mu$M $O_2$ in the Eastern Tropical Pacific (Ji et al., 2018), which are 2−10-fold lower than those from the AOB strain cultures under the 10−100 $\mu$M $O_2$ concentration (Goreau et al., 1980; Table R1). Our $N_2O$ yield ranged from 0.21 to 0.32% during nitrification (the initial in-situ $O_2$ concentration: 30−61.3 $\mu$M; the terminal $O_2$ concentration: 0.7−2.5 $\mu$M). The estimated range of $N_2O$ yield is 0.16±0.09 to 0.37±0.23% when fitting our measured $O_2$ concentrations into the empirical equation of the relationship between $N_2O$ yield (%) from nitrification and $O_2$ concentration ($\mu$M) given by Ji et al. (2018), which was comparable with our measured $N_2O$ yield.

**Table R1** Nitrification rates/ammonia oxidation rates and $N_2O$ yield from literatures and our study.

| Study area/Microorganisms | Method | Nitrification rates ($\mu M\ day^{-1}$) | $NH_3$ concentrations ($\mu M$) | $N_2O$ yield | Reference |
|---|---|---|---|---|---|
| Rhône River plume | Nutrients + N-serve | 0.23 – 2.20 | 0 – 10 | – | Bianchi et al., 1994 |
| Mississippi River | Nutrients | 0 – 13.44 | 0.3 – 2.4 | – | Pakulski et al., 1995 |
| Mississippi & Atchafalaya River plume | Nutrients | 0 – 14.16 | 0.5 – 2.5 | – | Pakulski et al., 2000 |
| Scheldt | [14]C+ methylfluoride / Nutrients | Up to 19.2 / Up to 153.6 | 0 – 400 | 0.10–0.40% | De Wilde & De Bie, 2000 |
| Saanich Inlet | Nutrients + allyithiourea | 0 – 7.66 | 0 – 4.9 | – | Grundle & Juniper, 2011 |
| Pearl River | Nutrients + allyithiourea | 12.47 – 33.10[a] | 1.2 – 341.9 | – | Dai et al., 2008 |
| Pearl River | [15]N, denitrifier method | 40.25 – 40.70[b] | – | – | Hou et al., 2018 |
| Pearl River | [15]N, denitrifier method | 1.23–28.32 | – | – | Zhang, 2016 |
| Eastern Tropical Pacific | [15]N tracer | – | 0 – 0.5 | 0.003−0.06%[c] >2%[d] | Ji et al., 2018 |

| Species | Method | Rate | Ammonium | N2O yield | Reference |
|---|---|---|---|---|---|
| *Nitrosomonas europaea* | Nutrients, N$_2$O concentrations | 2.38- 23.8[f] | 7.14- 714.3[g] | 2.6–26%[e] 2.6–18%[h] | Yoshida & Alexander, 1970 |
| *Nitrosomonas sp.* (Marine) | Nutrients, N$_2$O concentrations | – | – | 0.26–0.99%[i] 2.5–9.9%[j] | Goreau et al., 1980 |
| *Nitrosomonas marina C-113a* | Nutrients, N$_2$O isotopic analyses | – | 50[g] | 0.04–2.2% | Frame & Casciotti, 2010 |
| Pearl River | Nutrients | 11.28–26.88[a] | 33.3 – 167.2 | 0.21–0.32% | This study |

[a] The ammonia oxidation rates observed at the upper reach of PRE in summer.

[b] The nitrification rates observed at the upper estuary where the O$_2$ concentration from 0.67 to 1.41 mg L$^{-1}$, which were little lower than that ranging from 0.9 to 2.0 mg L$^{-1}$ in our study.

[c] N$_2$O yield from ammonia oxidation under the O$_2$ >50 μM.

[d] N$_2$O yield from ammonia oxidation under the O$_2$ <0.5 μM.

[e] This experiment was designed to study the influence of different levels of ammonium concentration on N$_2$O formation by *Nitrosomonas europaea*.

[f] The ammonia oxidation rates were estimated based on the difference of ammonium concentrations between initial- and terminal- incubation time using the data from Yoshida and Alexander, 1970.

[g] Ammonium concentrations in the medium.

[h] This experiment was designed to study the influence of cells in different growth stages on N$_2$O formation by *Nitrosomonas europaea*.

[i] N$_2$O yield from ammonia oxidation under the O$_2$ ranging from 5–20% (56.3 –218.8 μM ).

[j] N$_2$O yield from ammonia oxidation under the O$_2$ ranging from 0.5–1% (5.6 –10.9 μM).

The strength of the correlation between genes and rates absolutely cannot be used to apportion a relative importance of one group of ammonia oxidizers or the other to the total rates. Nothing can be concluded from the data presented about who the important nitrifiers are. One possibility would be to obtain to a range of cell-specific ammonia oxidation rates from the literature and then use those in combination with the qPCR data to calculation the relative contribution of each group to the observed "rates."

**Response**:

The cell-specific ammonia oxidation rates, nitrite production rates, and $N_2O$ production rates from the literature on AOA and AOB strains varied in a very large range, due to the different species cultures, cell densities, cell stages, and incubation conditions such as $O_2$ or substrates concentrations (see Table R2 below). It is fairly uncertain to use these greatly varying cell-specific rates from cultures to estimate the contribution of AOA and AOB to the $N_2O$ production in natural environments. Notably, although the cell-specific $N_2O$ production rates from AOB and AOA strains varied greatly, the $N_2O$ yields from the AOB strains, ranging from 0.09 to 26 % (Table R2), were generally higher than the $N_2O$ yield from the AOA strains (0.002−0.09%; Table R2). In addition, the higher $N_2O$ yield from AOB has been observed in soils although the abundance of AOB was lower than AOA (Hink et al., 2017, 2018). We modified the discussion based on more literature on AOA and AOB cultures for better support. (Page 16, Lines 10−14).

We admit that the conclusions of this study mainly based on the correlation analysis and statistical analysis between multi-parameters. But there are two analyses providing more strong evidence supporting these statistical analyses:

(1) We attempted to accurately assess the relative contributions of AOA and AOB to $N_2O$ production in the PRE by plotting the $N_2O$ production rates (Fig. 7a) and yields (Fig. 7b) normalized to total (sum of AOA and AOB) *amo*A gene copies or transcripts at sites P01 and P05 along the x-y axes that represent the relative contributions of AOA and AOB to the total *amo*A gene or transcript pools.

Notably, compared to AOA, higher AOB abundance in the *amo*A gene-based DNA or cDNA pool resulted in distinctly higher (disproportionately higher relative to enhanced abundance) average *amo*A gene copy or transcript-specific $N_2O$ production rates (Fig. 7a) and yields (Fig. 7b), suggesting that AOB may have higher cell-specific activities in the upper estuary and thus be more active in producing $N_2O$ than AOA.

(2) The values of N stable isotopes in $N_2O$ ($\delta^{15}N$) were analyzed. The much lower $\delta^{15}N$-$N_2O$ (−27.9 to −12.6‰) upstream of the Humen outlet is consistent with AOB nitrification or denitrification processes, whereas enriched $^{15}N_2O$ (5.2−7.1‰) in the lower reaches approaches AOA nitrification and air $^{15}N$-$N_2O$ (Santoro et al., 2011). Taken together, the isotopic compositions of $N_2O$ (Fig. 2h in the MS) and $N_2O$ concentration distribution (Fig. 2e−g) suggest that the high concentrations of $N_2O$ (oversaturation) were produced from strong nitrification by AOB and probably concurrent minor denitrification in the upper estuary, however in the lower reaches, low concentrations of $N_2O$ could be explained by AOA nitrification or water atmospheric exchange of $N_2O$.

**Table R2** Cell-specific ammonia oxidation rates, cell-specific N$_2$O production rates, and N$_2$O yield from archaeal and bacterial strains.

| Microorganisms | Species (source of isolate) | Ammonia oxidation rates (fmol cell$^{-1}$ h$^{-1}$)[a] | N$_2$O production rates (fmol cell$^{-1}$ h$^{-1}$)[b] | N$_2$O yield[c] | Reference |
|---|---|---|---|---|---|
| AOA | | 19.0 | – | – | Martens-Habbena et al., 2009 |
| | *Nitrosopumilus maritimus* (Marine) | – | 0.02–1.01 | 0.002–0.026% | Löscher et al., 2012 |
| | | – | – | 0.03–0.05% | Stieglmeier et al., 2014 |
| | *Nitrososphaera viennensis* (Soil) | 2.6–2.8 | 0.004–0.005 | 0.03–0.09% | Stieglmeier et al., 2014 |
| AOB | *Nitrosomonas sp.* (Marine) | 2.0–15.4 | 0.04–0.21 | 0.26–9.9% | Goreau et al., 1980 |
| | *Nitrosomonas marina* (Marine) | 0.9–4.9 | – | – | Glover, 1985 |
| | | 13.7–31.3 | – | – | Glover, 1985 |
| | *Nitrosococcus oceanus* (Ocean) | 83.3 | – | – | Waston, 1965 |
| | | – | – | 0.26±0.1% | Goreau et al., 1980 |
| | *Nitrosomonas europaea* (Soil) | 12.4–18.3 | – | 2.6–26% | Yoshida & Alexander, 1970 |
| | | – | – | 0.47±0.1% | Goreau et al., 1980 |
| | *Nitrosospira tenuis* NV12 (Soil) | – | 0.002 | – | Shaw et al., 2006 |
| | *Nitrosomonas europaea* ATCC 19718 | – | 0.06 | – | Shaw et al., 2006 |
| | *Nitrosospira multiformis* (Soil) | – | – | 0.09–0.27% | Stieglmeier et al., 2014 |

| | | | | |
|---|---|---|---|---|
| *Nitrosolobus multiformis* (Soil) | – | – | 0.09±0.02% | Goreau et al., 1980 |
| *Nitrosospira briensis* (Soil) | – | – | 0.11±0.04% | Goreau et al., 1980 |

[a] The units for cell-specific ammonia oxidation rates in the citied references were unified as fmol cell$^{-1}$ h$^{-1}$.

[b] The units for cell-specific $N_2O$ production rates in the citied references were unified as fmol cell$^{-1}$ h$^{-1}$.

[c] The range of $N_2O$ yield of different cell densities under different $O_2$ conditions.

All the physical dynamics in the system have been reduced to a very naive "water mass" identification. Basic concepts in estuarine biogeochemistry are absent–for example, using salinity as a conservative tracer in a two-end member mixing model to determine production and loss of the various biogeochemical parameters.

**Response**:

According to the reviewer's suggestion, we performed the end-member mixing analysis. We

conclude that the two end-member model is not appropriate for the upper estuary where however, the major N₂O produced and which acts as a strong N₂O source of the PRE and is the highlight of this study. The two-end member mixing analysis in Lingdingyang (the mid-estuary and lower-estuary) reveal the removal of $N_2O$ and this removal is attributed to the water-air exchange. However, a significant positive correlation between the removal portion of $N_2O$ ($\Delta N_2O$) and ammonia consumption ($\Delta(NH_3+NH_4^+)$) suggests that the removal portion of $N_2O$ maybe mostly related to ammonia oxidation in the Lingdingyang surface water. Please see below for detail.

We plotted a three-dimensional scatter (Figure R3; see below) to show the relationships between potential temperature ($\theta$) ($^oC$), salinity, and silicate ($SiO_3^{2-}$) concentration. Silicate has long been recognized as one of the most common indicators to trace river water in the ocean, and the low salinity and high silicate contents were the best indicators for river source (Moore, 1986). The results indicate that the waters from the upper estuary, where the salinity of most sites was close to zero, had high potential temperature and silicate concentration. The mixing behaviors of waters occurred at the Humen outlet (sites P07 and A01), and the waters from the off-shore sites (A10 and A11) had high salinity and low potential temperature and silicate concentration.

Therefore, we divide the transect into the northern (upstream of the Humen outlet) and southern (Lingdingyang) areas to analyze the end-member mixing. Obviously, the two end-member model with salinity is not appropriate for the upper estuary since the salinity of the upper-estuary sites (upstream of Humen) all is close to zero and the upper estuary is highly impacted by the wastewater inputs, sewage discharged from Guangzhou, and different tributaries (Dai et al., 2006). We only discuss the estuarine mixing in Lingdinngyang using the end-member mixing model (Lin et al., 2016). The water from the Humen-outlet site (A01) with the lowest salinity was defined as the freshwater end-member and the water from the most off-shore site (A11) with the highest salinity was defined as the seawater end member. The mixing model is based on mass balance equations for salinity and the water fractions originating from the two end-members (Lin et al., 2016).

[Figure]

**Figure R3 (Figure S1 in the revised MS):** Three-dimensional scatter plot of potential temperature (θ) (°C), salinity, and silicate ($SiO_3^{2-}$) concentration.

The conservative mixing behavior of silicate along salinity was observed in Figure R4a. Based on the theoretical dilution line, $NO_2^-$ and $NO_3^-$ (weak) accumulations were observed in the Lingdingyang transect (Figure R4c and d); in contrast, $N_2O$ removal was observed in Figure R4e.

[Figure]

**Figure R4:** Mixing processes of (a) $SiO_3^{2-}$, (b) $NH_3+NH_4^+$, (c) $NO_2^-$, (d) $NO_3^-$, and (e) $N_2O$ in Lingdingyang area of the PRE. End-members applied in the mixing model are presented as black solid triangles and the dashed lines represent the theoretical dilution line.

We calculated the fractions of the two end-members based on mass balance and salinity and estimated the conservative concentrations of ammonia (($NH_3+NH_4^+)_{con}$), nitrite ($NO_2^-_{con}$), nitrate ($NO_3^-_{con}$), and $N_2O$ ($N_2O_{con}$) from conservative mixing by model prediction. The relationships between the field-observed concentrations of ammonium (($NH_3+NH_4^+_{obs}$)), nitrite ($NO_2^-_{obs}$), nitrate ($NO_3^-_{obs}$), and $N_2O$ ($N_2O_{obs}$) and the model-predicted conservative concentrations were shown in Figure R5. The results show that the points of ($NH_3+NH_4^+)_{obs}$ versus ($NH_3+NH_4^+)_{con}$ are mostly below the 1:1 line (Figure R5a), suggesting the consumption of ammonia, and the points of $NO_2^-_{obs}$ versus $NO_2^-_{con}$ are mostly above the 1:1 line, suggesting the nitrite addition (Figure R5b). The points of $NO_3^-_{obs}$ versus $NO_3^-_{con}$ are mostly near the 1:1 line (Figure R5c). The points of $N_2O_{obs}$ versus $N_2O_{con}$ in the surface water are mostly below the 1:1 line (Figure R5d), suggesting the

removal of N₂O. This removal is attributed to the water-air exchange (Lin et al., 2016). Notably, there is a significant positive correlation between the removal portion of N₂O (ΔN₂O) and ammonia consumption (Δ(NH₃+NH₄⁺)) in the Lingdingyang surface water (Figure R5f), suggesting that the removal portion of N₂O maybe mostly related to ammonia oxidation in the surface water.

[Figure]

**Figure R5:** The field-observed versus the model-predicted conservative concentrations of (a) ammonia, (b) nitrite, (c) nitrate, and (d) N₂O. (e) The relationship between the removal portion of N₂O (ΔN₂O) and ammonia consumption (Δ(NH₃+NH₄⁺)).

**Response**:

Sorry for the confusion. We only used Eq. (8) ($N_2O_{yield}$ (%) = $\Delta N_2O$-N / $\Delta(NH_3 + NH_4^+)$-N) to estimate $N_2O$ yield. Eq. (9) ($N_2O_{yield}$ (%) = $\Delta N_2O$-N / $\Delta(NO_2^- + NO_3^-)$-N) was deleted in the revised MS (Page 9, Line 1), which is not suitable to estimate $N_2O$ yield from nitrification in this study since denitrification could occur and nitrate and nitrite concentrations decreased in the ending phase

of the incubation at site P01.

In addition, we also compared the $N_2O$ yield estimated by Eq (8) and Eq (9) for site P05, where the only nitrification occurred during 12-hour incubation. The $N_2O$ yield estimated by Eq (8) and Eq (9) was 0.21% and 0.19%, respectively in the surface water and 0.32% and 0.33%, in the bottom water. They are almost equal.

More details are needed about how you arrived at the Schmidt number for $N_2O$. Is this the Raymond and Cole reference?

**Response**:

We added more details in Methods 2.2 subsection (Page 6, Lines 15−24). Please also see below.
"$k_{N_2O}$ was estimated using Eq. (6) according to Wanninkhof (1992):

$$k_{N_2O} = 0.31 \times u_{av}^2 \times (Sc_{N_2O}/600)^{-0.5} \tag{6}$$

where $u_{av}$ is the average wind speed at 10 m above the water surface. In this study, a $CO_2$ Schmidt number (Sc) of 600 at 20 ℃ in fresh water (Wanninkhof, 1992) was used for estuarine systems (Raymond and Cole, 2001). The Sc is defined as the kinematic viscosity of water divided by the diffusion coefficient of the gas and calculated from temperature (Wanninkhof, 1992). For $N_2O$ in waters with salinities <35 and temperatures ranging from 0−30 ℃, $Sc_{N_2O}$ was estimated using Eq. (7) according to Wanninkhof (1992): ..."

Need additional details of the calibration of the isotopic values.

**Response**:

We added more details of the calibration of the isotopic values in the revised version (2.2 subsection).

Page 7, Lines 1−7: "*The molecular ions of $N_2O$ ($N_2O^+$, m/z 44, 45, and 46) were quantified by IRMS to calculate isotope ratios for the entire molecule ($^{15}N/^{14}N$ and $^{18}O/^{16}O$). The $\delta^{15}N$ values of $N_2O$ in samples were calculated using the $^{15}N/^{14}N$ of the pure $N_2O$ reference gas and samples (Frame and Casciotti, 2010; Mohn et al., 2014). The reference gas was previously calibrated against $N_2O$ isotopic standard gas ($\delta^{15}N$ (vs Air-$N_2$) = −0.320‰) produced by Shoko Co. Ltd. (Tokyo, Japan) and the $\delta^{15}N$ value (vs Air-$N_2$) of the $N_2O$ reference gas is 6.579 ± 0.030‰. The precision of the method for $\delta^{15}N$-$N_2O$ was estimated as 0.3‰.*"

**Response**:

The estimated water–air $N_2O$ fluxes were 100.4 to 344.0 $\mu$mol m$^{-2}$ d$^{-1}$ upstream and decreased in Lingdingyang (42.4 to -2.6 $\mu$mol m$^{-2}$ d$^{-1}$). Taken together, the PRE was a strong source. We revised this sentence as "*Together, the PRE acts as a N2O source*" (Page 10, Line 16).

p.11 lines 19-26: This paragraph confuses some important concepts. Some of these numbers are the isotopic composition of $N_2O$ produced by ammonia oxidizers, but some of these numbers are the isotope effect (epsilon). Also, the isotopic composition of the $N_2O$ being produced by nitrification is dependent on the isotopic composition of the $NH_3$ being oxidized, for which no measurements or even estimates are provided.

**Response**:

All data in this paragraph are from literature. We only used the isotopic composition of $N_2O$ from literature in this paragraph and supplementary Table S2. But sorry for the wrong title and footnote of supplementary Table S2. We revised the title as "*Isotopic composition of N2O during bacterial and archaeal ammonia oxidation, bacterial nitrifier-denitrification, and bacterial denitrification.*" We revised the footnote as "*bAlthough the $\delta^{15}N$-N2O when using NH2OH as a substrate are listed here, the isotopic composition of N2O only when using NH4+ as a substrate was discussed in natural environments.*"

p.12 lines 15-17: Doesn't make sense to refer to 'water masses' in estuaries. There is a tremendous amount of mixing that leads to variation in these parameters. Just because something is a different salinity doesn't mean it's a different 'water mass.' These parameters are just 'hydrography.'

**Response**:

Here, we revised "water masses parameters" as "hydrographic parameters" (Page 11, Line 18). We

also revised "water masses" as "water mixing" in other places of the MS (Page 15, Lines 8, 10, 13, 15, and 19, and Page 37, Line 2).

p. 12 lines 15-28 and p. 13 lines 1-18 A lot of results presented that should be moved to the results section.

**Response**:

Thanks for the reviewer's suggestion. We moved the Spearman correlations and RDA analyses into the Results 3.3 subsection "Correlations between genes abundances and biogeochemical parameters" (Page 11, Lines 15−27, and Page 12, Lines 1−5).

p. 12 line 27 "ammonia oxidizer community" The use of the word "community" throughout the paper is confusing. More accurate to state the abundances of AOA and AOB?

**Response**:

We revised "ammonia oxidizer community" as "AOA and AOB distributions", and moved this part into the Results 3.3 subsection according to the reviewer's suggestion (Page 11, Line 25). The similar use of "community" in other places was also revised (Page 9, Lines 10 and 17; Page 34, Lines 2; Page 37, Lines 2, 5 and 6).

p. 24 Fig 1 i,j It looks like two different slopes in the data upstream and Lingdingyang. This could be quantified using a break point analysis.

**Response**:

Thanks for the suggestion. We re-quantified using a breakpoint analysis (see below) and revised the related description —"*notably, a significant negative relationship was observed between $\Delta N_2O_{excess}$ or $N_2O$ flux and DO (P < 0.01 for each) in the upper estuary (Fig. 2i and j).*" (Page 10, Lines 17−18)

[Figure]

**Figure R6:** (i) $\Delta N_2O$ vs. DO and (j) $N_2O$ flux vs. DO.

p. 32 I found this figure confusing. Perhaps it would be useful to have a table with the data presented in the figure? It is unclear using AOB and AOA% if the normalized $N_2O$ production values are a result of the $N_2O$ yield or low/high amoA abundance.

**Response**:

We added a supplementary Table S3 (below) showing the data in Figure 7. We plotted the $N_2O$ production rates (Fig. 7a) and yields (Fig. 7b) normalized to total (sum of AOA and AOB) *amo*A gene copies or transcripts at sites P01 and P05 along the x-y axes that represent the relative contributions of AOA and AOB to the total *amo*A gene or transcript pools. The results indicate that compared to AOA, higher AOB abundance in the *amo*A gene-based DNA or cDNA pool resulted in distinctly higher (disproportionately higher relative to enhanced abundance) average *amo*A gene copy or transcript-specific $N_2O$ production rates (Fig. 7a) and yields (Fig. 7b), suggesting that AOB may have higher cell-specific activities in the upper estuary and thus be more active in producing $N_2O$ than AOA. We modified the statements for clarification (Page 16, Lines 2−10).

**Table R3 (Table S3 in the MS)** The abundances of DNA- and cDNA-based *amo*A gene and the $N_2O$ production net rates and yields

normalized to total *amo*A gene copy or transcript numbers of AOA and AOB in a given sample at the incubation experiment sites.

| Site_ Layer | DNA-based AOB (All) (copies L$^{-1}$) | DNA-based AOA (All) (copies L$^{-1}$) | $N_2O$ production rates (All) (f mol cell$^{-1}$ h$^{-1}$) | $N_2O$ yields (All) (10$^{-6}$) | DNA-based AOB (PA) (copies L$^{-1}$) | DNA-based AOA (PA) (copies L$^{-1}$) | $N_2O$ production rates (PA) (f mol cell$^{-1}$ h$^{-1}$) | $N_2O$ yields (PA) (10$^{-6}$) | cDNA-based AOB (PA) (copies L$^{-1}$) | cDNA-based AOA (PA) (copies L$^{-1}$) | $N_2O$ production rates (PA) (f mol cell$^{-1}$ h$^{-1}$) | $N_2O$ yields (PA) (10$^{-6}$) |
|---|---|---|---|---|---|---|---|---|---|---|---|---|
| P05_S | 14030 | 34427 | 23.70 | 21.30 | 12125 | 29082 | 27.90 | 25.00 | 382928 | 138646 | 2.20 | 1.97 |
| P05_B | 87915 | 397740 | 2.90 | 3.25 | 77820 | 357308 | 3.24 | 3.63 | 89559 | 12559 | 13.80 | 15.50 |
| P01_S | 19623 | 642905 | 0.91 | 1.93 | 9343 | 578974 | 1.02 | 2.18 | 500 | 461578 | 1.30 | 2.77 |
| P01_B | 21334 | 251163 | 5.91 | 5.47 | 16458 | 221184 | 6.77 | 6.27 | 362 | 7436 | 206.00 | 191.00 |

S, surface; B, bottom; All, sum of particle-attached and free-living fractions; PA, particle-attached fraction.

Technical corrections

[revised manuscript text omitted]

p. 5 lines 5-23 Not all variables in the equations are defined.

**Response**:

We added more details for the equations and defined all variables in the revised MS (Page 6, Lines 1−24).

p. 31 Fig 6 Should axes be swapped?

**Response**:

We swapped X and Y axes in Fig. 6 of the revised version (Page 37).

[revised manuscript text omitted]

Table S3 The abundances of DNA- and cDNA-based *amo*A gene and the $N_2O$ production net rates and yields normalized to total *amo*A gene copy or transcript numbers of AOA and AOB in a given sample at the incubation experiment sites.

| Site Layer | DNA-based AOB (All) (copies $L^{-1}$) | DNA-based AOA (All) (copies $L^{-1}$) | $N_2O$ production rates (All) (fmol cell$^{-1}$ h$^{-1}$) | $N_2O$ yields (All) ($10^{-6}$) | DNA-based AOB (PA) (copies $L^{-1}$) | DNA-based AOA (PA) (copies $L^{-1}$) | $N_2O$ production rates (PA) (fmol cell$^{-1}$ h$^{-1}$) | $N_2O$ yields (PA) ($10^{-6}$) | cDNA-based AOB (PA) (copies $L^{-1}$) | cDNA-based AOA (PA) (copies $L^{-1}$) | $N_2O$ production rates (PA) (fmol cell$^{-1}$ h$^{-1}$) | $N_2O$ yields (PA) ($10^{-6}$) |
|---|---|---|---|---|---|---|---|---|---|---|---|---|
| P05_S | 14030 | 34427 | 23.70 | 21.30 | 12125 | 29082 | 27.90 | 25.00 | 382928 | 138646 | 2.20 | 1.97 |
| P05_B | 87915 | 397740 | 2.90 | 3.25 | 77820 | 357308 | 3.24 | 3.63 | 89559 | 12559 | 13.80 | 15.50 |
| P01_S | 19623 | 642905 | 0.91 | 1.93 | 9343 | 578974 | 1.02 | 2.18 | 500 | 461578 | 1.30 | 2.77 |
| P01_B | 21334 | 251163 | 5.91 | 5.47 | 16458 | 221184 | 6.77 | 6.27 | 362 | 7436 | 206.00 | 191.00 |

S, surface; B, bottom; All, sum of particle-attached and free-living fractions; PA, particle-attached fraction.

[Figure]

2  **Figure S1**: Three-dimensional scatter plot of potential temperature (θ) (°C), salinity,

3  and silicate ($SiO_3^{2-}$) concentration.

---

## Editor Decision (ED1)

Review of bg-2019-132 Major role of ammonia-oxidizing bacteria in N2O production in the Pearl River Estuary

August 7, 2019

Dear Dr. Zhang

Thanks for submitting responses to reviewers of bg-2019-132 "Major role of ammonia-oxidizing bacteria in $N_2O$ production in the Pearl River Estuary". Based on those and my own reading I think the article is promising due to sound data and interpretation but it is not publishable in the present form in Biogeosciences since it lacks some structure as pointed out by Reviewer 1. This aspect precludes full understanding of your work in mainly two aspects:

1) Mixing of results and discussion such as in Figure 2 that shows field data combined with panels i and j that, according to text, $\Delta N_2O$ is calculated for incubations so, they are probably results from incubations. Since there is no explanation of experimental treatments with variable $O_2$ concentration, I would assume that those are derived from field sampling, right? If that were the case, results are mixed with discussion in this figure and text in page 8, paragraph starting in L24. Alternatively, oxygenation conditions of incubations are missing in the method section.

2) Better explanation is needed regarding "concentration-based "rate" measurements …" (Reviewer 2) that not only refers to " … changing of the nutrients can be sensitively detected during incubations" as you pointed out, but also to multiple and simultaneous sources and sinks, therefore at the most you obtain a net rate since it is likely that these nutrients are simultaneously removed.

In any case, this aspect is not clear in the text. Please clearly show how you interpret each of rates (net production or decay/ incubation time) of Table 1:

$\Delta N_2O$ (nmol $L^{-1}h^{-1}$)

$\Delta NO_3^-$ (μmol $L^{-1}h^{-1}$)

$\Delta NO_2^-$ (μmol $L^{-1}h^{-1}$)

$\Delta(NH_3 + NH_4^+)$ (μmol $L^{-1}h^{-1}$). You mean $NH_3 + NH_4^+$ instead of $NH_3/NH_4^+$ (a ratio), don't you?

A cartoon similar to the one below may help to explain your rationale for interpreting these rates and better sustain conclusions such as "… results clearly indicate that nitrification occurred during the entire P01 incubations, and suggest that denitrification may be present in the ending phase… " (Page 10, L11-12).

[Figure]

In addition, please consider the following:
1. Pag. 1. Line 19. All " $N_2O$ parameters " . Do you mean $N_2O$-related parameters?
2. P. 1, Line 22-25 "Taken together, the in situ incubation experiments, $N_2O$ isotopic composition and concentrations, and gene datasets suggested that the high concentration of $N_2O$ (oversaturated) is mainly produced from strong nitrification by the relatively high abundance of AOB in the upper reaches as the major source of $N_2O$ emitted to the atmosphere in the whole estuary. "

     What is the evidence for the whole estuary? What about seasonal variability?

3. Pag. 3, L.12. "anaerobic particle interiors " . Do you mean anoxic particle interiors?

4. Page 3, L 20 (de)nitrification. Why using ()?

5. Page 4, L25. "Temperature and salinity were continuously measured with the CTD system. " Define continuously and in detail depths

6. Page 4. L24. "2.2 Biogeochemical parameters, N2O emissions, and isotopic analysis " Detail whether this is in the water column in your 11 sites or in experiments, or both

Figure 1. Enlarge symbols + and * (or change colors). There is an extra red dot, isn't?

Table 1. a) Fix typos such as "Liner Equation". Use either regression or equation

 b) Since this is regression, R2 is the coefficient of determination!

Figure 6. Since there is an equation (y = f(x)) with a line, it is a regression analysis (one independent and one dependent variable) whereas in correlation there are not dependent or independent variable. Coefficient must be $R^2$ for regression.

Page 7, L 19. Explain this please
Page 10, L10. Why is there a "… but… " here? This sentence is not clear.

Sincerely your

Silvio Pantoja
Associate Editor

---

## Author Response (AR2)

Dear Dr. Pantoja,

Thank you very much for the valuable comments and careful revisions, based on which we have revised the manuscript again. Our responses to all comments are listed below. Thank you again for your time and kind efforts.

Best wishes,

Yao Zhang

**Manuscript Number: bg-2019-132**

**Manuscript title: Major role of ammonia-oxidizing bacteria in N$_2$O production in the Pearl River Estuary**

**Response to Editor**

Comments to the Author:
Review of bg-2019-132
October 25, 2019

Dear authors,

Thank you for providing a revised version of the manuscript. From my point of view some of the aspects brought up by the reviewers could be addressed more carefully:

1. Abstract, L.10 "However, biological sources of N$_2$O in estuarine ecosystems remain controversial, but are of great importance for understanding global N$_2$O emission patterns."

"However, knowledge on or discrimination of biological sources of N$_2$O in estuarine ecosystems remains controversial, but are of great importance for understanding global N$_2$O emission patterns. "

Knowing "biological sources of N$_2$O is of great importance for understanding global N$_2$O emission patterns", not sources by themselves. Please rephrase accordingly

**Response**:

We thank the editor for this correction. We have revised this sentence as "*However, knowledge on biological sources of N$_2$O in estuarine ecosystems remains controversial, but are of great importance for understanding global N$_2$O emission*

*patterns.*" (Page 1, Lines 9−10)

2. Abstract, L. 14. "Our results indicated that nitrification predominantly occurred, with significant $N_2O$ production during ammonia oxidation, in the hypoxic waters of the upper estuary where the maximum $N_2O$ and $DN_2O$ excess concentrations were observed, although minor denitrification might be concurrent at the site with the lowest dissolved oxygen." This sentence is difficult to understand as it is. Do you mean this?:

"Our results indicated that nitrification predominantly occurred, with significant $N_2O$ production during ammonia oxidation. In the hypoxic waters of the upper estuary where the maximum $N_2O$ and $DN_2O$ excess concentrations were observed, although minor denitrification might be concurrent at the site with the lowest dissolved oxygen."

**Response**:

We thank the editor for this correction. We have revised this sentence as "*Our results indicated that nitrification predominantly occurred, with significant $N_2O$ production during ammonia oxidation. In the hypoxic waters of the upper estuary, strong nitrification resulted in the observed maximum $N_2O$ and $\Delta N_2O_{excess}$ concentrations, although minor denitrification might be concurrent at the site with the lowest dissolved oxygen.*" (Page 1, Line 14–18)

3. Page 4, L. 5. "The Pearl River Estuary (PRE) is one of the world's most complex estuarine systems with a total discharge of 285.2×109 m3 yr−1 (Dai et al., 2014). The PRE is surrounded by complex regions with a rich nitrogen supply that produces eutrophic waters (Dai et al., 2008)."

complex is repeated in both continuous sentences.

**Response**:

We have revised this sentence as "*The Pearl River Estuary, surrounded by several big cities, is one of the world's most complex estuarine systems with a total discharge of $285.2 \times 10^9$ $m^3$ $yr^{-1}$ (Dai et al., 2014). A rich nitrogen supply with the river discharge produces eutrophic waters in the estuary (Dai et al., 2008).*" (Page 4, Line 5–7).

4. Discussion, How is that "The in situ incubation experiments clearly indicated that nitrification predominantly occurred …" from Figure 5?

**Response**:

This sentence was based on the results presented in Figure 2 and Figure 5. Figure 2 showed the distribution of biogeochemical factors along the estuary transect, including the $N_2O$ related parameters, dissolved oxygen concentration, etc. **Sites P01 and P05 were located in the hypoxic zone (DO < 63.0 μmol L$^{-1}$; Rabalais et al., 2010)** and the maximum $N_2O$ and $\Delta N_2O_{excess}$ concentrations were observed at P01 and P05. Figure 5 showed the incubations experiments results of P01 and P05, indicating that nitrification predominantly occurred at the both sites (i.e. in the hypoxic waters). The detailed explanation on Figure 5 results have been stated in the Result 3.4 subsection.

For a clearer statement, we have revised this sentence as "*The in situ incubation experiments clearly indicated that nitrification predominantly occurred in the hypoxic waters (e.g. both the P01 and P05 sites) of the upper estuary along with significant $N_2O$ production, and suggested that denitrification could be concurrent at the lowest DO site (P01) where the maximum $N_2O$ and $\Delta N_2O_{excess}$ concentrations were observed (Figs. 2 and 5).*" (Page 13, Lines 20–24)

**Reference:**

Rabalais, N. N., Díaz, R. J., Levin, L. A., Turner, R. E., Gilbert, D., and Zhang, J.: Dynamics and distribution of natural and human-caused hypoxia, Biogeosciences, 7, 585–619, 2010.

5. Show Lingdingyang in Fig. 1.

**Response**:

We have revised Figure 1 as suggested.

6. Caption Fig. 1. Biogeochemical analyses are in red and green, but gene analyses are green; same color? This is confusing for readers. Please modify

**Response**:

We have revised the Fig. 1 legend as "… *Biochemical parameters analyses were performed on samples from all sites (green and red circles). The green circles indicate sites where genes were* ==*additionally*== *analyzed.* …" (Page 29, Lines 2−4)

7. Figure 2. Is Upper estuary in graphs i and j same as upstream of the Humen outlet? Please normalize names and explain.

**Response**:

Yes, they are the same. We have revised "upper estuary" as "upstream of the Humen outlet" in Figure 2i and j and the corresponding text (Page 10, Line 18) as suggested.

We also have explained "upper estuary = upstream of the Humen outlet" in the Methods 2.1 subsection of the manuscript. —"*A total of 22 sites along the salinity gradient of the Pearl River Estuary were sampled during a research cruise in July 2015, including* ==*11 sites in the upper reaches (upstream of the Humen outlet)*== *and 11 sites in the lower reaches (Lingdingyang) (Fig. 1).*" (Page 4, Lines 21−23). So we kept the use of "upper estuary" in some other places of the manuscript.

8. Page 10, lines 20-22. "Overall, upstream of the Humen outlet was characterized by hypoxic waters rich in nitrogen-based nutrients, where ammonium concentrations decreased and the sum of nitrite and nitrate concentrations increased seaward, corresponding to distinctly higher N2O fluxes released to the atmosphere. "

a) In Fig. 2, only one data point (P01) is almost hypoxic (ca. 25 micromol O2 /L.

**Response**:

According to Rabalsais et al. (2010, Biogeoscience) indicating that "…a commonly used definition of hypoxia is dissolved oxygen < 2 mg $L^{-1}$, the equivalent of 1.4 ml $L^{-1}$ or 63 $\mu M$", we used 63 μM as a definition of hypoxia in this manuscript. We have stated this definition in subsection 3.1 —"*The DO concentrations were distinctly lower upstream of the Humen outlet with nearly one-half of the samples below* ==*the hypoxic threshold (63.0 $\mu mol\ L^{-1}$; Rabalais et al., 2010)*==". (Page 10, 5–7)

In addition, a commonly used definition of "==suboxic==" is DO < 10 μM and "==anoxic==" is ==DO $\approx$ 0 μM== according to Stramma et al. (2008, Science) —"Regions with oxygen concentrations below about 10 μmol $kg^{-1}$ are termed suboxic." "Anoxic regions have no dissolved oxygen."

Babbin et al. (2015, Science) also stated a similar definition of suboxic waters —"The most concentrated oceanic sources of $N_2O$ to the atmosphere are the suboxic (0 to 20 μmol $L^{-1}$) waters."

b) "…increased seaward, corresponding to distinctly higher N2O fluxes released to the atmosphere". Where are you seeing "higher N2O fluxes released" in Fig. 2d?
Response:
Figure 2g showed $N_2O$ flux. The plus values of this parameter represent the excess $N_2O$ in water release to the atmosphere through the air-sea interface. We have stated the detailed Fig. 2g results directly above this sentence (Page 10, Lines 14−17).

c) In the discussion, the hypoxia issue continues: "The in situ incubation experiments clearly indicated that nitrification predominantly occurred in the hypoxic waters of the upper estuary along with significant N2O production, and suggested that denitrification could be concurrent at the lowest DO site (P01) where the maximum N2O and DN2Oexcess concentrations were observed."
This aspect needs to be fixed throughout the whole ms.
Response:
Please refer to our above response. According to Rabalsais et al. (2010, Biogeoscience) indicating that "…a commonly used definition of hypoxia is dissolved oxygen < 2 mg $L^{-1}$, the equivalent of 1.4 ml $L^{-1}$ or 63 $\mu M$", we used 63 μM as a definition of hypoxia in this manuscript. We have stated this definition in subsection 3.1 —"*The DO*

*concentrations were distinctly lower upstream of the Humen outlet with nearly one-half of the samples below* the hypoxic threshold (63.0 μmol L$^{-1}$; Rabalais et al., 2010)*".* (Page 10, 5–7)

For a clearer statement, we also have revised this sentence as "*The in situ incubation experiments clearly indicated that nitrification predominantly occurred* in the hypoxic waters (e.g. both the P01 and P05 sites) *of the upper estuary along with significant N$_2$O production, and suggested that denitrification could be concurrent at the lowest DO site (P01) where the maximum N$_2$O and $\Delta$N$_2$O$_{excess}$ concentrations were observed* (Figs. 2 and 5)*."* (Page 13, Lines 20–24)

9. Could you avoid non-standard abbreviations such as PRE, PA, FA, FDR, RDA, etc. It is very difficult to follow for readers
**Response**:

Thanks for this comment. We have removed these abbreviations throughout the manuscript.

[revised manuscript text omitted]

[a]$O_2$ conditions of the incubation experiments.

[b]Although the $\delta^{15}$N-N$_2$O when using NH$_2$OH as a substrate are listed here, the isotopic composition of N$_2$O only when using NH$_4^+$ as a substrate was discussed in natural environments.

**Table S3** The abundances of DNA- and cDNA-based *amo*A gene and the $N_2O$ production net rates and yields normalized to total *amo*A gene copy or transcript numbers of AOA and AOB in a given sample at the incubation experiment sites.

| Site_Layer | DNA-based AOB (All) (copies $L^{-1}$) | DNA-based AOA (All) (copies $L^{-1}$) | $N_2O$ production rates (All) (fmol $cell^{-1} h^{-1}$) | $N_2O$ yields (All) ($10^{-6}$) | DNA-based AOB (PA) (copies $L^{-1}$) | DNA-based AOA (PA) (copies $L^{-1}$) | $N_2O$ production rates (PA) (fmol $cell^{-1} h^{-1}$) | $N_2O$ yields (PA) ($10^{-6}$) | cDNA-based AOB (PA) (copies $L^{-1}$) | cDNA-based AOA (PA) (copies $L^{-1}$) | $N_2O$ production rates (PA) (fmol $cell^{-1} h^{-1}$) | $N_2O$ yields (PA) ($10^{-6}$) |
|---|---|---|---|---|---|---|---|---|---|---|---|---|
| P05_S | 14030 | 34427 | 23.70 | 21.30 | 12125 | 29082 | 27.90 | 25.00 | 382928 | 138646 | 2.20 | 1.97 |
| P05_B | 87915 | 397740 | 2.90 | 3.25 | 77820 | 357308 | 3.24 | 3.63 | 89559 | 12559 | 13.80 | 15.50 |
| P01_S | 19623 | 642905 | 0.91 | 1.93 | 9343 | 578974 | 1.02 | 2.18 | 500 | 461578 | 1.30 | 2.77 |
| P01_B | 21334 | 251163 | 5.91 | 5.47 | 16458 | 221184 | 6.77 | 6.27 | 362 | 7436 | 206.00 | 191.00 |

S, surface; B, bottom; All, sum of particle-attached and free-living fractions; PA, particle-attached fraction.

[Figure]

**Figure S1**: Three-dimensional scatter plot of potential temperature ($\theta$) ($^{o}$C), salinity, and silicate ($SiO_3^{2-}$) concentration. PRE, Pearl River Estuary.

---

## Editor Decision (ED2)

Review of bg-2019-132

October 25, 2019

Dear authors,
Thank you for providing a revised version of the manuscript.  From my point of view some of the aspects brought up by the reviewers could be addressed more carefully:

1. Abstract, L.10 "However, biological sources of N2O in estuarine ecosystems remain controversial, but are of great importance for understanding global N2O emission patterns. "

"However, knowledge on or discrimination of biological sources of N2O in estuarine ecosystems remains controversial, but are of great importance for understanding global N2O emission patterns. "

   Knowing "biological sources of N2O is of great importance for understanding global N2O emission patterns", not sources by themselves.  Please rephrase accordingly

2. Abstract, L. 14.  "Our results indicated that nitrification predominantly occurred, with significant N2O production during ammonia oxidation, in the hypoxic waters of the upper estuary where the maximum N2O and $\Delta$N2O excess concentrations were observed, although minor denitrification might be concurrent at the site with the lowest dissolved oxygen. "

   This sentence is difficult to understand as it is. Do you mean this?:

   "Our results indicated that nitrification predominantly occurred, with significant N2O production during ammonia oxidation.  In the hypoxic waters of the upper estuary where the maximum N2O and $\Delta$N2O excess concentrations were observed, although minor denitrification might be concurrent at the site with the lowest dissolved oxygen. "

3. Page 4, L. 5.  "The Pearl River Estuary (PRE) is one of the world's most **complex** estuarine systems with a total discharge of 285.2×109 m3 yr−1 (Dai et al., 2014).  The PRE is surrounded by **complex** regions with a rich nitrogen supply that produces eutrophic waters (Dai et al., 2008)."

**complex** is repeated in both continuous sentences.

4. Discussion, How is that "The in situ incubation experiments **clearly** indicated that nitrification predominantly occurred ..." from Figure 5?

5. Show Lingdingyang in Fig. 1.

6. Caption Fig. 1. Biogeochemical analyses are in red and green, but gene analyses are green; same color? This is confusing for readers. Please modify

7. Figure 2. Is Upper estuary in graphs i and j same as upstream of the Humen outlet? Please normalize names and explain.

8. Page 10, lines 20-22. "Overall, upstream of the Humen outlet was characterized by hypoxic waters rich in nitrogen-based nutrients, where ammonium concentrations decreased and the sum of nitrite and nitrate concentrations increased seaward, corresponding to distinctly higher N2O fluxes released to the atmosphere. "
    a) Un Fig. 2, only one data point (P01) is almost hypoxic (ca. 25 micromol O2 /L.
    b) "...increased seaward, corresponding to distinctly higher N2O fluxes released to the atmosphere". Where are you seeing "higher N2O fluxes released" in Fig. 2d?
    c) In the discussion, the hypoxia issue continues: "The in situ incubation experiments clearly indicated that nitrification predominantly occurred in the hypoxic waters of the upper estuary along with significant N2O production, and suggested that denitrification could be concurrent at the lowest DO site (P01) where the maximum N2O and ΔN2Oexcess concentrations were observed. "

    This aspect needs to be fixed throughout the whole ms.

9. Could you avoid non-standard abbreviations such as PRE, PA, FA, FDR, RDA, etc.  It is very difficult to follow for readers

Please provide a point-to-point response how you addressed the individual aspects and if not, why you do not agree.  Thank you for your patience with the evaluation process and for choosing Biogeosciences for this publication.

Sincerely yours

Silvio Pantoja
Associate Editor